



# Groundwater mean transit times, mixing and recharge in faulted–hydraulic drop alluvium aquifers using chlorofluorocarbons (CFCs) and tritium isotope ($^3$H)

Bin Ma[1,2], Menggui Jin[1,2,3], Xing Liang[1,4], and Jing Li[1]

[1]School of Environmental Studies, China University of Geosciences, Wuhan, 430074, China
[2]State Key Laboratory of Biogeology and Environmental Geology, China University of Geosciences, Wuhan, 430074, China
[3]Laboratory of Basin Hydrology and Wetland Eco–restoration, China University of Geosciences, Wuhan, 430074, China
[4]Hubei Key Laboratory of Wetland Evolution & Ecological Restoration, China University of Geosciences, Wuhan, 430074, China

*Correspondence to*: Menggui Jin (mgjin@cug.edu.cn)

**Abstract.** Documenting the transit times and recharge sources of mixed groundwater is crucial for water resource management in the alluvium aquifers of arid basin. Environmental tracers (CFCs, $^3$H, $^{14}$C, $\delta^2$H, $\delta^{18}$O) and hydrochemistry of mixed groundwater were used to assist our understanding of groundwater mean transit times (MTTs), mixing and aquifer recharge in faulted–hydraulic drop alluvium aquifers in the Manas River Basin (China). $^3$H activities of groundwater

decrease with distance to the mountain from 41.1–60 TU in the Manas River upstream (south of the fault), indicating the rainfall recharge since the 1960s, to as low as 1.1 TU in the downstream (north of the fault). Carbon–14 groundwater ages increase with distance (3000–5000 yrs in the midstream to > 7000 yrs in the downstream) and depth, as well as with more depleted $\delta^{18}$O values, confirming that the deeper groundwater is derived from paleometeoric recharge in the semi–confined groundwater system. MTTs estimated using an exponential–piston flow model vary from 19 to 101 yrs for CFCs and from

19 to 158 yrs for $^3$H, which show much longer MTTs for $^3$H than CFCs may be due to the time lag through the thick unsaturated zone. The thrust faults were found to play a paramount role on groundwater flow paths and MTTs due to their block water features, where the relatively long MTTs were found near the Manas City with shorter distance and smaller hydraulic gradients. The remarkable correlations between CFCs rather than $^3$H MTTs and pH, SiO$_2$ and SO$_4^{2-}$ concentrations allow first–order proxies of MTTs for groundwater at different times to be made. The quite 'modern' recharge in the south of

the fault with young (post–1940) water fractions of 87–100 % is obtained, while in the north of the fault in the midstream area the young water fractions vary from 12 to 91 % based on the CFC binary mixing model. This study shows that the combination of CFCs and $^3$H transit time tracers have potential to study groundwater MTTs and identify the recharge sources for the different mixing end–members.



## 1 Introduction: mean transit times

Groundwater supplies the world's largest freshwater resource to billions of people, and plays a central part in energy and food security, human health and ecosystems (Gleeson et al., 2016). Documenting the transit times of groundwater from recharge to drainage in pumping wells, springs, or streams reveals information about water storage, mixing and transport in subsurface water systems (Cartwright et al., 2017; Dreuzy and Ginn, 2016; McGuire and McDonnell, 2006). It is particularly crucial in the alluvium aquifer where the fresh groundwater may be renewable and potable resources rather than saline

groundwater (Han et al., 2011; Huang et al., 2017), as well as more vulnerable to anthropogenic contaminants and land–use changes (Morgenstern and Daughney, 2012).

Mean transit times (MTTs) that determined from the transit time distributions (TTDs) cannot be measured directly in the field and may be inferred from transit time tracer concentrations by using the lumped parameter models (LPMs; Małoszewski and Zuber, 1982; Jurgens et al., 2012), which has commonly assumed that the hydrologic system is in a

steady–state. Three types of transit time tracers have been widely referred to in the previous literatures. First, as the water molecules, the isotopes of water ($^{18}O$, $^{2}H$, $^{3}H$) make them ideal tracers for determining MTTs shorter than ~5 yrs with stable isotopes (Kirchner et al., 2010; McGuire et al., 2005; Stewart et al., 2010) and up to ~100 yrs with $^{3}H$ (Beyer et al., 2016; Cartwright and Morgenstern, 2015, 2016; Morgenstern et al., 2010). Second, solute tracers like the radioisotopes of $^{14}C$, $^{36}Cl$ and noble gases ($^{4}He$, $^{85}Kr$, $^{39}Ar$, and $^{81}Kr$), as well as the atmosphere concentrations of synthetic organic compounds

(chlorofluorocarbons, CFC–11, CFC–12, and CFC–113; and sulfur hexafluoride, $SF_6$), permit their use in determining groundwater MTTs with much wider variabilities (decades to hundred millenniums) due to the radioisotopes long–span half–lives (Aggarwal, 2013). Third, major ion concentrations like the inert chloride (Cl) ion determine the MTTs in a similar function with the stable isotopes depending on damping of seasonal variation input cycles on passing through a system into the output. MTTs determined from the seasonal tracer cycle method (e.g., stable isotope values or Cl concentrations) which,

requires detailed such as weekly or sub–weekly time series measurements, may be more appropriate for water drainage through catchment and discharging into stream (Hrachowitz et al., 2009; Kirchner et al., 2010; McGuire et al., 2005) over for groundwater system. Nevertheless, strong correlation of major ion concentrations with groundwater age permitting the hydrochemistry to be used as proxy or complementary for age via previously established relationship in close lithological conditions (Beyer et al., 2016; Morgenstern et al., 2010, 2015).

Tritium ($^{3}H$), with a half–life of 12.32 yrs, is the only true age tracer for waters (Tadros et al., 2014). The Northern Hemispheric $^{3}H$ activities were several orders of magnitude higher than in the Southern Hemisphere (Clark and Fritz, 1997; Tadros et al., 2014) due to the atmospheric thermonuclear tests in the Northern Hemisphere between the 1950s and 1960s, which also resulted in mean annual $^{3}H$ activities peaks reaching several hundred times natural levels in the Northern Hemisphere. As a result, the nowadays rainfall $^{3}H$ activities are still affected by the tail–end of the bomb–pulse in the

Northern Hemisphere, which are particularly high in the arid northwest China due to both the continental effect (Tadros et al., 2014) and superimposed over the China atmospheric nuclear tests from 1964 to 1974. Only about 5 % of the bomb–test $^{3}H$



activities mixed via the stratospheric circulation into the Southern Hemisphere (Morgenstern et al., 2010), and the [3]H activities of remnant bomb–pulse waters have now decayed below those of modern rainfall. Accordingly, recent studies in estimating MTTs by [3]H in the Southern Hemisphere (e.g., Australia, Cartwright et al., 2018; Cartwright and Morgenstern,

2015, 2016; and New Zealand, Morgenstern et al., 2010, 2015; Morgenstern and Daughney, 2012) have used single [3]H rainfall measurements as the input data. Time–series [3]H measurements may be used to estimate MTTs in the Northern Hemisphere by LPMs (Han et al., 2007; Han et al., 2015).

Contrasting to [3]H, CFCs have relatively long transit times in the atmosphere of between 44 yrs (CFC–11) and 180 yrs (CFC–12), which permits their uniform atmospheric distributions over large areas, but there is 1–2 yrs lag time for Southern

Hemisphere than Northern Hemisphere (Cartwright et al., 2017; Cook et al., 2017; Darling et al., 2012). The build–up in the atmosphere of CFCs after the 1950s combining with their solubility (despite low solubility) in water have commonly permitted them as indicators of transit times in groundwater up to ~60 yrs (Darling et al., 2012; Han et al., 2012). Though the atmospheric concentrations of CFC–11, CFC–12 and CFC–113 have declined between 1994 and 2002 (different CFCs peaked at different times; Cook et al., 2017) and thus there are rooms for some ambiguity in the CFC ratios plot (Darling et

al., 2012), the different atmospheric CFCs ratios between nowadays and before the 1990s (Plummer et al., 2006b) make determining groundwater MTTs by CFCs possible. Consequently, CFCs have been commonly viewed as alternatives to [3]H for calculating transit times in groundwater as the bomb–pule [3]H activities declined (Cartwright et al., 2017; Cook et al., 2017; Qin et al., 2011). However, the CFCs input would be limiting for estimating groundwater MTTs when they are entrapped excess air in the unsaturated zone during recharge (under–estimation MTTs; Cook et al., 2006; Darling et al.,

2012), contaminated in urban and industrial environments (under–estimation MTTs; Carlson et al., 2011; Han et al., 2007; Mahlknecht et al., 2017; Qin et al., 2007), or degraded in anaerobic groundwater (over–estimation MTTs; most notably CFC–11 and CFC–113; Cook and Solomon, 1995; Horneman et al., 2008; Plummer et al., 2006b) in the confined aquifers.

Additionally, mixing between waters of different ages which, occurs both within the aquifer and pumping from long–screened wells (Cook et al., 2017; Visser et al., 2013), may result in difficulty of interpreting transit time tracers data like the

aggregation errors that calculated MTTs being less than actual MTTs in mixed waters (Cartwright and Morgenstern, 2016; Kirchner, 2016; Stewart et al., 2017). MTTs estimation from the multi–model approach based on incorporated transit time tracers should reduce the calculation uncertainty (Green et al., 2016; Visser et al., 2013) and verify whether the MTTs can be realistically estimated (Cartwright et al., 2017).

Mixing within the aquifers and pumping from the long–screened wells is particularly true in the faulted–hydraulic drop

alluvium aquifers of the Manas River Basin (MRB) in the arid northwest China (Fig. 1). Pumping from the long–screened over 10 000 boreholes (Ma et al., 2018) make groundwater mixing most likely. The water table depth is 180 m and a level difference of 130 m hydraulic drop is observed due to the thrust fault in the alluvium aquifer (Fig. 1c). The MTTs that impacted by multiple scale mixing in the aquifers with contrasting geological settings and deep unsaturated zone are still insufficient recognition. This study was designed to provide the first estimation of the MTTs of borehole groundwater

drainage (e.g., well withdrawal) using CFC and [3]H concentrations. The major ion hydrochemistry of groundwater as first–



order proxies for MTTs were then analyzed. In addition, we identify the recharge sources for the different mixing end–members and constrain mixing rates.

## 2 Geological and hydrogeological setting

The bedrock of the upper Manas River catchment in the mountain area consists of granites, sedimentary formations of Devonian and Carboniferous age, and Mesozoic limestone (Jelinowska et al. 1995). The pyroclastic rock is exposed in relatively small areas in the south mountain. The piedmont and oasis plain are filled with Cenozoic strata including the Tertiary and Quaternary deposits with a total depth more than 5000 m in the piedmont area and decreasing to 500–1000 m in the center of the plain (Zhao 2010). The vertical cross section (Fig. 1c) shows that the Quaternary deposits consist of pebbles, sandy gravel and sand in the piedmont plain. The clay content in the Quaternary deposits increases from the overflow spring zone to the north oasis plain, which consists of silty loam and clay. The Huoerguosi–Manas–Tugulu thrust fault–anticline occurred in the early Pleistocene and cut the Tertiary strata with totally length of approximately 100 km length in the piedmont alluvial fan (Figs. 1), which are water block features. This thrust fault–anticline was intermittent activity from the middle late–Pleistocene and then tended to be more active from the late Holocene (Cui et al., 2007).

In the mountain area, groundwater consists of metamorphic rock fissure water, magmatic rock fissure water, clastic rock fissure water and Tertiary clastic rock fissure water. In the piedmont plain of Shihezi (SHZ) zone, groundwater is from a single–layer unconfined aquifer. From the overflow spring zone to the central oasis plain, groundwater consists of shallow unconfined water and deep confined water. The hydraulic gradient, hydraulic conductivity and transmissivity show a large range of variations due to changes in grain size and local increases of clay content (Wu 2007). The groundwater flow direction is macroscopically similar to the Manas River flow. In the piedmont plain, the aquifer is recharged by the Manas River water and unconfined with saturated thickness more than 650 m, and is hydraulically connected to the hydrological network in the piedmont plain and north oasis plain (Ma et al., 2018; Wu 2007). The piedmont plain unconfined aquifer buried depth decreases gradually from south to north and has relatively fresh groundwater with TDS of $< 1$ g $L^{-1}$. Groundwater discharges via springs in the north area of SHZ (Fig. 1c). The shallow unconfined groundwater in the north oasis plain has TDS of $> 3$ g $L^{-1}$, and the underlying confined groundwater show relatively fresh water with TDS of 0.3–1.0 g $L^{-1}$ (Wu 2007). The water table depth is as large as 180 m and a level difference of 130 m hydraulic drop is observed due to the thrust fault in the south margin in SHZ (Fig. 1c).

## 3 Materials and methods

### 3.1 Water sampling

Groundwater (pumped from well, of which 3 are from spring and 3 are from the artesian well) were separated into three clusters along the Manas River motion (Table 1 and Fig. 2) including the upstream groundwater (UG, south of the Wuyi





Road), midstream groundwater (MG, area between the Wuyi Road and the West main canal–Yisiqi), and downstream groundwater (north of the West main canal–Yisiqi). Surface water samples (river water, ditch and reservoir water) and groundwater samples data from G30 to G39 were reported by Ji (2016) and Ma et al. (2018). Groundwater were sampled from wells for irrigation and domestic supply, in which shallow wells were pumped for a minimum 5 min before sampling

and deep wells were active for irrigation for more than 10 days prior to the sampling.

Water temperature (T), pH values, electrical conductivity (EC) and dissolved oxygen (DO) were measured (Table 1) in the field using calibrated Hach (HQ40d) conductivity and pH meters, which had been calibrated before use. Bicarbonate was determined by titration with 0.05 N HCl on site. Samples to be analyzed for chemical and stable isotopic values were filtered on site through 0.45 μm millipore syringe filters and stored in pre–cleaned polypropylene bottles at 4 ℃ until analysis. For

cation and strontium isotope analysis, the samples were acidified to pH < 2 with ultrapure $HNO_3$.

Extreme precautions are needed to be taken to avoid contamination from equipment such as pumps and tubing (Cook et al., 2017; Darling et al., 2012; Han et al., 2012) for CFCs samples. After purging the wells, water samples were collected directly from the borehole using a copper tube sampling pipe for CFC analysis. One end of the pipe was connected to the well casing, and the other end was placed in the bottom of a 120 mL borosilicate glass bottle, inside a 2000 mL beaker. The

well water was allowed to flow through the tubing for ten minutes, thoroughly flushing the tubing with well water. The bottle was submerged, filled and capped underwater when there was no bubbles appeared in the bottle, following the protocols described by Han et al. (2007). In this study, 5 bottles were collected at each well and 3 of which were analyzed. A total of 10 wells were collected for CFC (CFC–11, CFC–12 and CFC–113) analysis. Unfiltered samples for $^3$H analysis were collected and stored in 500 mL airtight polypropylene bottles. Dissolved inorganic carbon (DIC) for $^{14}$C activity analysis was

precipitated in field from 180 to 240 L water samples to $BaCO_3$ by addition of excess $BaCl_2$ previously brought to pH ≥ 12 by addition of NaOH, which then were sealed in 500 mL polypropylene bottles, following the procedure reported by Chen et al. (2003).

### 3.2 Analytical techniques

The CFC concentrations were analysed within 1 month of sample collection at the Groundwater Dating Laboratory of the

Institute of Geology and Geophysics, Chinese Academy of Sciences (IGG–CAS) using a purge–and trap gas chromatography procedure with an Electron Capture Detector (ECD), which has been reported by Han et al. (2012, 2015) and Qin et al. (2011). The procedures were followed by Oster et al. (1996). The detection limit for each CFC is about 0.01 pmol $L^{-1}$ of water, with the error less than ±5 %. The obtained results are shown in Table 1.

The $^3$H and $^{14}$C activities of groundwater were measured using liquid scintillation spectrometry (1220 Quantulus ultra–

low–level counters, PerkinElmer, Waltham, MA, USA) at the State Key Laboratory of Biogeology and Environmental Geology, China University of Geosciences in Wuhan. Water samples for $^3$H were distilled and electrolytically enriched prior to being analysed. Detailed procedures were followed by Morgenstern and Taylor (2009). $^3$H activities were expressed as tritium unit (TU), with 1 TU corresponding to a $^3$H/$^1$H ratio of $1 \times 10^{-18}$. For $^{14}$C samples, the obtained $BaCO_3$ samples were




first converted to $CO_2$, then to acetylene ($C_2H_2$) which in turn was trimerized catalytically to $C_6H_6$ as described by Polach

(1987), prior to being analysed. $^{14}$C activities were reported as percent modern carbon (pMC). The achieved precision for $^3$H
and $^{14}$C were ±0.2 TU and ±0.4 pMC respectively.

The cation, anion and stable isotope measurements were performed at the State Key Laboratory of Biogeology and
Environmental Geology, China University of Geosciences in Wuhan. Cations were analysed using an inductively coupled
plasma atomic emission spectrometry (ICP–AES) (IRIS Intrepid II XSP, Thermo Elemental). Anions were analysed on

filtered unacidified samples using ion chromatography (IC) (Metrohm 761 Compact IC). Analytical errors were inferred
from the mass balance between cations and anions (with $HCO_3$), and are within ±6 %. Stable isotopic values ($\delta^2$H and $\delta^{18}$O)
analyses were measured using a Finnigan MAT–253 mass spectrometer (Thermo Fisher, USA, manufactured in Bremen,
Germany), with the TC/EA method. The $\delta^2$H and $\delta^{18}$O values (Table 1) were presented in δ notation in ‰ with respect to the
Vienna Standard Mean Ocean Water (VSMOW), with an analytical precision of 0.5 ‰ vs. VSMOW for $\delta^2$H and of 0.1 ‰

for $\delta^{18}$O.

## 3.3 Apparent CFC age estimation

Knowledge of the history of the local atmospheric CFC concentrations is first required for groundwater dating. The
difference between the local and global background atmospheric CFC concentrations (Northern Hemisphere) refers to as
CFC excess which varies largely based on the industrial development. Elevated CFC concentrations of 10–15 % higher than

those of the Northern Hemisphere have been reported in the air of urban environments such as Las Vegas, Tucson, Vienna
and Beijing (Barletta et al., 2006; Carlson et al., 2011; Han et al., 2007; Qin et al., 2007), while in Lanzhou and Yinchuan
(northwest China) were about 10 % less (Barletta et al., 2006). In this study, MRB locates in the northwest of China (Fig. 1a)
with very low population density, far from the industrial city, and there is no clear difference in the atmospheric CFC
concentrations between North America and low latitude countries. To evaluate CFC ages, the time series trend of Northern

Hemisphere atmospheric mixing ratio (1940–2014, http://water.usgs.gov/lab/software/air/cure/) was adopted in this study.

Measured CFC concentrations (in pmol L$^{-1}$) can be interpreted in terms of partial pressures of CFCs (in pptv) in solubility
equilibrium with the water sample based on Henry's Law solubility. Concrete computational process was followed that by
Plummer et al. (2006a). In the arid northwest China, the local shallow groundwater temperature was more suitable than the
annual mean surface air temperature to be estimated for the recharge temperature (Qin et al., 2011) as the local low

precipitation usually cannot reach the groundwater. Previous studies in MRB (Ji, 2016; Wu, 2007) have also indicated that
much less vertical recharge water from the local precipitation as compared to the abundant groundwater lateral flow recharge
and river leakage from the mountain to the piedmont areas. In this study the measured groundwater temperature that vary
from 11.5 to 15.7 °C from each well (Table 1) as the recharge temperature was used to estimate the water ages. Surface
elevations of the recharge area vary from 316 to 755 m. The apparent age is then determined by comparing the calculated

partial pressures of CFCs in solubility equilibrium with the water samples with historical CFC concentrations in the air based
on the hypothesis of piston flow.



### 3.4 Mean transit times estimated by $^3$H and CFCs

The widely used lumped parameter models (LPMs) that exist is based on the assumptions that the hydrologic system is considered as a closed system, sufficiently homogeneous, being at steady–state, having a defined input and a corresponding output in the form of pumping wells, springs or streams draining the system (Małoszewski and Zuber, 1982). For steady–state hydrologic system, on accounting of the $^3$H and CFCs tracers enter groundwater with precipitation are injected proportionally to the volumetric flow rates by nature itself, the output concentration in water at the time of sampling relating to the input $^3$H and CFCs can be described by the convolution integral (Małoszewski and Zuber, 1982):

$$C_{\text{out}}(t) = \int_0^\infty C_{\text{in}}(t-\tau) g(\tau) e^{-\lambda_{^3\text{H}}\tau} \, d\tau \qquad \text{for} \quad ^3\text{H} \quad \text{tracer} \tag{1a}$$

$$C_{\text{out}}(t) = \int_0^\infty C_{\text{in}}(t-\tau) g(\tau) d\tau \qquad \text{for} \quad \text{CFCs} \quad \text{tracer}, \tag{1b}$$

where $C_{\text{out}}$ is the tracer output concentration, $C_{\text{in}}$ is the tracer input concentration, $\tau$ is the transit time, $t-\tau$ is the time when water entered the catchment, $\lambda_{^3\text{H}}$ is the $^3$H decay constant ($\lambda_{^3\text{H}} = \ln 2/12.32$), and $g(\tau)$ is the response function that describes the transit time distributions in the hydrologic system.

Several TTDs of various models have been described (Małoszewski and Zuber, 1982; Jurgens et al., 2012) and been widely used in studies of variable timescales and catchment areas (Cartwright and Morgenstern, 2015, 2016; Cartwright et al., 2018; Hrachowitz et al., 2009; Morgenstern et al., 2010, 2015; McGuire et al., 2005), of which the selection of each model depends on the hydrogeological situations in the hydrologic system to which it is applicable. In this study, the exponential–piston flow model (EPM), the dispersion model (DM), and the exponential mixing model (EMM) were given below as transit time distribution function:

Exponential–piston flow model

$$g(\tau) = 0 \qquad \text{for} \quad \tau < \tau_{\text{m}}(1-1/\eta) \tag{2a}$$

$$g(\tau) = \frac{\eta}{\tau_{\text{m}}} e^{(-\eta\tau/\tau_{\text{m}}+\eta-1)} \qquad \text{for} \quad \tau \geq \tau_{\text{m}}(1-1/\eta) \tag{2b}$$

Dispersion model

$$g(\tau) = \frac{1}{\tau\sqrt{4\pi D_{\text{P}}\tau/\tau_{\text{m}}}} e^{-\left(\frac{(1-\tau/\tau_{\text{m}})^2}{4\pi D_{\text{P}}\tau/\tau_{\text{m}}}\right)} \tag{3}$$

Exponential mixing Model

$$g(\tau) = \frac{1}{\tau_{\text{m}}} e^{(-\tau/\tau_{\text{m}})}, \tag{4}$$



where $\tau_m$ is the mean transit time, $\eta$ is the ratio defined as $\eta = (l_P + l_E)/l_E = l_P/l_E + 1$, where $l_E$ (or $l_P$) is the length of area at the water table (or not) receiving recharge, $D_P$ is the dispersion parameter defined as $D_P = D/(vx)$, where $D$ is the dispersion coefficient ($m^2\,day^{-1}$), $v$ is velocity ($m\,day^{-1}$), and $x$ is distance (m).

## 3.5 The $^{14}$C age model

The calculation of $^{14}$C ages may be complicated if groundwater dissolved inorganic carbon (DIC) is derived from a mixture of sources or/and the $^{14}$C originating from the atmosphere or soil zone is often significantly diluted by the dissolution of $^{14}$C–free carbonate minerals in the aquifer matrix and biochemical reactions along the groundwater flow paths (Clark and Fritz, 1997). While only minor carbonate dissolution is likely, determination of groundwater transit times requires $^{14}$C correction to be taken into account (Atkinson et al., 2014). When dissolution of carbonate during recharge or along the groundwater flow path may dilute the initial soil $CO_2$, $\delta^{13}$C can be used to trace the process (Clark and Fritz, 1997). An equation for the reaction between carbon–dioxide–containing water with a carbonate mineral is commonly written as (modified after Pearson and Hanshaw, 1970):

$$CO_2 + H_2O + CaCO_3(\delta^{13}C_{carb} = 0) \rightarrow Ca^{2+} + 2HCO_3^-(\delta^{13}C_{DIC}), \tag{R1}$$

where $\delta^{13}C_{carb}$ is the dissolved carbonate $\delta^{13}$C value (approximately 0; Clark and Fritz, 1997), and $\delta^{13}C_{DIC}$ is the measured $\delta^{13}$C value in groundwater.

If it assumed that any dissolution that occurs under truly open system conditions will equilibrate both $^{14}$C and $^{13}$C in the recharging water and that closed system dissolution of calcite in the aquifers is the major process (Atkinson et al., 2014), the initial $^{14}$C activity ($a_0\,^{14}C$) may be calculated from the following equation:

$$a_0\,^{14}C = a_{rech}\,^{14}C \frac{\delta^{13}C_{DIC} - \delta^{13}C_{carb}}{\delta^{13}C_{rech} - \delta^{13}C_{carb}}, \tag{5}$$

where $a_{rech}\,^{14}C$ is the recharging water $^{14}$C activity (assumed to be 100 pMC), and $\delta^{13}C_{rech}$ is the recharging water $\delta^{13}$C value in the recharge zone depending on vegetation type (C3 or/and C4).

Depending on knowing the measured $^{14}$C activity after adjustment for the geochemical and physical dilution processes in the aquifer (without radioactive decay), then the groundwater $^{14}$C ages (*t*) can be calculated from the following decay equation:

$$t = -\frac{1}{\lambda_{^{14}C}} \times \ln \frac{a\,^{14}C}{a_0\,^{14}C}, \tag{6}$$

where $\lambda_{^{14}C}$ is the $^{14}$C decay constant ($\lambda_{^{14}C} = \ln 2/5730$), and $a\,^{14}C$ is the measured $^{14}$C activity of the DIC in groundwater.

Previous studies in the arid northwest China (Edmunds et al., 2006; Huang et al., 2017) have concluded that a volumetric value of 20 % "dead" carbon derived from the aquifer matrix was recognized, which is consistent with the value (10–25 %) obtained by Vogel (1970). Once below the $^3$H detection limit, the initial $^{14}$C activity can be considered as the initial value by





plotting the $^3$H value versus the $^{14}$C activity, and based on that, the maximum $^{14}$C activity of 79.8 pMC with $^3$H value below the detection limit in the Baiyang River Basin, 260 km northwest to our study area, was obtained (Huang et al., 2017). Therefore, the initial $^{14}$C activity ($a_0\ ^{14}$C) of 80 pMC is used to correct groundwater $^{14}$C ages (results are shown in Table 1), despite this simple correction makes no attempt to correct the age of individual samples that may have experienced different
water–rock interaction histories.

## 4 Results and discussion

### 4.1 Stable isotope and major ion hydrochemistry

The $\delta^2$H and $\delta^{18}$O values in the study area vary from –75.88 to –53.40 ‰ and –11.62 to –6.76 ‰ for the surface water, and from –82.45 to –62.16 ‰ and –12.19 to –9.01 ‰ for the groundwater. Figure 3a shows the $\delta^2$H and $\delta^{18}$O values of surface
water and groundwater in relation to the precipitation isotopes of the closest GNIP station (Urumqi station in Fig. 1a). Both the linear slope (7.3) and intercept (3.1) of the Local Meteoric Water Line (LMWL) are lower than that of the Global Meteoric Water Line (GMWL, 8 and 10, respectively; Craig 1961). Surface water (ditch, river and reservoir water) are more enriched in heavy isotopes and defined an evaporation line with a slope of 4.5 (Fig. 3b), which is much higher than that solely calculated from the upstream river water and reservoir water (slop=3.2 from Ma et al., 2018).

Groundwater deuterium excess values ($d-\text{excess} = \delta^2\text{H} - 8\delta^{18}\text{O}$, Fig. 3b) defined by Dansgaard (1964) lie close to the annual mean LMWL ($d_{\text{LMWL}}$=13 ‰), which also suggest little isotope fractionation by evaporation as $d$–excess value decreases when water evaporates (Han et al. 2011; Ma et al., 2015). The $d$–excess values of surface water decrease from 17.12 ‰ in the upstream area to 0.68 ‰ in the downstream area, indicating strong evaporation effect, which is also demonstrated by the low slop (evaporation slop=4.5) of the surface waters.

The hydrochemistry compositions of surface water and groundwater in the MRB reflect evolution from fresh HCO$_3$–SO$_4$–Ca water type to HCO$_3$–SO$_4$–Na–Ca type and further to HCO$_3$–SO$_4$–Na type, and finally to brine Cl–SO$_4$–Na water type along the groundwater flow paths (Fig. 4). Groundwater in the unconfined aquifers (e.g., intermountain depression and piedmont plain aquifers in Fig. 1c) is dominated by Ca$^{2+}$ and HCO$_3^-$ with relatively low concentration of Na$^+$ (Fig. 4). Groundwater in the confined aquifers is characterized by a wide range with progressively increasing of Na$^+$ and Cl$^-$ ions,
meanwhile Ca$^{2+}$ and Mg$^{2+}$ ions decrease progressively towards the more concentrated end of salinity spectrum (Fig. 4). The concentration of SO$_4^{2-}$ ion gradually increases in the unconfined aquifers and becomes less dominant in the confined aquifers along the groundwater flow paths (Fig. 4).





### 4.2 Groundwater ages

Groundwater ages are shown in Table 1 for $^{14}C_{corr}$ and in Table 2 for CFCs. Figure 5 shows the estimated CFC atmospheric
partial pressures. $^{3}H$ activities of groundwater and the reconstructed precipitation time series results are shown in Fig. 6.
Figure 7 shows the distributions of tracer concentrations (Fig. 7a) and ages (Fig. 7b) with distance to mountain.

### 4.2.1 Apparent CFC ages

As can be seen in Table 1, groundwater with well depths between 13 and 150 m contain detectable CFC concentrations
(0.17–3.77 pmol L$^{-1}$ for CFC–11, 0.19–2.18 pmol L$^{-1}$ for CFC–12, and 0.02–0.38 pmol L$^{-1}$ for CFC–113) both in the
upstream and midstream areas, indicating at least a small fraction of young groundwater components (post–1940). The
highest concentration was observed in the UG (G3), south of the fault, median and the lowest were respectively observed in
the west and east bank of the 'East main canal' in the MG, north of the fault. In the midstream area (Fig. 2), CFC
concentrations generally decrease with well depth at the south of reservoirs (G25, G8, and G9), while increase with well
depth at the north of reservoirs (G15 and G16), which might indicate the different groundwater flow paths (e.g., downward
or upward flow directions). The estimated CFC atmospheric partial pressures and apparent ages are shown in Table 2 and
Fig. 5, with the apparent ages varying from 25 to 54.5 yrs. The UG (G3) has the same CFC–113 and CFC–12 apparent ages
(Table 2), indicating piston flow recharge in the upstream area. The MG CFC–11–based ages agreed within 2–4 yrs with that
based on CFC–12 concentrations, while that the CFC–113–based ages were much 2–9 yrs younger than that based on CFC–
11 and CFC–12 concentrations, indicating mixtures of young and old groundwater components in the midstream area.

Despite apparent CFC ages presuppose that the measured water is the result of simple piston flow with no mixing, they
provide a good first approximation for groundwater age (Darling et al., 2012; Han et al., 2015). The time lag for CFCs
transport through the thick unsaturated zone (Cook and Solomon, 1995), as well as degradation especially for CFC–11 is
being common in the anaerobic groundwater (Horneman et al., 2008; Plummer et al., 2006b), which both are important
consideration when dating groundwater using CFC concentrations. In this study, the groundwater aerobic environment
(Table 1, DO values vary from 0.7 to 9.8 mg L$^{-1}$) make CFC degradation under anoxic conditions unlikely. The groundwater
were mainly recharged by the river fast leakage in the upstream area and piedmont plain (Ma et al., 2018), where the soil
texture is consisted of pebbles and sandy gravel (Fig. 1c), which confirm one to assume that the unsaturated zone air CFC
closely follows that of the atmosphere and thus the recharge time lag through the unsaturated zone is not consideration.
Nevertheless, CFC–11 has shown a greater propensity for degradation and/or contamination than CFC–12 (Plummer et al.,
2006b). Therefore, one assign the CFC–12 apparent age in the following discussions.

The youngest groundwater apparent age is in the upstream area (G3 with 25 yrs), which is most likely due to the shortest
flow paths from recharge sources compared to the piedmont groundwater samples in the midstream area. The distribution of
CFC–12 apparent ages with distance to mountain (Fig. 7b) reveals two different trends from the upstream to midstream areas.
First, groundwater ages increased with elevated distance along the Manas River motion from the upstream to the midstream



areas, suggesting longer flow paths towards the north orientation in the West 'East main canal'. Second, the oldest ages of groundwater (G5 and G7) were located in the East 'East main canal' in the midstream area with much shorter distance than that in the reservoir north (G15 and G16), this could be explained by the lower groundwater velocities in the East 'East main canal', where the hydraulic gradient (Fig. 2) is much smaller than the West. Furthermore, it can be seen from Table 2 and Fig. 2 that groundwater ages increased from 29 to 38 yrs with well depth increasing from 48 to 100 m at the reservoir south (G25,

G8 and G9), while that in the reservoir north decreased from 40.4 to 29.6 yrs with well depth increasing from 23 to 56 m (G15 and G16). The different trends for the relationship between groundwater age and well depth might be ascribed to the different flow paths among the two sites (e.g., reservoir south and north).

### 4.2.2 $^3$H and $^{14}$C ages

$^3$H activity of groundwater samples vary from 60 to 1.1 TU (Fig. 6 and Table 1), with the highest value in the UG (G4) and

the relatively low values in the MG (mean 12.4 TU) and DG (mean 4.5 TU). The historical precipitation $^3$H activity in Urumqi station (Fig. 6) was reconstructed from the available data in the International Atomic Energy Agency (IAEA) following the method described by Han et al. (2015). The estimated results show that precipitation $^3$H activity has been decreasing since the 1960s and the estimated current tritium level has since stabilized around 31.5 TU (Fig. 6). All of the measured UG (G1, G2, and G4) and G23 (belong to MG) $^3$H data were higher than 34.3 TU, which indicates some fractions

of the 1960s precipitation recharge. Groundwater with $^3$H activity lower than 5.6 TU contains some pre–1950s recharge.

The distribution of $^3$H activity with distance to mountain (Fig. 7a) also reveals two different trends from the upstream to midstream areas. First, $^3$H activities of groundwater in the upstream area increase from 41.1 (G1 and G2) to 60 TU (G4) with distance, which probably indicate that more fractions of the 1960s precipitation recharge was occurred for G4 than G1 and G2 groundwater samples. It is seen from Fig. 2 that nearing G4 samples shows the highest hydraulic gradient values, which

imply that more fractions of the 1960s precipitation recharge was possible. Second, $^3$H activities of groundwater in the midstream area showed an obvious reduction trends along the Manas River motion from 37.5 (G23) to 1.1 TU (G14) with distance, indicating that more fractions of pre–bomb precipitation recharge may have occurred along the groundwater flow direction in the north of the fault. Furthermore, the $^{14}$C activity in the MG showed small increases with distance (Fig. 7a) from 43.4 to 54.6 pMC, with the exception of sample G12 at approximately 54 km of 86.9 pMC that has an $^{14}$C$_{corr}$ age of –

684 yrs (modern recharge), while in the DG decreased to 23.5 pMC. The presence of detectable $^3$H (2.9–6.91 TU) in DG with low $^{14}$C values (23.5–34.3 pMC) indicated that some mixing with post–bomb precipitation recharge may be occurred.

The distribution of groundwater both $^3$H ages (estimated by comparing the $^3$H activity in precipitation decayed to 2014 and in groundwater (Fig. 6) on condition that a piston flow recharge was across the study area) and $^{14}$C$_{corr}$ ages (calculated from Eq. (6)) with distance to mountain (Fig. 7b) were heterogeneous with no relationship along lateral groundwater flow

paths. The great age differences between $^3$H (28.8–60 yrs) and $^{14}$C$_{corr}$ ages (3158–10 127 yrs) from midstream to downstream areas (Fig. 7b) imply that mixing with variable proportions of young and old water have occurred. Although there is an overall $^{14}$C$_{corr}$ age increase from the midstream to downstream areas (Fig. 7b), inversely a decrease in age with distance in





the MG is observed. This probably be ascribed to the largely vertical mixing processes (Ma et al., 2018) within the confined aquifers in the reservoir north in the midstream area, because an increase in groundwater age with distance is usually

observed within the confined aquifers but not in the unconfined aquifers (Batlle–Aguilar et al., 2017). Another speculative possibility that aquifers in the midstream area have received significant volume of its recharge from the upstream unconfined groundwater by lateral flow with different flow paths, which is still not yet confirmed.

## 4.3 Modern and paleo–meteoric recharge features

Stable isotopes ($\delta^2$H and $\delta^{18}$O), components of the water molecule that record the atmospheric conditions at the time of

recharge (Batlle–Aguilar et al., 2017; Chen et al., 2003), provide valuable information on groundwater recharge processes. Generally, there are two possible meteoric recharge sources including precipitation in the modern climate and in the paleoclimate. Groundwater whose isotopic values are more depleted than the modern precipitation usually would be ascribed to two recharge sources including snowmelt/precipitation at higher elevation and precipitation fallen during cooler climate. Figure 3 shows that groundwater generally lie along the LMWL but do not define evaporation trend, implying little

evaporation and isotope exchange between groundwater and the rock matrix have occurred (Ma et al., 2018; Négrel et al., 2016). Transpiration over evaporation is likely to be dominant in the soil when infiltration as soil water uptake by root is not significantly isotope fractionated (Dawson and Ehleringer 1991).

Three groundwater clusters can be identified in the $\delta^2$H–$\delta^{18}$O plot (Fig. 3b), suggesting the different recharge sources among the upstream, midstream and downstream areas. The first group with $\delta^2$H and $\delta^{18}$O average values of –68.24 and –

10.08 ‰ is from UG, and is located much closer to the summer rainfall (Fig. 3a), reflecting more enriched summer rainfall inputs. Negligible evaporation trend was observed in the UG though the recharge is mostly in the summer due to the fast river leakage in the intermountain depression through highly permeable pebbles and gravel deposits (Fig. 1c). Furthermore, the detectable CFC concentrations and high $^3$H activities (Table 1) also indicate the modern precipitation recharge. An overlap between surface water and UG indicates the same recharge sources, as some alignment of river water and

groundwater isotopic values may indicate a qualitative recharge under climate conditions similar to contemporary conditions (Huang et al., 2017).

The second group with average $\delta^2$H and $\delta^{18}$O values of –73.10 and –11.0 ‰ overlapping with the annual amount–weighted mean rainfall isotopic value is from MG. Such isotopic values are comparable to the modern annual amount–weighted mean rainfall $\delta^2$H and $\delta^{18}$O values (–74.7 and –11.0 ‰, Fig. 3a), probably reflecting year–round modern

precipitation recharge. However, there could be another explanation for the relatively much scattered MG isotopic values in the $\delta^2$H–$\delta^{18}$O plot (Fig. 3b), which is mixing with different time–scales recharge of variable isotopic values at different aquifers and sites along the groundwater flow paths. A large both $^3$H and $^{14}$C activities (Fig. 7a) and groundwater age variabilities (Fig. 7b) distributions with distance to mountain along groundwater flow paths in the midstream area was observed, indicating that mixture of short to long timescales recharge was possible. Groundwater isotopes in the piedmont

plain are totally relatively enriched in heavy isotopes (Fig. 3b), which overlapping with the river water, indicating the fast



river leakage recharge through short time (Ma et al., 2018). Groundwater isotopes in the oasis plain diverge from that in the piedmont plain (Fig. 3b), as well as do not show alignment with surface water, indicating the recharge with longer flow paths rather than the fast river leakage recharge.

The third group with the most depleted in heavy isotopes (–82.36 and –12.03 ‰) is from DG, and is located much closer to the winter rainfall in the $\delta^2$H–$\delta^{18}$O plot (Fig. 3b). Previous studies (Ji, 2016; Ma et al., 2018) have shown that vertical recharge from the winter rainfall in the downstream area is mostly unlikely. As the altitude effects of precipitation recharge (Clark and Fritz, 1997) and paleo–meteoric recharge during cooler climate (Chen et al., 2003) could collectively account for the isotopically depleted groundwater, it is usually not easy to distinguish the precipitation recharge sources at higher elevation from paleo–meteoric recharge. However, the positive altitude gradient of isotopes in precipitation (Kong and Pang,

2016) over North Tianshan Mountain (Fig. 1a) ascribed to the moisture recycling and sub–cloud evaporation effects would give rise to more enriched isotopes from higher altitude precipitation recharge. The isotopically enriched UG (Fig. 3b) in the intermountain depression with higher altitude (Fig. 1c) are recharged from the high mountain, which likewise demonstrates that DG being from the high mountain recharge is mostly unlikely. Therefore, the depleted isotopic values from DG (Fig. 3b) should be ascribed to the paleo–meteoric recharge in a cooler climate. Temperature depressions of ~10 ℃ in Xinjiang region

(Li et al., 2015) and 6–9 ℃ in North China Plain (Chen et al., 2003) in the last glacial period cooler than the modern have been observed. Groundwater that has depleted $\delta^{18}$O value of around –12.0 ‰ from the paleo–meteoric recharge have been widely acknowledged in the arid northwest China, like in Minqin basin (Edmunds et al., 2006), and both in East (Li et al., 2015) and West (Huang et al., 2017) Junggar Basin (Fig. 1a).

### 4.4 Groundwater mean transit times

#### 4.4.1 $^3$H and CFCs

In this study the input CFCs concentrations are from the time series trend of Northern Hemisphere atmospheric mixing ratio (Fig. 5). The time series $^3$H activities as the input data (Fig. 6) are still necessary that is based on the following two considerations. First, the study area is located in the Northern Hemisphere, where the bomb–test $^3$H activities were several orders of magnitude higher than in the Southern Hemisphere (Clark and Fritz, 1997; Tadros et al., 2014). $^3$H activity in the

atmosphere was superimposed over the China atmospheric nuclear tests from 1964 to 1974 in the arid northwest China, and thus the remnant $^3$H activities are still affected by the tail–end of the bomb pulse. Second, the study area is more than 3500 km far away from the western pacific, where $^3$H activity in the atmosphere is evidently much higher than coastal sites due to the continental effect (Tadros et al., 2014). Furthermore, though $^3$H activity in the atmosphere is known to vary between seasons (Cartwright and Morgenstern, 2016; Morgenstern et al., 2010; Tadros et al., 2014), the year–round mean values (Fig.

6) were adopted in this study.

    Each transit time distribution function (Eqs. (2) to (4)) has its suited hydrogeological situations in the hydrologic system to which it is applicable (Małoszewski and Zuber, 1982), which the EPM is particularly useful for interpretation of MTTs in



aquifers that have regions of both exponential and piston flow (Cartwright et al., 2017). The unconfined aquifers that adjacent to the rivers (Fig. 1 c) are likely to exhibit exponential flow, recharge through the unsaturated zone (Fig. 1c) will

most likely resemble piston flow (Cartwright and Morgenstern, 2015; Cook and Böhlke, 2000). For the time series $^3$H and CFCs inputs, MTTs (Fig. 8) were initially calculated using the EPM with an EPM ratio of 1.5 ($l_E$ in Eq. (2) is by adding the intermountain depression to the piedmont plain in Fig. 1c) via Eqs. (1) and (2). River leakage and rainfall input were possible from the piedmont plain (Ma et al., 2018), thus a less proportion of piston flow by the EPM with an EPM ratio of 2.2 ($l_E$ in Eq. (2) is only in the piedmont plain in Fig. 1c) was also used. To test the veracity the DM with $D_P$ of 0.03 and

0.1 and the EMM were also used to calculate the MTTs via Eqs. (1), (3) and (4). Plots of the output concentrations for $^3$H (Fig. 8a) and CFCs (CFC–11 in Fig. 8b, CFC–12 in Fig. 8c and CFC–113 in Fig. 8d) vs. MTTs for different lumped parameter models show wide MTTs ranges that increase with the increasing MTTs.

Figure 9 shows that different LPMs yield different MTTs for the same time serious $^3$H activities and CFC concentrations, which MTTs obtained from different LPMs tend toward more discrete with their increase. For the CFCs rainfall inputs (Fig.

9a, b), MTTs from the EPM with an EPM ratio of 1.5 (Fig. 9a) vary from 19 to 101 yrs with a median of 51 yrs for the CFC–12 rainfall input, from 33 to 115 yrs with a median of 62.3 yrs for the CFC–11 rainfall input, and from 18 to 92 yrs with a median of 50.2 yrs for the CFC–113 rainfall input. Good linear relationships for the MTTs between the different CFCs rainfall inputs were obtained using the same EPM (EPM (1.5) in Fig. 9a and EPM (2.2) in Fig. 9b). MTTs increase with the decreasing EPM ratios (from 2.2 to 1.5; Fig. 9b), implying the longer flow paths that recharge from the intermountain

depression. For the range of CFC–12 concentrations in the UG and in the west bank of the 'East main canal' of MG, similar MTTs were estimated from the different lumped parameter models (Fig. 9b) with mean values varying from 28.6 to 64.8 yrs, while those in the east bank of the 'East main canal' (Fig. 9b) of MG show larger differences with mean values varying from 129.2 to 173 yrs. Totally, the youngest value was observed for the G3 sample (south of the fault) while the oldest was for G5 sample (east bank of the 'East main canal'; Fig. 2).

By contrast with the CFCs rainfall inputs, MTTs estimated using the $^3$H rainfall input by different LPMs (Fig. 9c) show larger uncertainties and wider ranges. For the EPM with an EPM ratio of 1.5, MTTs vary from 19 to 158 yrs with a median of 112.2 yrs (Fig. 9c), which are much longer than those calculated from the CFCs rainfall inputs by the same model (Fig. 9b). The differences could be due to the longer travel times through the thick unsaturated zone for $^3$H than CFCs. $^3$H moves principally in the liquid phase while CFCs travel in the gas phase through the unsaturated zone (Cook and Solomon, 1995).

The more rapid transport for gas–phase than liquid–phase in the unsaturated zone would be expected to give rise to longer transit times from $^3$H than those determined from CFCs (Cook, et al., 2017). Furthermore, the ranges in MTTs estimated from the EPM with EPM ratios of 1.5 and 2.2, the DM with $D_P$ of 0.03 and 0.1, and the EMM are 16–158 yrs, 72–285 yrs, and 30–360 yrs, respectively. Similar MTTs trends with that calculated from the CFC–12 input were observed (Fig. 9b), which separate the west and east bank of the 'East main canal' of MG and DG from each other. Uncertainties increase with





the increasing MTTs among the different models, especially when MTTs > 130 yrs (Fig. 9c), which the samples are mainly collected from the east bank of the 'East main canal' of MG and DG.

The east bank of the 'East main canal' groundwater with relatively short distance to mountain (Fig. 7b) and with much smaller hydraulic gradient (Fig. 2) have much older MTTs (Fig. 9b, c) than those in the west bank. That in the west bank shows an overall increasing trend with the distance to mountain in MG and DG (Fig. 9b, c) as the longer and deeper flow

paths usually give rise to the longer MTTs (Cartwright and Morgenstern, 2015, 2016; McGuire et al., 2005). Despite the influence from the uncertainties of the input concentrations and different models (Cartwright and Morgenstern, 2015, 2016), MTTs have been identified to vary on account of more complex interplay of factors like mixing and dispersion in the flow systems, which may result in significantly different MTTs from results calculated by LPMs assuming a homogeneous aquifer with a simple geometry to the actual results (Cartwright et al., 2017; Kirchner, 2016; Stewart et al., 2017).

Nevertheless, in this study the homogeneous aquifers, being at steady–state, have been assumed to use the LPMs to calculate the MTTs.

### 4.4.2 Hydrochemistry

Strong correlations of hydrochemistry components with groundwater age permitting the hydrochemistry to be used as proxies or complementary for age via previously established relationships in close lithological conditions. An excellent

correlation between $SiO_2$ and MTTs with the correlation coefficient $R^2$ of 0.997 was reported (Morgenstern et al., 2010), which shows much higher $R^2$ than in Fig. 10 and in other results (Morgenstern et al., 2015). Silica ($SiO_2$, Fig. 10a), sulfate ($SO_4^{2-}$), bicarbonate ($HCO_3^-$) and total dissolved solid (TDS) (Fig. 10b) both show good correlations with groundwater age, indicating that mineral dissolutions by water–rock interactions dominate the hydrochemistry changes (Ma et al., 2018), during which major ion concentrations increase with groundwater age. However, MTTs estimated by $^3$H activities showed

poor correlations with the ions (not shown). Besides, the lithology type groundwater flow through within the aquifer and the likely evolutionary path ways play an important role in the hydrochemistry compositions. The negative saturation indices (SI) with respect to gypsum of all waters (Ma et al., 2018) indicate that the high $SO_4^{2-}$ concentrations (Fig. 10b) would be ascribed to the gypsum dissolution in the tertiary stratum. Also note that high $SO_4^{2-}$ can be originated from the geothermal water (Morgenstern et al., 2015), in contrast to studies such as Guo et al. (2014) and Guo et al. (2017), and can be biased due

to anoxic $SO_4^{2-}$ reduction. However, the groundwater with relatively low temperatures and aerobic environment (Table 1) make the two cases above unlikely.

The combination of hydrochemistry concentrations and groundwater age data is also a powerful tool for investigating the groundwater flow processes and flow through conditions (McGuire and McDonnell, 2006; Morgenstern et al., 2010, 2015), identifying the natural groundwater evolution and the impact of anthropogenic contaminants (Morgenstern et al., 2015;

Morgenstern and Daughney, 2012), and giving more accurate prediction of the contaminants like nitrate than either source of information. The pH of groundwater decrease from 10.1 to 8.6 over the age range from 19 to 101 yrs, with a log law fit of



$pH = 0.72 \times \ln(MTTs) + 11.85$, $R^2 = 0.65$ (Fig. 10a). On the contrary, a trend of increasing pH with increasing groundwater age has been reported in New Zealand (the dashed red line shown in Fig. 10a; Morgenstern et al., 2015), where the pH values were overall less than 7.2. These two discrepant trends can be explained by the relationship between the pH and

$HCO_3^-$ concentrations in water (inserted plot in Fig. 10a), which the pH increase with increasing $HCO_3^-$ concentrations only when the pH is less than 8.34, otherwise decrease with increasing $HCO_3^-$ concentrations. Therefore, the trend of increasing $HCO_3^-$ concentrations with increasing groundwater age (Fig. 10b) in this study indicates that decreasing trend for the pH (from 10.1 to 8.6) is reasonable.

    The soda waters with the overall pH higher than 8.1 (Table 1) are in disequilibrium with primary rock–forming minerals

of the host rocks. The incongruent dissolutions of the albite and anorthite through hydrolysis reaction are:

$$2NaAlSi_3O_8 + 11H_2O = Al_2Si_2O_5(OH)_4 + 2Na^+ + 4H_4SiO_4 + 2OH^- \qquad (R2)$$

$$CaAl_2Si_2O_8 + 3H_2O = Al_2Si_2O_5(OH)_4 + Ca^{2+} + 2OH^-, \qquad (R3)$$

where all the chemical components of the albite and anorthite release into the solution phase and produce $OH^-$ with simultaneous precipitation of kaolinite. A trend of increasing pH with increasing well depth (Table 1) suggests that

groundwater with pH < 9 was likely recharged by $CO_2$–containing water, because the $OH^-$ generally interacts with $CO_2$ and/or organic acids in the soil to form $HCO_3^-$ in the water (Wang et al., 2009). Likewise, the trend of decreasing pH with increasing MTTs (Fig. 10a) indicates higher $CO_2$–containing water with longer MTTs, which seems to suggest the anthropogenic input. The nitrate ($NO_3^-$) concentrations vary from 4.5 to 20.2 mg L$^{-1}$ with a median of 12.2 mg L$^{-1}$ (not shown), which exceed the natural nitrate concentration in groundwater of 5–7 mg L$^{-1}$ (Appelo and Postma, 2005). The

development of the plough after the 1950s, N–NO$_3$ fertilizer (with low $^{87}Sr/^{86}Sr$ ratios; Ma et al., 2018) and the extensive withdrawal groundwater for irrigation (Ji, 2016) suggest that irrigation infiltration could account for the groundwater high $NO_3^-$ concentrations in the piedmont plain. However, little irrigation infiltration was observed in the downstream area with groundwater $NO_3^-$ concentrations of < 5 mg L$^{-1}$ (Ma et al., 2018) due to the water–saving irrigation style, which had non–contributes to groundwater recharge in the arid northwest China.

**4.5 Groundwater mixing**

**4.5.1 CFC ratios**

Comparing CFC concentrations has provided a powerful tool to recognize samples containing co–existence of young (post–1940) water with old (CFC–free) water (Han et al., 2007; Han et al., 2012; Koh et al., 2012) or exhibiting contamination or degradation (Plummer et al., 2006b). The cross–plot of the concentrations for CFC–113 and CFC–12 (Fig. 11a) demonstrates

that all of the groundwater can be characterized as binary mixtures between young and older components, though there is still room for some ambiguity around the crossover in the late 1980s (Darling et al., 2012). As shown in Fig. 11a, all of the MG samples are located in the shaded region, representing no post–1988.5 waters recharge. The UG (G3) sample is clearly quite 'modern' and seems to be recharged in 1989.5 through piston flow or mixed by the old water and post–1988.5 water.





Using the method described by Plummer et al. (2006b) with the binary mixing model (BMM), the fractions of young water
vary from 12 to 91 % (Table 2) for the MG samples with the relatively low young fractions of 12 and 18 % in the east bank
of the 'East main canal' of MG samples (G5 and G7). These two well water table depths are more than 40 m, probably
indicating a relatively slow and deep circulated groundwater flow. This hypothesis is also suggested by lower DO (3.7–4.6
mg L$^{-1}$; Table 1) and nitrate concentrations (8.6–9.5 mg L$^{-1}$ from Ma et al., 2018) and relatively much smaller hydraulic
gradient (Fig. 2). Furthermore, as high as 100 % fraction of young water for G3 sample is obtained with the recharge water
from 1989.5, or 87 % fraction is obtained by the binary mixture between post–1988.5 water and old water (Table 2). The
quite 'modern' recharge for G3 sample is likewise explained by its highest DO (9.8 mg L$^{-1}$; Table 1) and relatively low
nitrate concentration (7.9 mg L$^{-1}$ from Ma et al., 2018), which represent the contribution of high–altitude recharge rather
than the old age water.

The apparent ages (Table 2 and Fig. 5) estimated from the PFM and MTTs (Fig. 9a) estimated from the EPM (1.5) using
CFC–113 concentrations are both generally lower than those obtained using CFC–11 and CFC–12, indicating that mixing
processes are prevailing and that the simple EPM (1.5) is also not an appropriate description of the groundwater flow
processes. All of the MG samples, located in the shaded region (Fig. 11a), can be regarded either as simple binary mixtures
of young (1980–1988.5) water and old (CFC–free) water, or be regarded as waters with exponential–piston and exponential
age distributions, which seems to represent no post–1988.5 waters recharge. It should be noted that the EMM line lies at the
boundary of the shaded region (Fig. 11a), suggesting that the binary mixing and EMM can be distinguished from each other
for the MG samples. While waters collected from the east bank (G5, G7 and G8) and one from the west bank (G15) of the
'East main canal' of MG seems cannot be distinguished from the binary mixing and EPM (1.5).

Points lying off the curves in the cross–plot CFC concentrations may indicate that contaminations from the urban air with
CFC compounds during sampling (Carlson et al., 2011; Cook et al., 2006; Mahlknecht et al., 2017) or degradation/sorption
of CFC–11 or CFC–113 (Plummer et al., 2006b) have occurred. Figure 11 demonstrates that the urban air with CFC
compounds contaminations, which generally cause elevated CFC concentrations than the global background atmospheric
CFC concentrations (Northern Hemisphere), are unlikely. Elevated CFC concentrations have been reported in the air of
urban environments such as Las Vegas, Tucson, Vienna and Beijing (Barletta et al., 2006; Carlson et al., 2011; Han et al.,
2007; Qin et al., 2007), contrary to that in the arid northwest China (Barletta et al., 2006). Hence, the anomalous CFC–
11/CFC–12 (Fig. 11b) and CFC–113/CFC–11 (Fig. 11c) ratios plotting off the model lines might be ascribed to the
following two hypotheses:

First, sorption in the unsaturated zone during recharge rather than the degradation of CFC–11 (Cook et al., 2006; Plummer
et al., 2006b) under anoxic conditions (Table 1, DO values vary from 0.7 to 9.8 mg L$^{-1}$). Nevertheless, the small deviations
(Fig. 11b, c) indicate that the hypothesized sorption rate was low. Higher CFC sorption rate with high clay fraction and high
organic matter in soils have been proved (Russell and Thompson, 1983), and vice versa (Carlson et al., 2011). Therefore, the
hypothesis of a low sorption rate due to the low clay fraction and low organic matter content in the intermountain depression
and the piedmont plain (Fig. 1c) seems reasonable.



Second, the time lag for CFCs movement both in dissolved and gas phases through deep unsaturated zone. The time lag

for the diffusive transport of CFCs through deep unsaturated zone in simple porous aquifers, a function of the tracer solubility in water, tracer diffusion coefficients and soil water content (Cook and Solomon, 1995), have been widely proved (Darling et al., 2012; Qin et al., 2011). The little or no separation in CFC–11 and CFC–12 apparent ages (Table 2) demonstrates that the time lag would be short in the faulted–hydraulic drop alluvium aquifers with deep unsaturated zone (Fig. 1c), meaning that groundwater CFCs ages obtained in this study effectively represent transit times since recharge reached the water table.

**4.5.2 CFC–12 vs. tritium data**

Combined use of CFCs and $^3$H may provide further help to resolve even more complicated mixing scenarios due to the large difference of the temporal pattern of the input functions between CFCs and $^3$H. Tracer–tracer concentration plots have some advantages over plots comparing apparent ages and tracer ratios because they reflect more directly the measured quantities and potential mixtures (Plummer et al., 2006b), such as mixing with irrigation water (Han et al., 2012, 2015; Koh et al., 2012)

or young water mixtures in different decades (Han et al., 2007; Qin et al., 2011). Plot of $^3$H vs. CFC–12 (Fig. 11d; CFC–11 and CFC–113 can substitute for CFC–12) shows that some samples (G9, G15 and G20) plot on or slight upward deviation to the piston flow line, while in Fig. 11a plot away from the piston flow line but on the binary mixing lines. G15 and G20 samples have the shallowest well depths of 23 and 13 m, respectively. G9 sample is collected from the piedmont plain with pebbles and sandy gravel deposits, nearing the Manas River (Fig. 2). There are two possible explanations for this situation: (i)

binary mixing between post–1988.5 water and older water recharged from 1950 to 1970, not containing CFC–free waters (pre–1940) (ii) mixtures from two end–members with one end–member has an age distribution close to the EPM (1.5) or EMM and the other end–member has post–1988.5 water. The second explanation requires that the samples contain at least some post–bomb fractions in the 1960s (revealed by $^3$H concentrations; Fig. 11d), and it also allows that the samples contain both post–1988.5 and pre–1940 waters, which however not revealed by CFC data (Fig. 11a). If the first explanation valid,

the binary mixing hypothesis and the young water (post–1940) fractions in Table 2 for these three samples should be adjusted accordingly.

Because atmospheric $^3$H concentrations have been elevated for a long time, old water components can be identified by anomalously low $^3$H activities in comparison with CFCs (Plummer et al., 2006b). G5 sample contain very low CFC–113 with $^3$H of 3.8 TU (Table 1), likely indicating that this sample was mixed by the older water (pre–1940) and 1960–1970

water. The low $^3$H concentration can be ascribed to the dilution by a high fraction of old water, and thus the '$^3$H bomb–peak' cannot be recognized. G16 sample, outside of the shaded region (Fig. 11d), has low $^3$H but significant CFC concentration. Two possible explanations have been obtained: (i) exposed to the atmosphere before sampling during large water table fluctuations due to groundwater pumping or add excess air to water through fractured system, (ii) river water or reservoir water with high CFC but minimal $^3$H recharge. Furthermore, the relatively high fractions of young water (89 %; Table 2)

preclude the dilution effect by the old water. Irrigation re–infiltration can cause a shift of the CFC concentrations to higher



values but not alter the $^3$H concentration (Han et al., 2015). However, the relatively low $NO_3^-$ concentrations (4.51 mg L$^{-1}$; data from Ma et al., 2018) of G16 sample suggest that irrigation re–infiltration is also not significant compared to the two explanations mentioned above. It seems that river water or reservoir water with very low $NO_3^-$ concentrations (2.7–7.3 mg L$^{-1}$; data from Ma et al., 2018) recharge is possible.

**5 Conclusions**

In this study, the environmental tracers and hydrochemistry have enabled us to identify the modern and paleo–meteoric recharge sources, to constrain the different end–members mixing rates, and to study the mixed groundwater mean transit times in faulted–hydraulic drop alluvium aquifer systems. The paleo–meteoric recharge in a cooler climate rather than the lateral flow from the higher elevation precipitation in the Manas River downstream area was distinguished. The thrust faults

were found to play a paramount role on groundwater flow paths and mean transit times due to their block water features, where the quite 'modern' groundwater with young (post–1940) water fractions of 87–100 % was obtained, indicating small mixing degree in the south of the fault. The short mean transit times (19 yrs) along with the higher $NO_3^-$ concentration (7.86 mg L$^{-1}$) than natural groundwater (5 mg L$^{-1}$) in the south of the fault (headwater area), implying the modern contaminants invading, which should arouse people's attention. Large amplitudes of mixing rate varying from 12 to 91 % were widespread

in the north of the fault due to the varying depth of long–screened boreholes or within the aquifer itself. Furthermore, the large water table fluctuations during groundwater pumping, vertical recharge through the thick unsaturated zone, and young water mixtures in different decades highlight the mixing diversity. The obtained strong correlations between groundwater mean transit times and hydrochemistry concentrations allow the first–order proxy at different times to be made. In addition, our study has also highlighted that mean transit times estimated by CFCs rather than $^3$H were more appropriate due to the

highly complex groundwater systems with thick unsaturated zone.

*Author contributions.* Xing Liang and Jing Li were responsible for the $^3$H and $^{14}$C analyses. Bin Ma undertook the sampling program and oversaw the analysis of the hydrochemistry and CFCs. Bin Ma and Menggui Jin prepared the manuscript.

*Competing interests.* The authors declare that they have no conflict of interest.

*Acknowledgements.* This research was financially supported by the National Natural Science Foundation of China (2015–
2018, no. U1403282).




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



**Table 1.** Chemical–physical parameters, stable isotopes, tritium ($^3$H), $^{14}$C, and CFC concentrations in groundwater samples in the Manas River Basin.

| Sample ID | Sampling date (d/m/y) | Elevation (m a.s.l.)[a] | Well depth (m) | pH | T (℃) | EC (μS cm$^{-1}$) | DO (mg L$^{-1}$) | $\delta^2$H (‰) | $\delta^{18}$O (‰) | CFC–11 (pmol L$^{-1}$) | CFC–12 (pmol L$^{-1}$) | CFC–113 (pmol L$^{-1}$) | $^3$H (TU) | $a^{14}$C (pMC) | $^{14}$C$_{corr}$ age (years) |
|---|---|---|---|---|---|---|---|---|---|---|---|---|---|---|---|
| *Upstream groundwater (UG)* | | | | | | | | | | | | | | | |
| G1 | 5/6/2015 | 1083 | 170[b] | | | | | −67.60 | −10.15 | | | | 41.07 | | |
| G2 | 5/6/2015 | 1107 | 170[b] | | | | | −67.40 | −10.17 | | | | 41.13 | | |
| G3 | 9/8/2015 | 755 | 150 | 10.1 | 11.5 | 387 | 9.8 | −70.39 | −10.50 | 3.14 | 2.18 | 0.38 | | | |
| G4 | 6/6/2015 | 532 | 58 | | | | | −66.80 | −9.91 | | | | 60.04 | | |
| *Midstream groundwater (MG)* | | | | | | | | | | | | | | | |
| G5 | 8/8/2015 | 467 | 100 | 8.6 | 13.4 | 896 | 4.6 | −69.35 | −10.73 | 0.17 | 0.19 | 0.02 | 3.80 | | |
| G6 | 8/6/2015 | 472 | 175 | | | | | | | | | | 28.90 | | |
| G7 | 7/8/2015 | 422 | 100 | 8.8 | 15.7 | 620 | 3.7 | −69.87 | −10.98 | 0.27 | 0.27 | 0.03 | | | |
| G8 | 7/8/2015 | 412 | 90 | 9.3 | 13.6 | 513 | 2.1 | −69.92 | −11.08 | 1.99 | 1.21 | 0.18 | 5.00 | | |
| G9 | 8/8/2015 | 484 | 100 | 9.1 | 14.5 | 612 | 9.1 | −74.58 | −11.01 | 1.31 | 1.03 | 0.13 | 7.10 | | |
| G10 | 8/6/2015 | 463 | 145 | | | | | −72.30 | −11.05 | | | | 9.09 | | |
| G11 | 8/6/2015 | 439 | 60 | | | | | −68.50 | −10.47 | | | | 15.75 | | |
| G12 | 7/8/2015 | 368 | 260 | 9.3 | 19.0 | 327 | 6.7 | −69.33 | −10.73 | | | | | 86.9 | −684 |
| G13 | 4/8/2015 | 370 | 300 | 9.4 | 17.1 | 307 | 1.2 | −76.20 | −11.22 | | | | | 54.6 | 3158 |
| G14 | 4/8/2015 | 370 | 60 | 9.0 | 13.2 | 556 | 1.4 | −68.96 | −10.43 | | | | 1.10 | | |
| G15 | 5/8/2015 | 364 | 23 | 8.1 | 12.7 | 1650 | 1.0 | −69.45 | −9.86 | 0.99 | 0.91 | 0.14 | 7.10 | | |
| G16 | 5/8/2015 | 357 | 56 | 9.0 | 15.2 | 291 | 0.7 | −76.59 | −11.57 | 2.69 | 1.54 | 0.22 | 4.80 | | |
| G17 | 5/8/2015 | 367 | 280 | 9.8 | 17.2 | 263 | 2.5 | −82.45 | −12.19 | | | | | 53.2 | 3373 |
| G18 | 6/8/2015 | 377 | 350[b] | 9.0 | 15.3 | 233 | 6.6 | −75.97 | −11.50 | | | | | 46.8 | 4432 |
| G19 | 6/8/2015 | 381 | 118[b] | 9.0 | 15.4 | 309 | 5.2 | −76.46 | −11.46 | | | | 6.90 | | |
| G20 | 6/8/2015 | 381 | 13 | 8.7 | 12.6 | 615 | 2.1 | −74.99 | −11.27 | 1.68 | 1.14 | 0.16 | 8.20 | | |





| Sample ID | Sampling date (d/m/y) | Elevation (m a.s.l.)[a] | Well depth (m) | pH | T (℃) | EC (µS cm$^{-1}$) | DO (mg L$^{-1}$) | $\delta^2$H (‰) | $\delta^{18}$O (‰) | CFC–11 (pmol L$^{-1}$) | CFC–12 (pmol L$^{-1}$) | CFC–113 (pmol L$^{-1}$) | $^3$H (TU) | $a^{14}$C (pMC) | $^{14}$C$_{corr}$ age (years) |
|---|---|---|---|---|---|---|---|---|---|---|---|---|---|---|---|
| G21 | 5/8/2015 | 424 | 180 | 8.8 | 15.6 | 378 | 8.0 | −77.30 | −11.60 | | | | | 43.4 | 5056 |
| G22 | 6/6/2015 | 428 | 150 | | | | | −69.72 | −10.41 | | | | 26.29 | | |
| G23 | 6/6/2015 | 446 | 70 | | | | | −67.63 | −9.92 | | | | 37.50 | | |
| G24 | 8/8/2015 | 453 | 110 | 9.1 | 14.7 | 571 | 8.6 | −77.35 | −11.23 | 1.53 | C[c] | C | | | |
| G25 | 8/8/2015 | 457 | 48 | 9.5 | 13.6 | 512 | 9.8 | −77.91 | −11.36 | 2.93 | 1.67 | 0.24 | | | |
| *Downstream groundwater (DG)* | | | | | | | | | | | | | | | |
| G26 | 10/6/2015 | 348 | 40 | | | | | −85.19 | −12.11 | | | | 6.91 | | |
| G27 | 29/7/2015 | 323 | 280 | 9.0 | 18.3 | 244 | | −79.83 | −12.21 | | | | | 23.5 | 10127 |
| G28 | 3/8/2015 | 353 | 45 | 9.0 | 13.2 | 246 | 8.0 | −78.02 | −11.47 | | | | 2.90 | | |
| G29 | 11/6/2015 | 347 | 380 | | | | | −86.39 | −12.33 | | | | 3.64 | 34.3 | 7001 |

[a] m a.s.l. = m above sea level. [b] Artesian well. [c] Contamination.





**Table 2.** Calculated results for CFC atmospheric partial pressures (pptv), groundwater apparent ages (PFM), fraction of post−1940 water (BMM) and mean transit times (DM, EPM, EMM).

| Sample ID | Atmospheric partial pressures (pptv) | | | Mixing post−1940 water in decimal year (F12/F113) | Fraction of post−1940 water (BMM[a], %) | Apparent age (PFM) (years)[b] | | | Mean transit times (F12) (years)[c] | | | | |
|---|---|---|---|---|---|---|---|---|---|---|---|---|---|
| | CFC−11 | CFC−12 | CFC−113 | | | CFC−11 | CFC−12 | CFC−113 | EPM (1.5) | EPM (2.2) | DM (0.03) | DM (0.1) | EMM |
| G3 | 179.59 | 476.18 | 70.88 | 1989.5 | 100 | 32.8 | 25 | 25 | 19 | 22 | 39 | 47 | 16 |
| | | | | 2002.5 | 87 | | | | | | | | |
| G5 | 10.42 | 43.99 | 4.04 | 1983 | 12 | 54.5 | 52.7 | 47 | 101 | 73 | 91 | 160 | 440 |
| G7 | 18.49 | 68.99 | 6.85 | 1984.5 | 18 | 51.25 | 49.2 | 43.2 | 89 | 66 | 82 | 139 | 270 |
| G8 | 122.11 | 280.24 | 36.42 | 1987.5 | 64 | 39 | 36.2 | 30.7 | 43 | 39 | 52 | 71 | 49 |
| G9 | 85.03 | 251.10 | 27.96 | 1985 | 66 | 41.9 | 38 | 32.9 | 47 | 42 | 54 | 76 | 58 |
| G15 | 58.15 | 202.68 | 26.99 | 1988 | 45 | 44.5 | 40.4 | 33.1 | 55 | 47 | 59 | 86 | 77 |
| G16 | 177.81 | 380.91 | 48.36 | 1987 | 89 | 33 | 29.6 | 28.3 | 30 | 31 | 45 | 57 | 29 |
| G20 | 100.11 | 257.11 | 31.36 | 1986.5 | 62 | 40.5 | 37.8 | 32 | 45 | 41 | 54 | 75 | 56 |
| G24 | 99.90 | | | | | 40.7 | | | | | | | |
| G25 | 180.79 | 388.92 | 48.83 | 1984.9 | 91 | 32.6 | 29 | 28.3 | 30 | 30 | 44 | 56 | 28 |

[a] BMM=binary mixing model, assuming a mixture of old water with young water (post–1940). [b] PFM=piston flow model, apparent ages are calculated by contrasting the Northern Hemisphere atmospheric air curve (http://water.usgs.gov/lab/software/air/cure/) based on the PFM. [c] Lumped parameter models: DM=dispersion model with $D_P$ (in Eq. (3)) of 0.1 and 0.03, EPM=exponential piston–flow model with $\eta$ (in Eq. (2)) of 2.2 and 1.5, EMM=exponential mixing model. F12 is short for CFC–12.

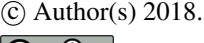



**Figure 1.** Maps showing (**a**) regional location of the Manas River Basin (modified after Ma et al., 2018), (**b**) surface water (river, reservoir and irrigation ditch) system (modified after Cui et al, (2007) and Ji, (2016)) and (**c**) geological cross–section of the study area for A–A′ line shown in (**b**).





**Figure 2.** Water sampling sites and unconfined groundwater head contours (in meters) in the headwater catchments of Manas River. UG=Upstream Groundwater, MG=Midstream Groundwater, DG=Downstream Groundwater.




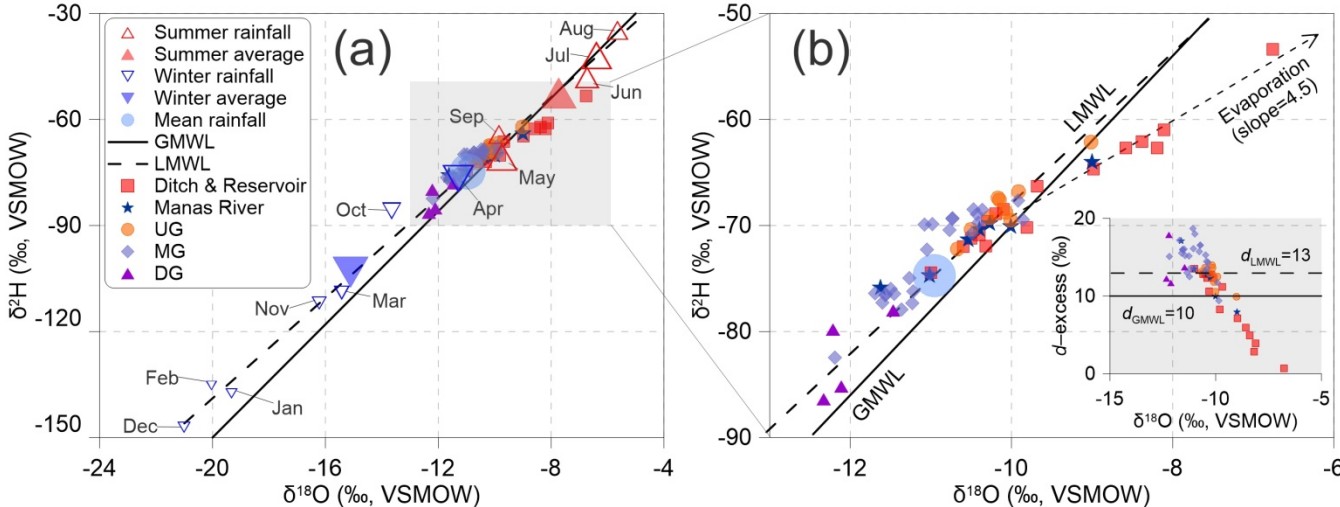

**Figure 3.** (**a**) Plot of stable isotopes of surface water and groundwater from the mountain to the oasis plain as compared to the global meteoric water line (GMWL; Craig, 1961) and the local meteoric water line (LMWL, rainfall in Urumqi station of International Atomic Energy Agency (IAEA) networks during 1986 and 2003; IAEA, 2006). The size of the hollow triangles stands for the relative amount of precipitation. 'Mean rainfall' refers to the annual amount–weighted mean rainfall isotopic value. (**b**) Plot of $\delta^2H$ vs. $\delta^{18}O$ and inserted plot $d$–excess vs. $\delta^{18}O$. UG=Upstream Groundwater, MG=Midstream Groundwater, DG=Downstream Groundwater.





**Figure 4.** Piper diagram highlights the HCO₃–SO₄–Na type of waters. The coloured symbols represent the mean values calculated from the hydrochemistry data (light grey hollow symbols) reported by Ma et al. (2018). The error bars are shown in the cation and anion diagrams



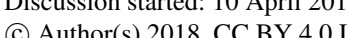

Figure 5. Concentrations of CFC–11, CFC–12 and CFC–113 (pptv) in the groundwater of this study area sampled in 2015 compared with the time series trend of Northern Hemisphere atmospheric mixing ratio at a recharge temperature of 10 ºC. Data is available at < http://water.usgs.gov/lab/software/air/cure/>.





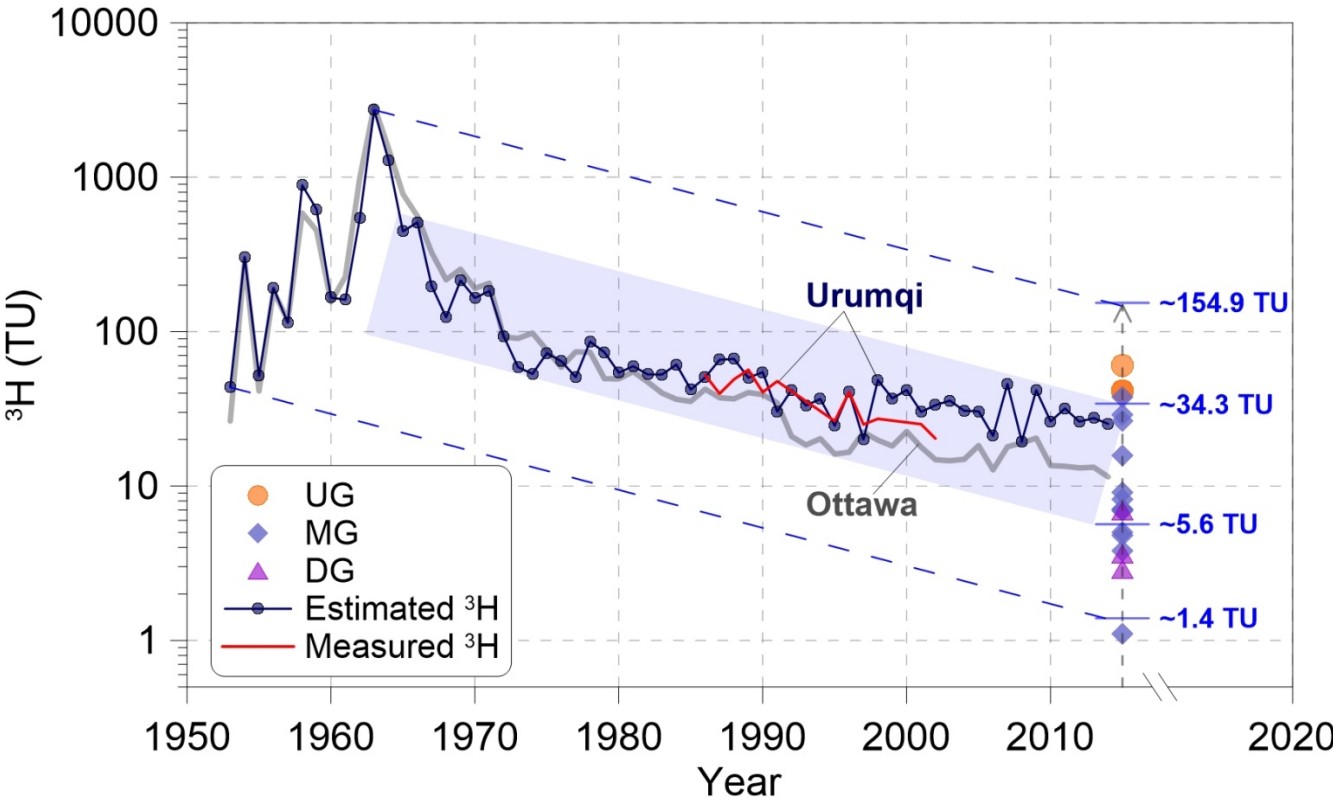


**Figure 6.** Plot of the reconstructed and measured time series of $^3$H activities in precipitation in Urumqi station between 1953 and 2014. The blue dashed lines and shaded field were drawn using the half–life (12.32 yrs) of tritium decayed to 2014.





**Figure 7.** (a) Distributions of $^3$H and $^{14}$C activities with distance to mountain; (b) Distributions of $^3$H, CFC–12 and $^{14}$C$_{corr}$ ages with distance to mountain. 'East bank' refers to groundwater CFC–12 apparent ages in the east bank of the 'East main canal'.



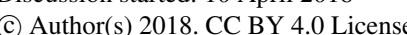

**Figure 8.** Tritium and CFCs (CFC–11, CFC–12 and CFC–113) output vs. mean transit times for different lumped parameter models

estimated using Eqs. (1) to (4). The input $^3$H activity and CFCs concentration are using the estimated $^3$H activities in precipitation in

Urumqi station (Fig. 6) and Northern Hemisphere atmospheric mixing ratio (Fig. 5), respectively. The UG (orange), MG (light blue) and

DG (purple) $^3$H activities and CFC concentrations are shown by shaded fields.





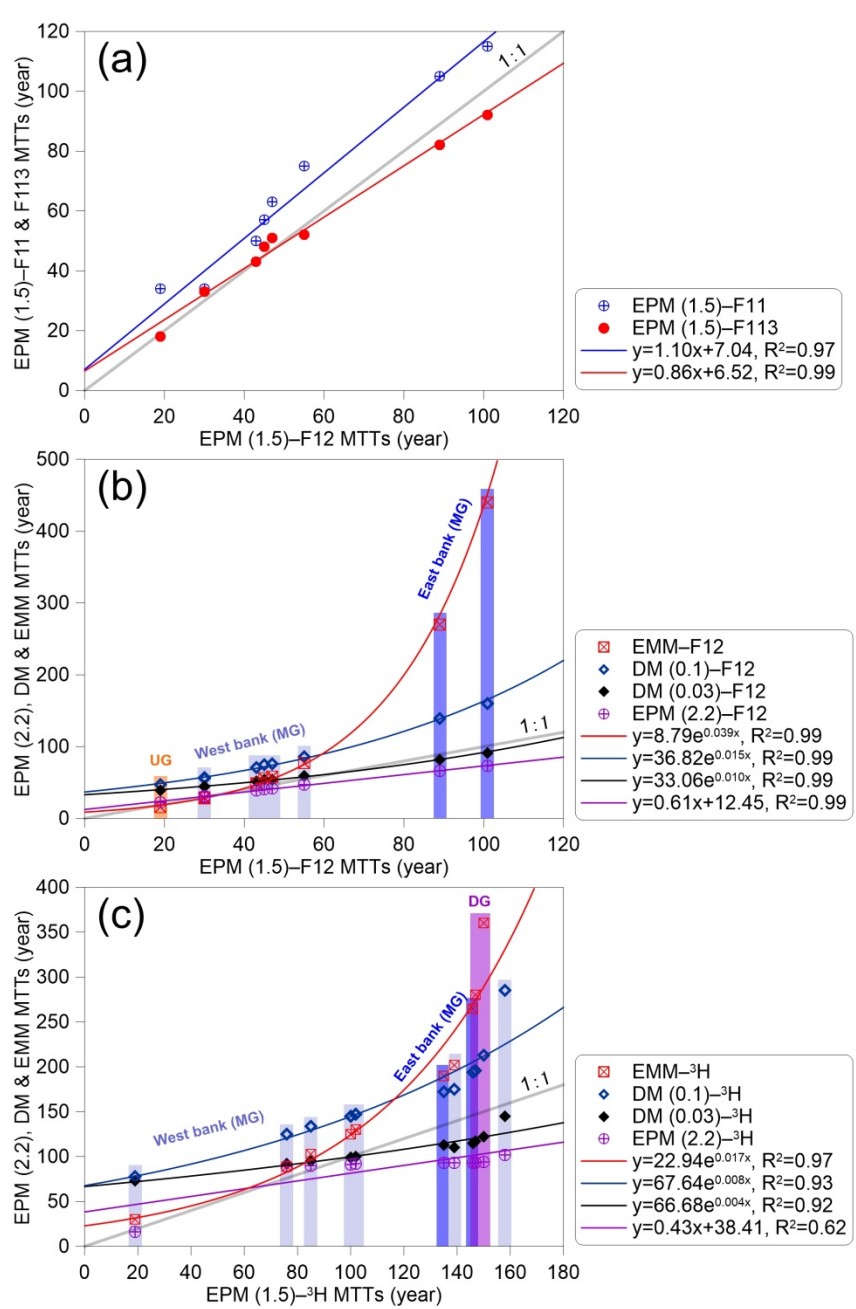

**Figure 9.** (**a**) EPM (1.5) MTTs and (**b**) EPM (2.2), DM & EMM MTTs vs. EPM (1.5)–F12 MTTs (CFC–12 MTTs using EPM (1.5)), (**c**) EPM (2.2), DM & EMM MTTs vs. EPM (1.5)–$^3$H MTTs. F11, F12 and F113 are short for CFC–11, CFC–12 and CFC–113. The shaded fields correspond to water samples from the UG (orange), West (light blue) and East bank (blue) of the 'East main canal' of MG, and DG (purple).



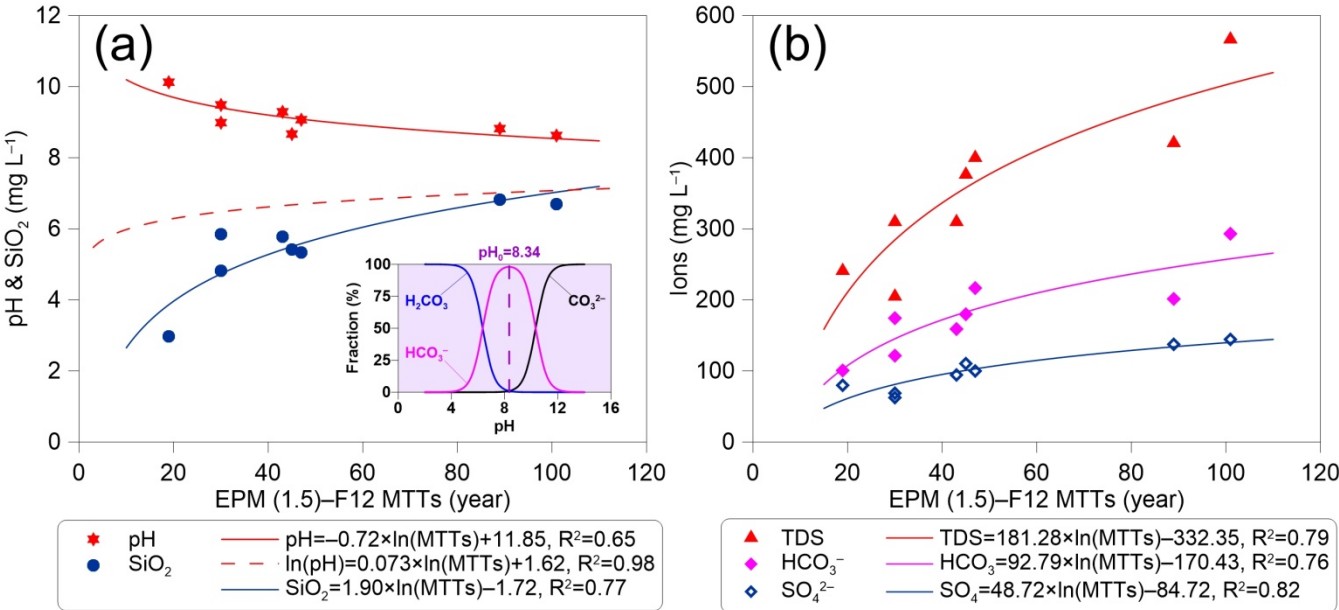

**Figure 10.** (**a**) pH & silica (SiO$_2$) and (**b**) ions including sulfate (SO$_4^{2-}$), bicarbonate (HCO$_3^-$) and total dissolved solids (TDS) vs. EPM (1.5)–F12 MTTs (CFC–12 MTTs using EPM (1.5)). The dashed red line in (**a**) is from Morgenstern et al. (2015).





**Figure 11.** Plots showing relationships of (**a**) CFC–113 vs. CFC–12 (**b**) CFC–11 vs. CFC–12, (**c**) CFC–113 vs. CFC–11, and (**d**) $^3$H activity (TU) in Urumqi precipitation decayed to 2014 vs. CFC–12 in pptv for Northern Hemisphere air. The data are compared to four hypothetical mixing models. The solid lines correspond to the piston flow model (PFM) and the short–dashed lines show the binary mixing model (BMM). The number referenced to the EPM (1.5) and EMM lines are mean transit times. The '+' denotes selected apparent ages. The shaded regions in (**a**) indicate no post–1988.5 waters mixing and in (**d**) indicate concentrations that could arise due to mixing water of different ages.