# Peer review of "Groundwater mean transit times, mixing and recharge in faultedhydraulic drop alluvium aquifers using chlorofluorocarbons (CFCs) and tritium isotope (3H)"

_Hydrology and Earth System Sciences, 2018_

## Referee Comment (RC1) · Anonymous Referee #1 · 14 May 2018

General Comments The paper reports CFC, tritium, carbon-14 and stable isotope measurements for groundwater in the Manas River Basin in China and uses them to estimate mean transit times for the complex mixtures of groundwaters in the area resulting from the complicated geology.

The complications of the subject combined with English that is not quite right make this a difficult read. However, the paper addresses relevant scientific questions suitable for publication in HESS, with novel concepts and ideas. Substantial conclusions are reached.

[Figure]

The methods are valid and described satisfactorily, and title and references are well done. There is a problem with the abstract (see below) and consequently the overall structure needs improvement. Some of the figures are complex and could be explained better.

Specific Comments

1) A major problem is that there appears to be a disconnect between the abstract/conclusions and the rest of the paper. The following sentence from the abstract/conclusions:

"The thrust faults were found to play a paramount role on groundwater flow paths and MTTs due to their block water features, where the relatively long MTTs were found near the Manas City with shorter distance and smaller hydraulic gradients."

is not supported by any discussion in the paper. Yes, it may be supported by implication from the results, but such support needs to be made explicit (possibly in its own subsection since this is an important conclusion).

2) The meaning of the phrase "block water features" is not clear, possibly it means areas where there are strong (semi-vertical) contrasts in hydraulic conductivity (due to the thrust faults).

3) Use of "apparent" ages in the preliminary discussion (Section 4.2.1) is defensible as described.

4) Strictly, groundwater has "residence time" or "mean residence time"/"MRT" (being the time water takes to travel through a groundwater system to where it is sampled by a bore), rather than "transit time" or "mean transit time"/"MTT" which is generally reserved for streamflow (being the time for water to transit through the catchment and into the stream). Consequently, the word "residence" should be substituted for the word "transit" wherever "transit" appears. And also "MRT" for "MTT".

5) A selection of comments on the English are given below, to help the clarity of the

writing. There are many other very small infelicities in the English.

Technical Corrections

P1 L24-25 Change to "Quite 'modern' recharge is found in the south of the fault with young (post–1940) water fractions of 87–100 %, . . ." from "The quite 'modern' recharge in the south of the fault with young (post–1940) water fractions of 87–100 % is obtained, . . ."

P2 L51 "instead of" not "over for"

P2 L53 "closed" not "close"

P3 L89 "common" not "true"

P3 L90-91 "Pumping from long-screened wells (of which there are over 10,000, Ma et al., 2018) . . ." not "Pumping from the long–screened over 10 000 boreholes (Ma et al., 2018) . . ."

P3 L93 "result from" not "impacted by"

P3 L94 "insufficiently recognised" not "insufficient recognition"

P4 L106 "total" not "totally"

P4 L107 "intermittently active" not "intermittent activity"

P4 L117 "depth" not "buried depth"

P6 L177 "Manas River Basin" not "MRB"

P9 L259 & 264 "slope" not "slop"

P10 L300 "we use" not "one assign"

P10 L304 "increasing" not "elevated"

P11 L323 "indicates a larger fraction of 1960s precipitation recharge for G4 . . ." not

"indicate that more fractions of the 1960s precipitation recharge was occurred for G4 . . ."

P12 L370 "generally" not "totally" and "overlap" not "overlapping"

P13 L397 delete "far"

P14 L413 "series" not "serious"

P14 L423 "Overall" not "Totally"

P15 L448 "permit" not "permitting"

P15 L465 "other sources" not "either source" (?)

P15 L466 "decreases" not "decrease"

P16 L488 "which did not contribute groundwater recharge" not "which had non–contributes to groundwater recharge"

P17 L520 delete "have occurred"

P19 L578 ". . . area) imply invasion of modern contaminants, . . ." not ". . . area), implying the modern contaminants invading, . . ."
* * *

---

## Referee Comment (RC2) · Anonymous Referee #2 · 15 May 2018

As indicated by the title, this manuscript presents the results of a groundwater dating and mixing study conducted using two different atmospheric tracers (CFCs and tritium). The two aims of the study were to (i) relate "ages" to local and general hydrogeological conditions and (ii) explore the possibility to use mineralisation as proxy for environmental tracers. I agree with referee #1 concerning the style, which is a huge disservice to the manuscript by its approximate use of technical terms and the general turn of phrase. I disagree however with the novelty (I do not see any) and the "substantial conclusions" (very unsubstantial and too dependent on mean transit time

calculations that at present look extremely weak). As far as comment 4 of referee #1 is concerned, I think it is simply a matter of opinion and taste to use "transit time" instead of "residence time" (I prefer transit time because my work is related to solute transport problems, and "transit time" conveys this very idea of transport). One can argue over that, but it is really a hair splitting exercise.

Overall, the authors seem to have read sufficiently thoroughly the existing literature on the subject as well as the most recent developments (such as Kirchner's analysis of the effect of heterogeneity on mean transit time estimation using amplitude damping) and understood the different problems and pitfalls relevant for their study. However, the phrasing is sometimes very awkward and tends to obfuscate what the authors mean (see specific comments below). But above all, I am missing a strong reason for this study to be published at all. As case study, it does not go beyond the classical scheme of sampling a few boreholes, analyse the groundwater samples for one or more tracers, calculate some kind of "age" and correlate it to depth or water chemistry. Doing so however, the authors try to apply different methods (lumped-parameter modelling, binary mixing) without presenting a clear roadmap. Model choice in particular is strangely presented: first, "apparent age" is presented as "based on the hypothesis of piston-flow". Then that very piston-flow model is used although mixing is supposed to be "most likely" either within the aquifer or at the sampling point. This is completely contradictory and there is no reason not to apply another model to the CFC data (and for that matter, to the 14C data as well. See Custodio et al., 2018). I know it is customary to interpret CFC data assuming piston-flow, but it is nonetheless a priori wrong. Model choice must be substantiated from knowledge of the hydrogeological situation and the sampling scheme (Maloszewski and Zuber, 1982; Leray et al., 2016). Later on in the manuscript however different models are used in the binary mixing plots, and model choice is discussed briefly. Why use the "apparent age" concept at all, then ? This is confusing and reads like the two authors have written separately different parts of the manuscript and then pasted the two parts together. I also have my doubts concerning the calculations of the mean transit times as they are presented. The method

with which the tritium input has been reconstructed is not documented properly (which stations were used, and how long were the availabe time series ?) and the estimated modern value (31 TU) seems extremely high compared to Western Europe for instance (about 6 TU). Is that because of the Chinese nuclear tests of the 60s and 70s that are being referred to in the introduction, or the result of some kind of regional effect ? Furthermore, the authors do address the non-uniqueness problems that are bound to arise when calibrating an exponential piston-flow or a dispersion model (2 free parameters each) using a single tritium measurement in aquifers that still retain some of the "bomb tritium" (see Stewart et al., 2010 for details), but in a terribly confusing way and without first explaining the rationale and the approach taken. I suppose figure 8 was meant to show the range of parameters that match the measured tracer concentrations. That's commendable, but badly explained. In the final step relating mineralisation to transit time, the authors finally select the EPM calibrated with the CFC12 measurements, but this is once again presented in a unclear fashion.

The discussion is too long, relies too much on untested and untestable hypotheses, and presents so many singular and unfocused results that it is difficult for the reader to grasp a clear picture of their meaning and significance. The paragraph on "apparent age" should be scraped altogether and the different estimates of "age" (i.e. mean transit time of the respective model) and mixing ratios organised in a clear and synthetic manner.

All in all, the manuscript must be seriously reorganised and streamlined. The calculation of mean transit times of the different tracers must be redone, removing entirely the "apparent age" nonsense and explaining clearly the different steps taken by the authors to (I) select a model (II) explore model parameter range and (III) compare the different results obtained from tritium, the CFCs and carbon 14. Interpretation of the obtained "ages" in terms of hydrogeology and its correlation to hydrochemistry must then be presented in a clear and synthetic fashion. Only when this is done might the manuscript rise above an unoriginal and confusing rehash of previous studies, and

could be considered for publication.

References: Custodio, E., Jodar, J., Herrera, C., Custodio-Ayala, J., Medina, A., Changes in groundwater reserves and radiocarbon and chloride content due to a wet period intercalated in an arid climate sequence in a large unconfined aquifer, Journal of Hydrology 556, 2018, 427-437

Leray, S., Engdahl, N.B., Massoudieh, A., Bresciani, E., McCallum, J., Residence time distributions for hydrologic systems: Mechanistic foundations and steady-state analytical solutions, Journal of Hydrology 543, 2016, 67-87

Maloszewski, P., Zuber, A., Determining the turnover time of groundwater systems with the aid of environmental tracers: 1. Models and their applicability, Journal of Hydrology 57, 1982, 207-231

Stewart, M.K, Morgenstern, U., McDonnell, J.J., Truncation of stream residence time: how the use of stable isotopes has skewed our concept of streamwater age and origin, Hydrological Processes 24, 2010, 1646-1659

Specific comments: Please ask the help of a proof reader to help improve readability

L11: Why is it crucial ? Please explain or leave that out.

L15: "indicating the rainfall recharge..." You mean that the young water component is higher than in samples with lower tritium activity.

L.29: The title of this section is not very telling, and this is not really what the study is about, is it?

L33: "may be renewable". What do you mean ? Something about short turnover time ?

L37-39: Rewrite the entire sentence.

L38: "and may be inferred". You mean "must be inferred".

[Figure]

L39: "at" steady-state, not "in" steady-state.

L40: "Three types of transit time". You mean three time windows ?

L46: It's not variability, rather time span.

L48: "in a similar function with" should read "in a similar way to"

L51: replace "over" with "than".

L54: You mean that increasing transit time through the aquifer leads to increasing mineralisation.

L55: Please explain why tritium is "the only true age tracer", namely because it is part of the water molecule.

L56 (entire paragraph): Why mention the southern hemisphere at all, since the study takes place in the northern hemisphere ? This is useless information.

L66: "may be used to estimate MTTs" should read "must be used to estimate MTTs". And explain why (non-unicity problems. . .).

L69: You are confusing residence time and degradation half-life. The residence time of the CFCs in the atmosphere is no different from that of tritium or any other tracer. The difference lies in their half-lives (degradation for CFCs, decay for tritium), which are very long for the CFCs.

L78-82: Please rephrase the entire sentence.

L89: "Mixing [. . .] is particularly true...". You don't know that, it's a probable hypothesis !

L93-95: "The MTTs that impacted...". This sentence makes no sense. Rewrite.

L106: "with totally length" should read "with total length".

L107: "was intermittent activity" should read "was intermittently active".

L110: So the different aquifers are all fractured rock aquifers.

L114: "is macroscopically similar". What do you mean with "macroscopically" ?

L120: "is as large" should read "is as deep".

L124: How many samples were taken altogether ? And are there any information concerning screening depth and size (fully penetrating wells or not) ? This is important information to guide model choice.

L126: What was the rationale for separating the samples into three groups ? For instance, why is G13 MG while G26 is DG ? DG seems like the downgradient boundary. Did you use the piper diagrams to separate the samples ?

L152: "were followed" is used multiple times, but should read "after" or "following".

L173: "refers" is not the proper verb. Use "depends" for instance.

L179: What are low latitude countries ?

L189: The entire procedure is correctly explained, and also the fact that "apparent age" implies piston-flow transit time distribution, but why use apparent age in the first place ? Piston-flow is one model among many, as the authors explain later in the manuscript. Furthermore, the entire concept of "age" is problematic and should be replaced by mean transit time or mean residence time (for an in-depth discussion, see Suckow, The age of groundwater-Definitions, models and why we do not need this term, Applied Geochemistry 50, 2014, 222-230).

L194: What do you mean by "closed system" ? Physically bounded ?

L208: "were given below as transit time distribution function" should read "were selected and are given below".

L219: You should also explain here how you planned to choose between these competing models.

L235: Why present an equation you will not be using for lack of appropriate data ?

L240: This is true for the piston-flow model only ! See Custodio et al. for details.

L250: So the entire paragraph boils down to using literature values for the initial 14C activity. Make it shorter and to the point.

L264: Check the discussion paper by Benettin et al. in review in HESS for the latest developments on the "evaporation slope".

L274: The entire paragraph is too short and should explain clearly the approach adopted to calculate "ages" from the tracer data (model and model parameter choices !). I strongly advise against using binary mixing diagrams, and encourage the authors to use a multi-tracer modelling approach trying to find a single optimum or optimal parameter regions for the different tracers.

L277: The paragraph on "apparent age" makes no sense for the reasons given above. I disagree with the proposition that "they [apparent ages] provide a good first approximation for groundwater age". There is no reason to prefer the piston-flow model which is implied by the "apparent age" concept over other models. This argument has been for years a lazy way to skip responsibility in choosing one model based on knowledge of the hydrogeological situation and sampling.

L297: "which confirms". A performative statement confirms nothing. You are supposing this is the case !

L317: Shortly explain the method used to estimate the tritium input (linear regression ? And how long were the time series used ?). The reference to Han et al. is not very useful as the authors of that paper themselves refer to an IAEA publication without further explanations.

L318: A background of 31 TU is very high compared to Western Europe (about 6 TU). How come ?

[Figure]

L413: What do you mean by "serious" ?

L414: "tend toward more discrete with their increase". I do not understand this part of the sentence.

L448: The paragraph on hydrochemistry is not bad, but underdeveloped and badly organised. State again what you're looking for first. A good correlation between hydrochemistry and "ages" calculated using some of the TTD models might be a way to constrain or guide model choice, but the authors do not really state that explicitly, although that would be interesting and relatively new.

L491: The entire chapter 4.5.1 makes no sense. You must first decide which model is the most appropriate, and then calculate metrics such as mean transit time, young water fraction, etc... You cannot both calculate water fractions using a binary mixing strategy (assuming piston-flow) AND later use an EPM. The same remark applies to chapter 4.5.2.

L498: "no post-1988.5". Please round this off...

L509: Why do you treat "apparent age" as some kind of different measure of transit time than MTTs "estimated from the EPM" ? This is doing the analysis the wrong way around. First find a way to select a model, then discuss the obtained "ages" instead of hypothesizing on tons and tons of different "ages" that are meaningless because they were obtained disregarding the actual situation. This leads nowhere.

L541: Before engaging in complicated mixing scenarios, you should first try to find one model and one parameterisation that fits both the CFCs, tritium and carbon 14. Only if that search does not succeed should additional mixing be introduced. Please note that the binary mixture approach proposed by Plummer et al. is only one way of doing so, and a particularly weak one at that because it assumes per default a piston-flow distribution of transit time of each component (other models can be integrated, but it becomes quickly very cumbersome). Another way to include the mixing of different
reservoirs is to combine models (say two exponential models, each representing one distinct source) following Piotr Maloszewski and coworkers or Mike Stewart and Uwe Morgenstern. Binary plots such as those of figure 11 suffer from the limitation that you have to recalculate the mixing line for each parameterisation of each model, and they cannot really replace a multi-tracer lumped-parameter modelling approach, where the objective function reduces simultaneously the prediction error of all tracers.

L562: Solutions are obtained, explanations are devised.

L572: What are mixing rates ? You mean mixing ratios ?

L575: "The thrust faults were found to play a paramount role on groundwater flow path". These are not conclusions, but hypotheses very weakly suggested by the analysis of the environmental tracers, which is itself very shaky. I hardly call that evidence. Please refrain from drawing conclusions if the data necessary to test hypotheses is not available (as is the case here).

L585: "due to the highly complex groundwater system...". This is no explanation at all ! Indeed, devising a conceptual model that could explain why CFC derived "ages" correlate well with mineralisation while tritium derived "ages" do not could be a useful task (but you should first redo the calculation of the "ages" as suggested above). On the one hand, the correlation between CFC12 and hydrochemistry might be an artifact, given that the area sampled is so large and hydrogeologically diverse. On the other hand, there might be some kind of systematic shift between tritium and CFC ages if differences are due to the unsaturated zone. Maybe a diffusion model using the unsaturated zone thickness might be useful. Still much work to do...

Figure 7, 8 and 9: The figures are incredibly cluttered and very difficult to read, especially figure 9 (not to mention the legend).

Figure 10: Why are there so few points on each graph, since you sampled at 29 locations according to table 1 ?

Figure 11: As I wrote above, binary mixing diagrams rapidly tend to show their limits. After two or three mixing lines for different models are drawn, reading becomes nigh impossible. Importantly, error bars are missing for the CFCs and for tritium. I suspect that with error bars, selecting a model visually will become impossible (the lack of sensitivity is another limitation of binary mixing diagrams, ).

[Figure]

---

## Author Comment (AC1) · 25 Aug 2018

"Groundwater mean transit times, mixing and recharge in faulted–hydraulic drop alluvium aquifers using chlorofluorocarbons (CFCs) and tritium isotope ($^3$H)" by Ma, B., Jin, M., Liang, X., Li, J., Hydrol. Earth Syst. Sci., doi:10.5194/hess-2018-143.

We appreciate the many valuable suggestions and helpful comments of **Anonymous Referee #1**. We have seriously considered all of the suggestions and comments and have attempted to address each of the comments point-by-point. Detail explanations are as follows.

Author's response – Line numbers referring to the old and revised version manuscripts are preceded by L and RL, respectively.

**Anonymous Referee #1**

**General Comments**

The paper reports CFC, tritium, carbon-14 and stable isotope measurements for groundwater in the Manas River Basin in China and uses them to estimate mean transit times for the complex mixtures of groundwaters in the area resulting from the complicated geology.

The complications of the subject combined with English that is not quite right make this a difficult read. However, the paper addresses relevant scientific questions suitable for publication in HESS, with novel concepts and ideas. Substantial conclusions are reached.

The methods are valid and described satisfactorily, and title and references are well done. There is a problem with the abstract (see below) and consequently the overall structure needs improvement. Some of the figures are complex and could be explained better.

Response: We would like to thank you very much for taking the time to review our manuscript and for their generally positive feedback. We will ask a proof reader to modify the language to help improve readability. We have reorganized the structure and tried our best to present a clear roadmap to readers. We also agree with you that some of the figures are complex which have also been pointed out by Ref #2.

The outline of the manuscript have been reorganized as follows:

Title: Application of environmental tracers for investigation of groundwater mean residence time and aquifer recharge in faulted–hydraulic drop alluvium aquifer

1. Introduction
2. Geological and hydrogeological setting
3. Materials and methods
    3.1 Water sampling
    3.2 Analytical techniques
    3.3 Groundwater dating
        3.3.1 CFCs indicating modern water recharge
        3.3.2 The apparent $^{14}$C ages
        3.3.3 Groundwater mean residence time estimation
4. Results and discussion

Figures 6, 8 and 9 have been redrawn as follows:

[Figure]

Figure 6. Tritium concentration (TU) of groundwater water samples of upstream groundwater (UG), midstream groundwater (MG), and downstream groundwater (DG). Time series of tritium concentration in precipitation at Ottawa, Urumqi, Hongkong, and Irkutsk were obtained by GNIP in IAEA (https://www.iaea.org/). The blue dashed lines and shaded field were drawn using the half–life (12.32 yrs) of tritium decayed to 2014. (It is Fig. 4 in the revised manuscript)

[Figure]

**Figure 8.** Tritium and CFCs (CFC–11, CFC–12 and CFC–113) output vs. mean residence times for different lumped–parameter models estimated using Eqs. (2) to (5). The input $^3$H activity and CFCs concentration are using the estimated $^3$H activities in precipitation in Urumqi station (Fig. 4) and Northern Hemisphere atmospheric mixing ratio (Fig. 3), respectively. (It is Fig. 10 in the revised manuscript)

[Figure]

**Figure 9.** (**a**) MRTs with EPM (1.5) of CFC–12 vs. CFC–11 & CFC–113, (**b**) CFC–12 MRTs with EPM

(1.5) vs. EPM (2.2), DM & EMM, and (**c**) 3H MRTs with EPM (1.5) vs. EPM (2.2), DM & EMM. (It is Fig. 11 in the revised manuscript)

**Specific Comments**

1)    A major problem is that there appears to be a disconnect between the abstract/conclusions and the rest of the paper. The following sentence from the abstract/conclusions:

"The thrust faults were found to play a paramount role on groundwater flow paths and MTTs due to their block water features, where the relatively long MTTs were found near the Manas City with shorter distance and smaller hydraulic gradients."

is not supported by any discussion in the paper. Yes, it may be supported by implication from the results, but such support needs to be made explicit (possibly in its own subsection since this is an important conclusion).

Response: This sentence has been deleted. To make the abstract and conclusions to be more clear and well-founded, we have revised the abstract/conclusions and delete some incorrect statements. Yes, this conclusion is important but not supported by strong supporting evidences in the paper. Indeed, there are some results that show large differences on both sides of the thrust fault. For examples, there is a level difference of 130 m hydraulic drop (Fig. 1c) in the south margin in Shihezi (SHZ), $^3$H activities of groundwater decrease rapidly along the Manas River motion in the north of the fault but show relatively the highest values in the south of the fault (Fig. 8). These results still can not support the conclusion explicitly "The thrust fault were found to play a paramount role on groundwater flow paths …".

The revised abstract is as follows (RL12–26):

"Documenting the groundwater residence time and recharge source is crucial for water resource management in the alluvium aquifer of arid basin. Environmental tracers (CFCs, $^3$H, $^{14}$C, $\delta^2$H, $\delta^{18}$O) and hydrochemistry of groundwater were used to assist our understanding of groundwater mean residence times (MRTs) and aquifer recharge in faulted–hydraulic drop alluvium aquifers in the Manas River Basin (China). The very high $^3$H activities (41.1–60 TU) of groundwater in the Manas River upstream (south of the fault) indicate the rainfall recharge during the nuclear bomb (since the 1960s). Carbon–14 groundwater ages increase with distance (3000–5000 yrs in the midstream to > 7000 yrs in the downstream) and depth, as well as with low $^3$H activities (1.1 TU) and more depleted $\delta^{18}$O values, confirming that the deeper groundwater is derived from paleometeoric recharge in the semi–confined groundwater system. MRTs estimated using an exponential–piston flow model vary from 19 to 101 yrs for CFCs and from 19 to 158 yrs for $^3$H, which show much longer MRTs for $^3$H than CFCs may be due to the time lag through the thick unsaturated zone. The remarkable correlations between CFCs rather than $^3$H MRTs and pH, $SiO_2$ and $SO_4^{2-}$ concentrations allow first–order proxies of MRTs for groundwater at different times to be made. Quite 'modern' recharge is found in the south of the fault with young (post–1940) water fractions of 87–100 %, while in the north of the fault in the midstream area the young water fractions vary from 12 to 91 % based on the CFC binary mixing method. This study shows that the combination of CFCs and $^3$H residence time tracers have potential to study groundwater MRTs and identify the recharge sources for the different mixing end–members."

2)   The meaning of the phrase "block water features" is not clear, possibly it means areas where there are strong (semi-vertical) contrasts in hydraulic conductivity (due to the thrust faults).

Response: Yes, the phrase "block water features" is not a very appropriate statement in this paper. What we want to tell the reader is that there are strong contrasts in hydraulic conductivity due to the thrust fault. The variant hydraulic conductivity also can be reflected by the geological and hydrogeological settings. Previous studies (Wu, 2007; Zhao, 2010) and other geological survey works in the Manas River Basin have indicated that the thrust faults shown in Fig. 1b are compressional faults and thus of water–blocking feature, which can explain the "a level difference of 130 m hydraulic drop is observed due to the thrust fault in the alluvium aquifer (Fig. 1c)".

A recent study by Bresciani et al (2018) has distinguished the mountain–front recharge (MFR) and mountain–block recharge (MBR) by using hydraulic head, chloride and electrical conductivity data in the arid basin. MFR predominantly consists of stream infiltration in the mountain–front zone, and MBR consists of subsurface flow from the mountain towards the basin. Manas River Basin aquifers may receive the recharge from the south mountain through the MFR mechanism, and more specific analysis will be carried out in the future work.

3)   Use of "apparent" ages in the preliminary discussion (Section 4.2.1) is defensible as described.

Response: We find that the phrase "apparent CFC ages" has been widely used in many other literatures (e.g. Darling et al., 2012; Hagedorn et al., 2011; Han et al., 2012; Happell et al., 2006; Koh et al., 2012; Plummer et al., 2006; Qin et al., 2011, 2012). However, a review paper by Suckow (2014) pointed out that the "apparent age" is "only well defined if the formula is given and if the tracer is stated". There are appropriate formulas for different tracers, such as $^{14}C$, $^{36}Cl$, $^{81}Kr$, $^{3}H/^{3}He$, and so on, but not for the CFCs, for $SF_6$ and for $^{85}Kr$. Therefore, Suckow (2014) thinks that, strictly speaking, the term "apparent age", should not be used for CFCs. This erroneous term "apparent age" for CFCs is also pointed out by Ref #2 ("L277: The paragraph on "apparent age" makes no sense for … and sampling").

We agree that the term "apparent age" for CFCs will not be used anywhere in our paper. As we know that the CFCs are synthetic organic compounds and largely released to the air since 1930s, and thus they have been regarded as very good tracers for dating young water recharge time (post–1940 recharge). Therefore, we would like to use CFCs to explain the modern water recharge features.

The revised contents can be seen in Section 3.3.1 (RL172–192) and in Section 4.2.2 (RL344–414):

Section 3.3.1 (RL172–192):

[revised manuscript text omitted]

4)    Strictly, groundwater has "residence time" or "mean residence time"/"MRT" (being the time water takes to travel through a groundwater system to where it is sampled by a bore), rather than "transit time" or "mean transit time"/"MTT" which is generally reserved for streamflow (being the time for water to transit through the catchment and into the stream). Consequently, the word "residence" should be substituted for the word "transit" wherever "transit" appears. And also "MRT" for "MTT".

Response: Agree and changes made. The term "transit" was changed to "residence" and term "MTT" was changed to "MRT", and we insisted on the "residence" and "MRT" throughout the manuscript.

We re-read the literatures and found that term "transit time" was numerously used to indicate the time for water to transit through the catchment and into the stream (Cartwright et al., 2018; Cartwright and Morgenstern, 2015, 2016; Hrachowitz et al., 2009, 2010; Morgenstern et al., 2010; Stewart and Morgenstern, 2016). Stewart et al. (2010) pointed out that "Residence time is the time spent in the catchment since arriving as rainfall. Transit time is the time taken to pass through the catchment and into the stream." Leray et al. (2016) have adopted a general but robust definition for the residence time "the amount of time a moving element has spent in a hydrologic system", and considered the terms residence time, transit time, travel time, age, and exposure time as equivalent in their discussions. Custodio et al. (2018) used both residence times and transit times for groundwater samples collected from springs and deep wells. In our study, all of the groundwater samples were collected from the wells/artesian wells. Thus, we tend to use the term "residence" instead of "transit" in our manuscript.

5)    A selection of comments on the English are given below, to help the clarity of the writing. There are many other very small infelicities in the English.

Response: We thank you very much for modifying the expressions of the manuscript. We will ask a proof reader to modify the language to help improve readability.

**Technical Corrections**

1)    P1 L24-25 Change to "Quite 'modern' recharge is found in the south of the fault with young (post–1940) water fractions of 87–100 %, …" from "The quite 'modern' recharge in the south of the fault with young (post–1940) water fractions of 87–100 % is obtained, …"

Response: Agree and changes made. The sentence was changed to (RL23): "Quite 'modern' recharge is found in the south of the fault with young (post–1940) water fractions of 87–100 %, while …".

2) P2 L51 "Instead of" not "over for"

Response: Changes made. "over" was replaced by "than".

3) P2 L53 "closed" not "close"

Response: Agree and changes made. "close" was replaced by "closed".

4) P3 L89 "common" not "true"

Response: Agree and changes made. "true" was replaced by "common".

5) P3 L90-91 "Pumping from long-screened wells (of which there are over 10,000, Ma et al., 2018) …" not "Pumping from the long–screened over 10 000 boreholes (Ma et al., 2018) …"

Response: Agree and changes made. The sentence was changed to (RL89–90): "Pumping from long-screened wells (of which there are over 10 000 boreholes, Ma et al., 2018) make groundwater mixing mostly likely.". Number "10 000" (not "10,000") is divided in groups of three using a thin space, complying with the "manuscript preparation guidelines for authors".

6) P3 L93 "result from" not "impacted by"

Response: Agree and changes made. "impacted by" was replaced by "result from".

7) P3 L94 "insufficiently recognised" not "insufficient recognition"

Response: Agree and changes made. "insufficient recognition" was replaced by "insufficiently recognised".

8) P4 L106 "total" not "totally"

Response: Agree and changes made. "totally" was replaced by "total".

9) P4 L107 "intermittently active" not "intermittent activity"

Response: Agree and changes made. "intermittent activity" was replaced by "intermittently active".

10) P4 L117 "depth" not "buried depth"

Response: Agree and changes made. Word "buried" was deleted.

11) P6 L177 "Manas River Basin" not "MRB"

Response: Agree and changes made. "MRB" was replaced by "Manas River Basin".

12) P9 L259 & 264 "slope" not "slop"

Response: Agree and changes made. Erroneous "slop" was changed to "slope".

13) P10 L300 "we use" not "one assign"

Response: Agree and changes made. "one assign" was replaced by "we use".

14) P10 L304 "increasing" not "elevated"

Response: Agree and changes made. "elevated" was replaced by "increasing".

15)  P11 L323 "indicates a larger fraction of 1960s precipitation recharge for G4 …" not "indicate that more fractions of the 1960s precipitation recharge was occurred for G4 …"

Response: Agree and changes made. The sentence was changed to (RL425–427): "First, $^{3}$H activities of groundwater in the upstream area increase from 41.1 (G1 and G2) to 60 TU (G4) with distance indicates a larger fraction of 1960s precipitation recharge for G4 than G1 and G2 groundwater samples".

16)  P12 L370 "generally" not "totally" and "overlap" not "overlapping"

Response: Agree and changes made. "totally" was replaced by "generally", and "overlapping" was replaced by "overlap".

17)  P13 L397 delete "far"

Response: Agree and changes made. "far" was deleted.

18)  P14 L413 "series" not "serious"

Response: Agree and changes made. "serious" was replaced by "series".

19)  P14 L423 "Overall" not "Totally"

Response: Agree and changes made. "Totally" was replaced by "Overall".

20)  P15 L448 "permit" not "permitting"

Response: Agree and changes made. "permitting" was replaced by "permit".

21)  P15 L465 "other sources" not "either source" (?)

Response: Changes made. ", and giving more accurate prediction of the contaminants like nitrate than either source of information" was deleted.

22)  P15 L466 "decreases" not "decrease"

Response: Agree and changes made. Erroneous "decrease" was changed to "decreases".

23)  P16 L488 "which did not contribute groundwater recharge" not "which had non–contributes to groundwater recharge"

Response: Agree and changes made. The sentence was changed to (RL552–553): "…, which did not contribute groundwater recharge in the arid northwest China".

24)  P17 L520 delete "have occurred"

Response: Agree and changes made. "have occurred" was deleted.

25)  P19 L578 "… area) imply invasion of modern contaminants, …" not "… area), implying the modern contaminants invading, …"

Response: Agree and changes made. The sentence was changed to (RL561): "… area) imply invasion of modern contaminants, which …".

---

## Author Comment (AC2) · 25 Aug 2018

"Groundwater mean transit times, mixing and recharge in faulted–hydraulic drop alluvium aquifers using chlorofluorocarbons (CFCs) and tritium isotope ($^3$H)" by Ma, B., Jin, M., Liang, X., Li, J., Hydrol. Earth Syst. Sci., doi:10.5194/hess-2018-143.

We appreciate the many valuable suggestions and helpful comments of **Anonymous Referee #2**. We have seriously considered all of the suggestions and comments and have attempted to address each of the comments point-by-point. Detail explanations are as follows.

Author's response – Line numbers referring to the old and revised version manuscripts are preceded by L and RL, respectively.

**Anonymous Referee #2**

1)    As indicated by the title, this manuscript presents the results of a groundwater dating and mixing study conducted using two different atmospheric tracers (CFCs and tritium). The two aims of the study were to (i) relate "ages" to local and general hydrogeological conditions and (ii) explore the possibility to use mineralisation as proxy for environmental tracers. I agree with referee #1 concerning the style, which is a huge disservice to the manuscript by its approximate use of technical terms and the general turn of phrase. I disagree however with the novelty (I do not see any) and the "substantial conclusions" (very unsubstantial and too dependent on mean transit time calculations that at present look extremely weak). As far as comment 4 of referee #1 is concerned, I think it is simply a matter of opinion and taste to use "transit time" instead of "residence time" (I prefer transit time because my work is related to solute transport problems, and "transit time" conveys this very idea of transport). One can argue over that, but it is really a hair splitting exercise.

Response: We would like to thank you very much for taking the time to review our manuscript. We think that the many valuable suggestions and helpful comments will help to improve our manuscript quality greatly.

We agree with you about the aims of the study and the opinion on the terms "transit time" and "residence time". We re-read the literatures and found that term "transit time" was numerously used to indicate the time for water to transit through the catchment and into the stream (Cartwright et al., 2018; Cartwright and Morgenstern, 2015, 2016; Hrachowitz et al., 2009, 2010; Morgenstern et al., 2010; Stewart and Morgenstern, 2016). Stewart et al. (2010) pointed out that "Residence time is the time spent in the catchment since arriving as rainfall. Transit time is the time taken to pass through the catchment and into the stream." Leray et al. (2016) have adopted a general but robust definition for the residence time "the amount of time a moving element has spent in a hydrologic system", and considered the terms residence time, transit time, travel time, age, and exposure time as equivalent in their discussions. Custodio et al. (2018) used both residence times and transit times for groundwater samples that collected from springs and deep wells. In our study, all of the groundwater samples were collected from the wells/artesian wells. Thus, we tend to use the term "residence time" instead of "transit time" in our manuscript.

In addition, we have reorganized the structure and tried our best to present a clear roadmap to readers. Some incorrect phrase or expressions, for example the "apparent CFC ages" (L277) and "The apparent ages (Table 2 and Fig. 5) estimated from the PFM …" (L509), which mislead the study and thus

have been revised and rewritten. In the revised manuscript, we tried to discuss the mean residence times and hydrochemistry evolution clearly and reasonably.

2)    Overall, the authors seem to have read sufficiently thoroughly the existing literature on the subject as well as the most recent developments (such as Kirchner's analysis of the effect of heterogeneity on mean transit time estimation using amplitude damping) and understood the different problems and pitfalls relevant for their study. However, the phrasing is sometimes very awkward and tends to obfuscate what the authors mean (see specific comments below).

Response: We have checked the manuscript carefully and have tried our best to modify some erroneous words and phrases that you and Ref #1 pointed out. Some ambiguous statements and sentences have also been rewritten. On the other hand, we will ask a proof reader to modify the language to help improve readability.

3)    But above all, I am missing a strong reason for this study to be published at all. As case study, it does not go beyond the classical scheme of sampling a few boreholes, analyse the groundwater samples for one or more tracers, calculate some kind of "age" and correlate it to depth or water chemistry. Doing so however, the authors try to apply different methods (lumped-parameter modelling, binary mixing) without presenting a clear roadmap.

Response: Manas River with the longest channel length and the largest river flow is representative among the hundreds of inland rivers in the Junggar Basin in the arid northwest China (Fig. 1a). There are more than 1 million people in the Manas River Basin. Groundwater is the very important and even the only water resources for the local residents. However, the paucity of research on groundwater mean residence time, mixing and recharge features have hampered the rational development and utilization of water resources in the Junggar Basin. We think that the case study of our manuscript is very essential and can be a good proxy for other analogous areas in the Junggar Basin.

Yes, we tried to estimate the mean residence time using lumped–parameter models. In addition, we also have tried to recognize the modern and paleo–meteoric recharge features. However, it is a pity that there are many deficiencies for the initial manuscript. We have reorganized the structure and tried our best to present a clear roadmap to readers. The outline of the manuscript have been reorganized as follows:

Title: Application of environmental tracers for investigation of groundwater mean residence time and aquifer recharge in faulted–hydraulic drop alluvium aquifer

1. Introduction
2. Geological and hydrogeological setting
3. Materials and methods
    3.1 Water sampling
    3.2 Analytical techniques
    3.3 Groundwater dating
        3.3.1 CFCs indicating modern water recharge
        3.3.2 The apparent $^{14}$C ages
        3.3.3 Groundwater mean residence time estimation
4. Results and discussion
    4.1 Stable isotope and major ion hydrochemistry

4)   Model choice in particular is strangely presented: first, "apparent age" is presented as "based on the hypothesis of pistonflow". Then that very piston-flow model is used although mixing is supposed to be "most likely" either within the aquifer or at the sampling point. This is completely contradictory and there is no reason not to apply another model to the CFC data (and for that matter, to the 14C data as well. See Custodio et al., 2018).

Response: We have deleted "apparent CFC ages" and rewritten the content of the model choice for mean residence time estimation (Section 3.3.3 (RL219–272)):

"Groundwater mixing may occur both within the aquifer and in the long–screened wells (Cook et al., 2017; Custodio et al., 2018; Visser et al., 2013). The groundwater residence times (ages) often display a wide range due to the recharge under various climate conditions (Custodio et al., 2018). With the aid of gaseous tracers (e.g. $^3$H, CFCs, $SF_6$ and $^{85}$Kr) one can describe the mixing distribution by a mixing model (Stewart et al., 2017; Zuber et al., 2005) and to obtain the groundwater mean residence times (MRTs). Lumped–parameter models (LPMs) is an alternative approach to interpret the MRTs for water flow through the subsurface systems to the output. For steady–state subsurface hydrologic system, on accounting of the $^3$H and CFCs tracers enter groundwater with precipitation are injected proportionally to the volumetric flow rates by nature itself, the output concentration in water at the time of sampling relating to the input $^3$H and CFCs can be described by the convolution integral (Małoszewski and Zuber, 1982):

$$C_{out}(t) = \int_0^\infty C_{in}(t-\tau)\, g(\tau)\, e^{-\lambda_{^3H}\tau}\, d\tau \qquad \text{for} \quad ^3H \quad \text{tracer} \tag{2a}$$

$$C_{out}(t) = \int_0^\infty C_{in}(t-\tau)\, g(\tau)\, d\tau \qquad \text{for} \quad \text{CFCs} \quad \text{tracer}, \tag{2b}$$

where $C_{out}$ is the tracer output concentration, $C_{in}$ is the tracer input concentration, $\tau$ is the residence time, $t-\tau$ is the time when water entered the catchment, $\lambda_{^3H}$ is the $^3$H decay constant ($\lambda_{^3H} = \ln 2/12.32$), and $g(\tau)$ is the system response function that describes the residence time distributions (RTDs) in the subsurface hydrologic system.

    In this study the CFC concentrations from the time series trend of Northern Hemisphere atmospheric mixing ratio (Fig. 3) and $^3$H concentrations in precipitation in Urumqi (Fig. 4) are treated as proxies for CFC and $^3$H recharge concentrations ($C_{in}$), respectively. The historical precipitation $^3$H activity in Urumqi station (Fig. 4) was reconstructed from the available data in the International Atomic Energy Agency (IAEA) using a logarithmic interpolation method. Precipitation $^3$H activities between 1969 and 1983 at Hongkong and Irkutsk with different latitudes were used (data is available at <https://www.iaea.org/>). The time series $^3$H activities as the input data are still necessary that is based on the following two considerations. First, Manas River Basin is located in the Northern Hemisphere, where the bomb–test [3]H activities were several orders of magnitude higher than in the Southern Hemisphere (Clark and Fritz, 1997; Tadros et al., 2014). [3]H activity in the atmosphere was superimposed over the China atmospheric nuclear tests from 1964 to 1974 in the arid northwest China, and thus the remnant [3]H activities are still affected by the tail–end of the bomb pulse. Second, the study area is more than 3500 km away from the western pacific, where [3]H activity in the atmosphere is evidently much higher than coastal sites due to the continental effect (Tadros et al., 2014). Furthermore, though [3]H activity in the atmosphere is known to vary between seasons (Cartwright and Morgenstern, 2016; Morgenstern et al., 2010; Tadros et al., 2014), the year–round mean values (Fig. 4) were adopted in this study.

Several RTDs have been described (Małoszewski and Zuber, 1982; Jurgens et al., 2012) and been widely used in studies of variable timescales and catchment areas (Cartwright and Morgenstern, 2015, 2016; Cartwright et al., 2018; Hrachowitz et al., 2009; Morgenstern et al., 2010, 2015; McGuire et al., 2005), of which the selection of each model depends on the hydrogeological situations in the hydrologic system to which it is applicable. The exponential–piston flow model (EPM) describes aquifer that containing a segment of exponential flow followed by a segment of piston flow. Piston flow assumes that water mixing from different flow lines is minimal and receiving little or no recharge in the confined aquifer, and the exponential flow assume a full mixing of water in the unconfined aquifer and receiving areally distributed recharge (Jurgens et al., 2012; Małoszewski and Zuber, 1982). Its RTD is

$$g(\tau) = 0 \qquad \text{for} \quad \tau < \tau_{\mathrm{m}}\left(1 - 1/\eta\right) \tag{3a}$$

$$g(\tau) = \frac{\eta}{\tau_{\mathrm{m}}} e^{\left(-\eta\tau/\tau_{\mathrm{m}} + \eta - 1\right)} \qquad \text{for} \quad \tau \geq \tau_{\mathrm{m}}\left(1 - 1/\eta\right) \tag{3b}$$

The dispersion model (DM) mainly measures the relative importance of dispersion to advection, and is applicable for confined or partially confined aquifers (Małoszewski, 2000). Its RTD is given by

$$g(\tau) = \frac{1}{\tau\sqrt{4\pi D_{\mathrm{P}}\tau/\tau_{\mathrm{m}}}} e^{-\left(\frac{\left(1 - \tau/\tau_{\mathrm{m}}\right)^2}{4\pi D_{\mathrm{P}}\tau/\tau_{\mathrm{m}}}\right)} \tag{4}$$

The exponential mixing Model (EMM) is given by

$$g(\tau) = \frac{1}{\tau_{\mathrm{m}}} e^{\left(-\tau/\tau_{\mathrm{m}}\right)}, \tag{5}$$

where $\tau_{\mathrm{m}}$ is the mean residence time, $\eta$ is the ratio defined as $\eta = \left(l_{\mathrm{P}} + l_{\mathrm{E}}\right)/l_{\mathrm{E}} = l_{\mathrm{P}}/l_{\mathrm{E}} + 1$, where $l_{\mathrm{E}}$ (or $l_{\mathrm{P}}$) is the length of area at the water table (or not) receiving recharge, $D_{\mathrm{P}}$ is the dispersion parameter, which is the reciprocal of the Peclet number (Pe) and defined as $D_{\mathrm{P}} = D/(vx)$, where $D$ is the dispersion coefficient (m$^2$ day$^{-1}$), $v$ is velocity (m day$^{-1}$), and $x$ is distance (m).

Each RTD has one or two parameters, which the MRT (e.g. $\tau_{\mathrm{m}}$) is determined by convoluting the input (the time series [3]H and CFCs input in rainfall) to each model in a way that matches the output (the measured [3]H and CFC concentrations in groundwater), and other parameters (e.g. $\eta$ and $D_{\mathrm{p}}$) are determined depended on the hydrogeological conditions. To interpret the ages of the Manas River Basin data set, the exponential–piston flow model (EPM, with $\eta$=1.5 and 2.2), the dispersion model (DM, with $D_{\mathrm{p}}$=0.03 and 0.1), and the exponential mixing model (EMM) were used and then the MRTs with different RTDs were cross–referenced."

We read the literature carefully and think that both the method and idea are very innovative by Custodio et al. (2018), who related chloride (Cl) and [14]C activity changes in recharge for aquifers in the arid area to indicate the effect of variations in recharge rate during the previous wetter–than–present period. We think that it would be also good applications in other arid areas around the world because the climatic change is global. In our manuscript, the apparent [14]C ages estimation was adopted. The changes in groundwater reserves and [14]C content will be conducted in the future in the Manas River Basin.

5) I know it is customary to interpret CFC data assuming piston-flow, but it is nonetheless a priori wrong. Model choice must be substantiated from knowledge of the hydrogeological situation and the sampling scheme (Maloszewski and Zuber, 1982; Leray et al., 2016). Later on in the manuscript however different models are used in the binary mixing plots, and model choice is discussed briefly. Why use the "apparent age" concept at all, then ? This is confusing and reads like the two authors have written separately different parts of the manuscript and then pasted the two parts together.

Response: Sorry, we have realized that the apparent CFC ages are not appropriate. As we know that the CFCs are synthetic organic compounds and largely released to the air since 1930s, and thus they have been treated as good tracers for dating young water recharge time (post–1940 recharge).

We totally agree with you that "Model choice must be substantiated from knowledge of the hydrogeological situation and the sampling scheme". Therefore, in the revised manuscript, we have rewritten the content and have discussed the model choice in detail in Section 3.3.3 (See Response 4).

The mean residence times estimated by the lumped–parameter models have been deleted in the binary mixing plots (Fig. 7). However, the binary plots of CFC vs. CFC and CFCs vs. [3]H are widely used in the literatures and can provide useful information on the co–existence of young water with old water (Cook et al., 2017; Han et al., 2007; Han et al., 2012; Koh et al., 2012; Qin et al., 2011), as well as can provide a powerful tool to recognize contamination, degradation and irrigation infiltration and so on (Cook et al., 2017; Han et al., 2015; Mahlknecht et al., 2017; Qin et al., 2012). However, these information mentioned above have not been clarified clearly before in the Manas River Basin, and even not in the Junggar Basin (Fig. 1a). Therefore, we think that the binary mixing plots are essential in the manuscript in order to tell the reader more information about groundwater mixing or recharge features.

6) I also have my doubts concerning the calculations of the mean transit times as they are presented.

Response: Take the CFC–12 and exponential–piston flow model (1.5) for example, the calculation process of the mean residence times is as follows: First, we chose the Eq. (2b) as the convolution integral, and chose the exponential–piston flow model (Eq. (3a, 3b)) as the system response function. Second, we used the time series CFC–12 trend of Northern Hemisphere atmospheric mixing ratio (1940–2014, http://water.usgs.gov/lab/software/air/cure/) as input concentrations. We treated the calendar year 2015 (groundwater sampling time) as age=0 yrs by convoluting the input (times series of CFC–12 input) to the EPM (1.5), and increased the mean residence time $\tau_m$ from 1 to 500 yrs with the time step of 1 yr within 100 yrs and with the time step of 5 yrs between 101 to 500 yrs. Then we can get a sequence of results of output CFC–12 concentrations and mean residence times (vary from 1 to 500 yrs). Third, we plotted the output CFC–12 concentrations vs. mean residence times and then compared the measured groundwater CFC–12 concentrations to get the groundwater mean residence

times. The computational procedures were conducted by using MATLAB (version R2014a). The output CFC–12 concentrations decreased from 526.4 pptv with $\tau_m$ of 1 yr to 3.0 pptv with $\tau_m$ of 155 yrs. As the detection limit for each CFC is about 0.01 pmoL$^{-1}$ of water (be equal to 3.54 pptv with the laboratory temperature of 25°C), the output CFC–12 concentrations lower than 3.54 pptv can be neglected.

7)  The method with which the tritium input has been reconstructed is not documented properly (which stations were used, and how long were the availabe time series?) and the estimated modern value (31 TU) seems extremely high compared to Western Europe for instance (about 6 TU). Is that because of the Chinese nuclear tests of the 60s and 70s that are being referred to in the introduction, or the result of some kind of regional effect?

Response: Agree and changes made. The input tritium ($^3$H) activity in Urumqi station was reconstructed as follows: First, as we know that there is a proportionality between the logarithmic precipitation $^3$H activity and the latitude in the Northern Hemisphere, we can reconstruct the Urumqi precipitation $^3$H activity according to the following equation:

$$\lg C_U = \lg C_H + \frac{\lg C_I - \lg C_H}{X_I - X_H} * \left( X_U - X_H \right) \tag{S1}$$

where $C_U$ is Urumqi precipitation $^3$H activity, $C_H$ is Hongkong precipitation $^3$H activity, $C_I$ is Irkutsk (in Russia) precipitation $^3$H activity, $X_U$ is Urumqi latitude, $X_H$ is Hongkong latitude, and $X_I$ is Irkutsk latitude. The precipitation $^3$H activity are available in the International Atomic Energy Agency (IAEA). The results based on Eq (S1) are show in Table S1.

Table S1. The reconstructed Urumqi precipitation $^3$H activity between 1969 and 1983.

| Date | $C_H$ (TU) | $C_I$ (TU) | $C_U$ (TU) | Ottawa precipitation (TU) |
|------|------|------|------|------|
| 1969 | 46.93 | 454.79 | 248.63 | 253.66 |
| 1970 | 29.38 | 464.15 | 222.82 | 190.77 |
| 1971 | 24.38 | 516.61 | 229.38 | 206.1 |
| 1972 | 27.86 | 247.23 | 138.36 | 92.34 |
| 1973 | 13.22 | 173.34 | 87.44 | 90.41 |
| 1974 | 17.47 | 216.58 | 110.90 | 98.07 |
| 1975 | 12.42 | 190.93 | 92.33 | 75.86 |
| 1976 | 11.71 | 148.8 | 75.69 | 58.91 |
| 1977 | 12.24 | 150.72 | 77.31 | 73.93 |
| 1978 | 10.89 | 103.11 | 56.72 | 73.63 |
| 1979 | 8.96 | 92.06 | 49.55 | 49.62 |
| 1980 | 7.63 | 141.8 | 65.20 | 49.54 |
| 1981 | 11.53 | 138.3 | 71.44 | 55.09 |
| 1982 | 9.73 | 66.92 | 40.08 | 47.29 |
| 1983 | 21.14 | 127.26 | 78.96 | 40 |

Second, a good linear regression between the $C_U$ and Ottawa precipitation $^3$H activity can be obtained as follows:

$$C_{Ui} = 1.0102 C_{Ottawa} + 11.647 \left( R^2 = 0.933, n = 15 \right) \tag{S2}$$

where $C_{Ottawa}$ is Ottawa precipitation $^3H$ activity, $C_{Ui}$ is the Urumqi annual precipitation $^3H$ activity.

Third, considering the precipitation amount effect, the reconstructed Urumqi precipitation $^3H$ activity can be obtained as follows:

$$C_{RU} = C_{Ui} * \frac{P_i}{P_m} \tag{S3}$$

where $C_{RU}$ is the reconstructed Urumqi precipitation $^3H$ activity shown in Fig. 6 (estimated $^3H$), $P_i$ is the annual precipitation amount (1953–2015), $P_m$ is the annual mean precipitation amount in Urumqi between 1953 and 2015.

Yes, the estimated modern precipitation $^3H$ activity are indeed extremely high (mean value of 31.55 TU between 2000 and 2015). We speculate that the very high precipitation $^3H$ activity may be ascribed to both the Chinese atmospheric nuclear tests (around 350 km away to Manas River Basin) from 1964 to 1974 (Zhou, 1992) and the continental effect (Tadros et al., 2014), where the Manas River Basin is more than 3500 km far away from the western pacific. The high precipitation $^3H$ activity was also shown in Fig. S1 (Cauquoin et al., 2015).

[Figure]

Figure S1. Comparison of modelled tritium in precipitation with surface snow samples from two ITASEAntarctic traverses (Proposito et al., 2002; Becagli et al., 2004) and some pre-bomb measurements. (The figure is from Cauquoin et al., 2015, Fig. 10d)

8)    Furthermore, the authors do address the non-uniqueness problems that are bound to arise when calibrating an exponential piston-flow or a dispersion model (2 free parameters each) using a single tritium measurement in aquifers that still retain some of the "bomb tritium" (see Stewart et al., 2010 for details), but in a terribly confusing way and without first explaining the rationale and the approach taken.

Response: Sorry. We should have given more details about the two free parameters each for the exponential–piston flow and the dispersion model. These parameters are substantial based on the geological/hydrogeological conditions. We have added some more detailed explanations concerning the rational and the approach taken in Section 3.3.3 (Response 4). The revised version can be seen in the "Response 4)".

9)  I suppose figure 8 was meant to show the range of parameters that match the measured tracer concentrations. That's commendable, but badly explained.

Response: Yes, thanks. We have redrawn the figure 8 to show the most useful information to the readers (see the "Specific Comments Response 55").

10)  In the final step relating mineralisation to transit time, the authors finally select the EPM calibrated with the CFC12 measurements, but this is once again presented in a unclear fashion.

Response: There are some reasons that we select the EPM (1.5) calibrated with the CFC–12 to relate hydrochemistry evolution: First, groundwater CFCs results show that the urban air with CFC compounds contaminations, which generally cause elevated CFC than the global background atmospheric CFC concentrations (Northern Hemisphere), are unlikely (Fig. 7) in the Manas River Basin. Second, CFC–12 has much less propensity for degradation and/or contamination than CFC–11 and CFC–113. Third, the exponential–piston flow model is one of the mostly commonly used system response functions. The choice of exponential–piston flow model is suitable for the aquifer that has two segments, in which a segment of exponential flow followed by a segment of piston flow. As shown in Fig. 1c and relative hydrogeological discussions, exponential–piston flow model is a good choice to be chose as the system response function. In addition, the mean residence times mathematic relation between EPM (1.5) and other system response functions are obtained (Fig. 11). We can relate mineralization to mean residence time based on these mathematic relations.

11)  The discussion is too long, relies too much on untested and untestable hypotheses, and presents so many singular and unfocused results that it is difficult for the reader to grasp a clear picture of their meaning and significance. The paragraph on "apparent age" should be scraped altogether and the different estimates of "age" (i.e. mean transit time of the respective model) and mixing ratios organised in a clear and synthetic manner.

Response: To avoid misleading, we have deleted some irrelevant content, such as the incorrect statement of "apparent age" for the CFCs, and the incorrect hypothesis of piston flow as mixing is supposed to be "most likely" either within the aquifer or at the sampling point. We also tried to present a clear roadmap to readers, including modern and paleo-meteoric recharge features, mean residence time estimated by LPMs, and mixing features as well as irrigation infiltration indications.

12)  All in all, the manuscript must be seriously reorganised and streamlined. The calculation of mean transit times of the different tracers must be redone, removing entirely the "apparent age" nonsense and explaining clearly the different steps taken by the authors to (I) select a model (II) explore model parameter range and (III) compare the different results obtained from tritium, the CFCs and carbon 14. Interpretation of the obtained "ages" in terms of hydrogeology and its correlation to hydrochemistry must then be presented in a clear and synthetic fashion. Only when this is done might the manuscript rise above an unoriginal and confusing rehash of previous studies, and could be considered for publication.

Response: We reorganized the structure and tried our best to present a clear roadmap to readers. We rewrote the Section 3.3.3 in which system response functions were selected by considering the applicable hydrogeological situations. Parameter and mean residence times were recalculated and then explained in a much clearer way. Figures 8 and 9 (They are Figs. 10 and 11 in the revised manuscript) were redrawn to show much useful information to readers. The controversial "apparent CFC age" was

deleted and then was shown in a more proper way to indicate the modern precipitation recharge.

**Specific Comments**

1) Please ask the help of a proof reader to help improve readability

Response: Agree. We will ask a proof reader to modify the language to help improve readability.

2) L11: Why is it crucial ? Please explain or leave that out.

Response: Agree. As we know that groundwater are the very important and even the only water resources for the local residents in the Manas River Basin of arid northwest China (It is also the same in many other arid areas around the world). Two recharge sources which, modern precipitation recharge and paleo–meteoric recharge, are known as the main sources for the groundwater in arid areas. Knowledge the recharge sources is thus can provide correct guidance for extraction of groundwater. In addition, groundwater renewability is also an important factor for guiding extracting groundwater. Groundwater residence time reveals information about water storage, mixing and transport in subsurface water systems, which has contributions to the rational development of groundwater (Cartwright et al., 2017; Custodio et al., 2018; Dreuzy and Ginn, 2016; Leray et al., 2016). Therefore, we tend to retain the sentence (L11) in consideration of the explanation mentioned above.

3) L15: "indicating the rainfall recharge…" You mean that the young water component is higher than in samples with lower tritium activity.

Response: The inexact expression generates misunderstanding. We mean the very high $^3$H activities (41.1–60 TU) of groundwater indicate rainfall recharge during the nuclear bomb.

4) L29: The title of this section is not very telling, and this is not really what the study is about, is it?

Response: Agree and changes made. The title was changed to "Introduction".

5) L33: "may be renewable". What do you mean ? Something about short turnover time ?

Response: Changes made. We think that "renewable" may means groundwater being recharged by other water resources after extraction (or discharge). The ambiguous sentence was changed to "It is particularly crucial in the alluvium aquifer where the fresh groundwater renewability is generally strong (Huang et al., 2017) and thus as potable water resources in the arid areas, as well as more vulnerable to anthropogenic contaminants and …".

6) L37-39: Rewrite the entire sentence.

Response: Agree and changes made. The entire sentence was changed to (RL35–40): "Since the residence time distributions in the subsurface water systems cannot be empirically measured, a commonly used approach is parametric fitting of trial distributions to chemical concentrations (Leray et al., 2016; Suckow, 2014). The widely used lumped–parameter models (LPMs; Małoszewski and Zuber, 1982; Jurgens et al., 2012), which commonly assume that the hydrologic system is at a steady–state, have been applicable to subsurface water systems containing young water with modern tracers of variable input concentrations (e.g. seasonally variable stable isotope $^2$H and $^{18}$O, tritium, and $^{85}$Kr; Cartwright et al., 2018; McGuire et al., 2005; Morgenstern et al., 2015; Stewart et al., 2010)."

7) L38: "and may be inferred". You mean "must be inferred".

Response: Agree and changes made. Yes, the residence time distribution cannot be empirically measured. See "Specific Comments Response 6".

8) L39: "at" steady-state, not "in" steady-state.

Response: Agree and changes made. "in" was replaced by "at".

9) L40: "Three types of transit time". You mean three time windows ?

Response: Changes made. The sentence was changed to (RL40–41): "The groundwater residence time tracers can be classified into three types possessing the different time scales".

10) L46: It's not variability, rather time span.

Response: Agree and changes made. "variabilities" was replaced by "time span".

11) L48: "in a similar function with" should read "in a similar way to".

Response: Agree and changes made. "in a similar function with" was replaced by "in a similar way to".

12) L51: replace "over" with "than".

Response: Agree and changes made. "over" was replaced by "than".

13) L54: You mean that increasing transit time through the aquifer leads to increasing mineralisation.

Response: No change made. That "increasing transit time through the aquifer leads to increasing mineralisation" is reasonable under certain circumstance. However, groundwater mineralisation is a very complex process, which contains mineral dissolution, precipitation, hydrolysis, evaporation concentration, water–rock interaction, and so on (Ma et al., 2018). In addition, water corrosion, solubility of rocks, and groundwater runoff will also have strong impacts on mineralization (Shen et al., 1993). Thus, increasing transit time not necessarily leads to increasing mineralization. The sentence and cited literatures (L54) in the manuscript tend to show that there are some correlations between the transit time and mineralsation in some situations. Nevertheless, the correlation is still unclear in the Manas Rriver Basin, and that is right the research objective in our manuscript.

14) L55: Please explain why tritium is "the only true age tracer", namely because it is part of the water molecule.

Response: Agree and changes made. Yes, it is. The sentence was changed to (RL56): "Tritium ($^3$H), a component of the water molecule with a half–life of 12.32 yrs, is the only true age tracer for waters (Tadros et al., 2014)".

15) L56 (entire paragraph): Why mention the southern hemisphere at all, since the study takes place in the northern hemisphere ? This is useless information.

Response: Agree and changes made. We have deleted the statements concerning the Southern Hemisphere $^3$H activities and relevant references (L61–66).

16) L66: "may be used to estimate MTTs" should read "must be used to estimate MTTs". And explain why (non-unicity problems…).

Response: Agree and changes made. Yes, we totally agree with your opinion. As the mean annual $^3$H activities peaks were several hundred times natural levels in the Northern Hemisphere due to the atmospheric thermonuclear tests in the Northern Hemisphere between the 1950s and 1960s, the nowadays rainfall $^3$H activities are still affected by the tail–end of the bomb–pulse in the Northern Hemisphere. Groundwater residence time cannot be assigned based on measurement of a single sample $^3$H activity.

The sentence was changed to (RL62–65): "Thus, measurement of a single sample $^3$H activity does not allow the groundwater residence time to be assigned (Cook et al., 2017), and time series $^3$H measurements must be used to estimate MRTs in the Northern Hemisphere by LPMs (Han et al., 2007; Han et al., 2015)."

17) L69: You are confusing residence time and degradation half-life. The residence time of the CFCs in the atmosphere is no different from that or tritium or any other tracer. The difference lies in their half-lives (degradation for CFCs, decay for tritium), which are very long for the CFCs.

Response: Agree and changes made. We agree with your opinion on the "residence time" and "degradation half–lives" for the atmospheric environmental tracers (e.g. CFCs and $^3$H). CFCs degrade slowly in the atmosphere and have relatively long degradation half–lives, while the decay half–life for $^3$H is much shorter with 12.32 yrs. The input function for CFCs is not area–specific as is the case with $^3$H. Therefore, the atmospheric concentrations for CFCs are uniform over large areas (Cartwright et al., 2017; Cook et al., 2017), while that for $^3$H often varies across latitudes and between seasons (Tadros et al., 2014).

The sentence was changed to (RL66–67): "Contrasting to $^3$H, CFCs degrade slowly in the atmosphere and have relatively long degradation half–lives, which permits their uniform atmospheric distributions over large areas, …".

18) L78-82: Please rephrase the entire sentence.

Response: Agree and changes made. The sentence was rewritten as follows (RL76–81): "Furthermore, inaccurate groundwater MRTs might be obtained when the CFCs entrapped excess air or contaminated in urban environments due to the CFCs degradation and/or contamination. For example, MRTs are much less than actual values if CFCs inputs are entrapped excess air in the unsaturated zone during recharge (Cook et al., 2006; Darling et al., 2012) or contaminated in urban and industrial environments (Carlson et al., 2011; Han et al., 2007; Mahlknecht et al., 2017; Qin et al., 2007), and are much higher if CFCs inputs are degraded in anaerobic groundwater (most notably CFC–11 and CFC–113; Cook and Solomon, 1995; Horneman et al., 2008; Plummer et al., 2006b)."

19) L89: "Mixing […] is particularly true…". You don't know that, it's a probable hypothesis!

Response: Agree and changes made. Though the previous study (Ma et al., 2018) have confirmed that the groundwater lateral–flow mixing is common in the alluvium aquifers of the Manas River Basin, more work and studies should be conducted to confirm the mixtures within the aquifers and long–screened wells. On the other hand, groundwater mixing can occur within the aquifer itself (Cook et al., 2017) as well as occur during pumping from long screened wells. However, for the case study of the Manas River Basin, mixing ascribed to pumping from long–screened wells has not been demonstrated before.

The sentence was changed to (RL88–89): "The hypothesis mixing within the aquifers and pumping from the long–screened wells is particularly common in the faulted–hydraulic drop alluvium aquifers of the Manas River Basin in the arid northwest China (Fig. 1)."

20) L93-95: "The MTTs that impacted….". This sentence makes no sense. Rewrite.

Response: Agree and changes made. The sentence was changed to (RL91–93): "The MRTs that result from a deep unsaturated zone (water table depth is 180 m) and contrasting geological settings (a level difference of 130 m hydraulic drop caused by the thrust fault) are still insufficiently recognized in the alluvium aquifer (Fig. 1c)."

21) L106: "with totally length" should read "with total length".

Response: Agree and changes made. "totally" was replaced by "total".

22) L107: "was intermittent activity" should read "was intermittently active".

Response: Agree and changes made. "intermittent activity" was replaced by "intermittently active".

23) L110: So the different aquifers are all fractured rock aquifers.

Response: Yes, the substantial fractures and fissures of rock aquifers in the mountain area allow groundwater migration to be possible. The literatures by Cui et al. (2007) and Zhou (1992) were cited in the relevant discussion.

24) L114: "is macroscopically similar". What do you mean with "macroscopically" ?

Response: Changes made. The sentence was changed to (RL113): "The groundwater flow direction is consistent with the Manas River flow direction."

25) L120: "is as large" should read "is as deep".

Response: Agree and changes made. "large" was replaced by "deep".

26) L124: How many samples were taken altogether ? And are there any information concerning screening depth and size (fully penetrating wells or not) ? This is important information to guide model choice.

Response: Agree and changes made. The important information concerning sample numbers as well as screening depth and size were added in the revised manuscript.

The sentences were changed to (RL123–127):

"In total, 29 groundwater (pumped from fully penetrating well, of which 3 are from spring and 3 are from the artesian well) were collected along the Manas River motion during June to August, 2015 (from G1 to G29 in Table 1 and Fig. 2). Groundwater were separated into three clusters including the upstream groundwater (UG, south of the Wuyi Road), midstream groundwater (MG, area between the Wuyi Road and the West main canal–Yisiqi), and downstream groundwater (north of the West main canal–Yisiqi) based on the hydrochemistry and stable isotope data."

27) L126: What was the rationale for separating the samples into three groups ? For instance, why is G13 MG while G26 is DG ? DG seems like the downgradient boundary. Did you use the piper diagrams to separate the samples ?

Response: Agree and changes made. Yes, you are exactly right that the three groundwater groups are based on the result of hydrochemistry and stable isotope. Yisiqi (Fig. 2) is the dividing line where there is an obvious change for the hydrochemistry and stable isotope along the Manas River motion (Fig. S2 from Ma et al., 2018). We also use piper diagrams to separate the samples (Fig. 4).

[Figure]

Figure S2. Relationship between $\delta^{18}O$ and Cl concentrations for waters as a means to differentiate hydrogeological processes in the Manas River Basin. The purple circles represent the upstream and midstream groundwater, and green, red, and black legends represent downstream groundwater (from Ma et al., 2018).

28) L152: "were followed" is used multiple times but should read "after" or "following".

Response: Agree and changes made. We have checked the manuscript carefully and seriously have corrected the erroneous words. "were followed" was changed to "after" and we insisted on the "after" throughout the manuscript.

29) L173: "refers" is not the proper verb. Use "depends" for instance.

Response: Agree and changes made. The sentence was changed to (RL174–175): "The difference between the local and global background atmospheric CFC concentrations (Northern Hemisphere), which we intitule as CFC excess, varies largely based on the industrial development."

30) L179: What are low latitude countries ?

Response: Changes made. The vague expression was deleted. As the atmospheric CFC concentrations are affected by the industrial development, the time series of Northern Hemisphere atmospheric CFC concentrations were widely treated as the background values for the underdeveloped areas (it is also applied to the Manas River Basin in our manuscript).

31) L189: The entire procedure is correctly explained, and also the fact that "apparent age" implies piston-flow transit time distribution, but why use apparent age in the first place ? Piston-flow is one model among many, as the authors explain later in the manuscript. Furthermore, the entire concept of "age" is problematic and should be replaced by mean transit time or mean residence time (for an in-depth discussion, see Suckow, The age of groundwater-Definitions, models and why we do not need this term, Applied Geochemistry 50, 2014, 222-230).

Response: Agree and changes made. The erroneous "apparent CFC age" has been deleted. The concept of "age" is replaced by mean residence times and estimated by lumped–parameter models with the exponential–piston flow model, dispersion model, and exponential model, which are discussed in Section 4.3.1. In addition, the groundwater recharge sources (e.g. to distinguish the modern and paleo–meteoric recharge features) identified by the CFCs are also discussed in Section 3.3.1 and Section 4.2.2.

32) L194: What do you mean by "closed system" ? Physically bounded ?

Response: Changes made. As we know that the measured tracer output concentration in the groundwater is compared to its historical input using the convolution integral, in which the system response function is selected based on the different hydrogeological subsurface flow systems (Jurgens, et al., 2012; Małoszewski and Zuber, 1982; the details of calculation is shown in "Response 6"). The tracer output concentration can be measured from the spring, stream, well, and so on. There is no absolutely physical boundary for the subsurface flow system. Therefore, the ambiguous statements "closed system" is not rigorous and thus is deleted. See Section 3.3.3 (Response 4).

33) L208: "were given below as transit time distribution function" should read "were selected and are given below".

Response: Agree and changes made. This sentence was deleted and more detailed interpretations were added (See Section 3.3.3 in the Response 4).

34) L219: You should also explain here how you planned to choose between these competing models.

Response: Agree and changes made. The system response function is selected based on the hydrogeological settings and the sampling position of tracer output concentration measured. We explain the basis on the choice of the models. See Section 3.3.3 (Response 4).

35) L235: Why present an equation you will not be using for lack of appropriate data ?

Response: Agree and changes made. We have deleted this equation and the accompanying description.

36) L240: This is true for the piston-flow model only ! See Custodio et al. for details.

37) L250: So the entire paragraph boils down to using literature values for the initial 14C activity. Make it shorter and to the point.

Response: Agree and changes made. Custodio et al. (2018) indicated that the calculated groundwater $^{14}$C age is really an apparent age due to the mixture of waters with different transit times, which was also indicated by other researchers (e.g. Cook et al., 2017; Suckow, 2014). Therefore, we deleted some unnecessary description and unimportant information in the text. Some sentences were reduced and revised. The revised content can be seen in Section 3.3.2 (RL193–218):

"Carbon–14 ($^{14}$C, with a half–life of 5730 yrs) activity in groundwater is often used to estimate groundwater ages over time periods of approximately 200 and 30 000 yrs, and to determine the recharge from mixing waters in various climate conditions (Cook, et al., 2017; Custodio et al., 2018; Huang et al., 2017). The calculation of $^{14}$C ages may be complicated if groundwater dissolved inorganic carbon (DIC) is derived from a mixture of sources or/and the [14]C originating from the atmosphere or soil zone is often significantly diluted by the dissolution of [14]C–free carbonate minerals in the aquifer matrix and biochemical reactions along the groundwater flow paths (Clark and Fritz, 1997). While only minor carbonate dissolution is likely, determination of groundwater residence times requires [14]C correction to be taken into account (Atkinson et al., 2014). When dissolution of carbonate during recharge or along the groundwater flow path may dilute the initial soil $CO_2$, $\delta^{13}C$ can be used to trace the process (Clark and Fritz, 1997). An equation for the reaction between carbon–dioxide–containing water with a carbonate mineral is commonly written as (modified after Pearson and Hanshaw, 1970):

$$CO_2 + H_2O + CaCO_3(\delta^{13}C_{carb} = 0) \rightarrow Ca^{2+} + 2HCO_3^-(\delta^{13}C_{DIC}) , \tag{R1}$$

where $\delta^{13}C_{carb}$ is the dissolved carbonate $\delta^{13}C$ value (approximately 0; Clark and Fritz, 1997), and $\delta^{13}C_{DIC}$ is the measured $\delta^{13}C$ value in groundwater.

Depending on knowing the measured [14]C activity after adjustment for the geochemical and physical dilution processes in the aquifer (without radioactive decay), then the groundwater apparent [14]C ages ($t$) can be calculated from the following decay equation:

$$t = -\frac{1}{\lambda_{^{14}C}} \times \ln\frac{a^{14}C}{a_0{}^{14}C} , \tag{1}$$

where $\lambda_{^{14}C}$ is the [14]C decay constant ($\lambda_{^{14}C} = \ln 2/5730$), and $a^{14}C$ is the measured [14]C activity of the

DIC in groundwater. As mentioned above, the estimated ages are really apparent ages due to the mixture of waters with wide range of ages (Custodio et al., 2018; Suckow, 2014).

Previous studies in the arid northwest China (Edmunds et al., 2006; Huang et al., 2017) have concluded that a volumetric value of 20 % "dead" carbon derived from the aquifer matrix was recognized, which is consistent with the value (10–25 %) obtained by Vogel (1970). Therefore, the initial [14]C activity ($a_0{}^{14}C$) of 80 pMC is used to correct groundwater [14]C ages (results are shown in Table 1), despite this simple correction makes no attempt to correct the age of individual samples that may have experienced different water–rock interaction histories."

38) L264: Check the discussion paper by Benettin et al. in review in HESS for the latest developments on the "evaporation slope".

Response: Agree and changes made. We agree with the opinion on the evaporation slope that a steeper slope would be obtained due to the large source variability under different meteorological conditions in different seasons indicated by Benettin et al. (2018). As we know that water fractionation is affected by various factors, like the water surface temperature, air humidity, wind speed, and so on, but we can always obtain a linear trend from a source water with low salinity (Ma et al., 2015). However, in practice, the evaporation trend line is often obtained from various sources water, which is often not true evaporation line. In our manuscript, the evaporation line is estimated according to the surface water (ditch water, reservoir water, and Manas River water), which are all collected in summer of 2015. The ditch, reservoir and Manas River are always connected to one another (Fig. 1b), and all are recharged from the mountain areas in the same season. Though there are minor differences of the water sources for the surface water, the linear trend obtained based on these surface water may have implications for the surface water evaporation. Therefore, we tend to use the evaporation slope and to add some statements (as follows) for the rationality in the revised manuscript (RL286–290):

"A recent study (Benettin et al., 2018) has indicated that the evaporation line that obtained from various sources water is often not the true evaporation line. Our surface water were all collected in summer of

2015 and were recharged from the mountain areas in the same season, though they were collected from the different point (ditch water, reservoir water and Manas River water), the linear trend obtained based on these surface water may have implications for surface water evaporation."

39) L274: The entire paragraph is too short and should explain clearly the approach adopted to calculate "ages" from the tracer data (model and model parameter choices !). I strongly advise against using binary mixing diagrams, and encourage the authors to use a multi-tracer modelling approach trying to find a single optimum or optimal parameter regions for the different tracers.

Response: Changes made. The entire paragraph was deleted. We reorganized the structure of the Section "4. Results and discussion", rewrote some contents and deleted some incorrect expressions, and added statements concerning the model choice in Section "3. Materials and methods". However, we tend to retain the binary mixing diagrams (the mean residence times with different models have been deleted in the diagrams) to explain the young and old groundwater mixing features and to identify the impact of human activity on the groundwater.

40) L277: The paragraph on "apparent age" makes no sense for the reasons given above. I disagree with the proposition that "they [apparent ages] provide a good first approximation for groundwater age". There is no reason to prefer the piston-flow model which is implied by the "apparent age" concept over other models. This argument has been for years a lazy way to skip responsibility in choosing one model based on knowledge of the hydrogeological situation and sampling.

Response: Agree and changes made. The "apparent CFC ages" has been deleted. In addition, we tend to discuss the modern precipitation recharge features and to distinguish the modern and paleo–meteoric recharge using CFCs in Section 4.2.2 (RL344–414):

"It is seen from Table 1 that groundwater with well depths between 13 and 150 m contain detectable CFC concentrations (0.17–3.77 pmol $L^{-1}$ for CFC–11, 0.19–2.18 pmol $L^{-1}$ for CFC–12, and 0.02–0.38 pmol $L^{-1}$ for CFC–113) both in the upstream and midstream areas, indicating at least a small fraction of young groundwater components (post–1940). The highest concentration was observed in the UG (G3), south of the fault, median and the lowest were respectively observed in the west and east bank of the 'East main canal' in the MG, north of the fault. In the midstream area (Fig. 2), CFC concentrations generally decrease with well depth at the south of reservoirs (G25, G8, and G9), while increase with well depth at the north of reservoirs (G15 and G16), which might indicate the different groundwater flow paths (e.g., downward or upward flow directions).

Groundwater aerobic environment (Table 1, DO values vary from 0.7 to 9.8 mg $L^{-1}$) make CFC degradation under anoxic conditions unlikely. Nevertheless, CFC–11 has shown a greater propensity for degradation and/or contamination than CFC–12 (Plummer et al., 2006b). Therefore, we use the CFC–12 to interpret the modern groundwater recharge in the following discussions. The estimated CFC atmospheric partial pressures and possible recharge year are shown in Table 2 and Fig. 3. The UG (G3) CFC–113 and CFC–12 both indicate the 1990 precipitation recharge (Table 2), probably indicating piston flow recharge in the upstream area. The MG CFC–11–based modern precipitation recharge agreed within 2–8 yrs with that based on CFC–12 concentrations, while that the CFC–113–based recharge were much 4–11 yrs later than that based on CFC–11 and CFC–12 concentrations, indicating mixtures of young and old groundwater components recharge in the midstream area. The latest groundwater recharge is in the upstream area (G3 with 1990 recharge), which is most likely due to the shortest flow paths from recharge sources compared to the piedmont groundwater samples in

the midstream area.

[revised manuscript text omitted]

41) L297: "which confirms". A performative statement confirms nothing. You are supposing this is the case !

Response: Agree and changes made. We have revised this sentence: "…, which makes one to assume that …"

42) L317: Shortly explain the method used to estimate the tritium input (linear regression ? And how long were the time series used ?). The reference to Han et al. is not very useful as the authors of that paper themselves refer to an IAEA publication without further explanations.

Response: Agree and changes made. The explanation is seen in "Response 7". We added short statements in the revised manuscript as follows (RL236–239):

"The historical precipitation $^3$H activity in Urumqi station (Fig. 4) was reconstructed from the available data in the International Atomic Energy Agency (IAEA) using a logarithmic interpolation method. Precipitation $^3$H activities between 1969 and 1983 at Hongkong and Irkutsk with different latitudes were used (data is available at <https://www.iaea.org/>)."

We think that the reconstructed tritium ($^3$H) activity in Urumqi station can be added as the supplement if necessary.

[Figure]

Figure 6. Tritium concentration (TU) of groundwater water samples of upstream groundwater (UG), midstream groundwater (MG), and downstream groundwater (DG). Time series of tritium concentration in precipitation at Ottawa, Urumqi, Hongkong, and Irkutsk were obtained by GNIP in IAEA (https://www.iaea.org/). The blue dashed lines and shaded field were drawn using the half–life (12.32 yrs) of tritium decayed to 2014. (It is Fig. 4 in the revised manuscript)

43) L318: A background of 31 TU is very high compared to Western Europe (about 6 TU). How come ?

Response: As shown in Fig. 4 and reconstructed tritium $^3$H activity in Urumqi, the estimated modern precipitation $^3$H activity are indeed extremely high (mean value of 31.55 TU between 2000 and 2015). We speculate that the very high precipitation $^3$H activity may be ascribed to both the Chinese atmospheric nuclear tests (around 350 km away to Manas River Basin) from 1964 to 1974 (Zhou, 1992) and the continental effect (Tadros et al., 2014), where the Manas River Basin is more than 3500 km far away from the western pacific.

44) L413: What do you mean by "serious" ?

45) L414: "tend toward more discrete with their increase". I do not understand this part of the sentence.

Response: Changes made. The spelling error "serious" was changed to "series". The sentence was changed to (RL479–480): "Figure 11 shows that different LPMs yield different MRTs for the same time series $^3$H activities and CFC concentrations. The MRTs obtained from different LPMs tend to more discrete among each model with the increase of MRTs."

46) L448: The paragraph on hydrochemistry is not bad, but underdeveloped and bad organized. State again what you're looking for first. A good correlation between hydrochemistry and "ages" calculated using some of the TTD models might be a way to constrain or guide model choice, but the authors do not really state that explicitly, although that would be interesting and relatively new.

Response: Yes, we agree with your opinion on the relationship between hydrochemistry and "ages"

calculated using the TTD models. Thanks very much, the suggestion of using hydrochemistry to guide model choice is definitely interesting and relatively new. We will bear in mind in the future studies in the Manas River Basin. But we think that we tend to seek the relationship between groundwater hydro–chemistry and mean residence times firstly in this manuscript.

47) L491: The entire chapter 4.5.1 makes no sense. You must first decide which model is the most appropriate, and then calculate metrics such as mean transit time, young water fraction, etc… You cannot both calculate water fractions using a binary mixing strategy (assuming piston-flow) AND later use an EPM. The same remark applied to chapter 4.5.2.

Response: We agree with your opinions that "first decide which model is the most appropriate, and then calculate metrics such as mean transit time, young water fraction, etc…". We think that the application of mixing models (even combining two lumped–parameter models) is a good method to quantitatively analysis groundwater mixing ratios, and this method is also used more and more widespread. However, CFCs are also good tracers to distinguish groundwater mixing from the different recharge sources, like to recognize modern and paleo–meteoric recharge features, to distinguish the fraction of young groundwater. We have totally revised the entire chapter 4.5.1 and reorganized the structure of the manuscript (See "Response 3"). The erroneous phrases and expressions including apparent CFC age and groundwater ages estimated from the piston flow model have been deleted.

48) L498: "no post-1988.5". Please round this off…

Response: Agree and changes made. "no post–1988.5" was changed to "no post–1989". "1989.5" was changed to "1990".

49) L509: Why do you treat "apparent age" as some kind of different measure of transit time than MTTs "estimated from the EPM" ? This is doing the analysis the wrong way around. First find a way to select a model, then discuss the obtained "ages" instead of hypothesizing on tons and tons of different "ages" that are meaningless because they were obtained disregarding the actual situation. This leads nowhere.

Response: Agree and changes made. The erroneous sentence and term "apparent CFC ages" been deleted. We reorganized the structure and revised the relative content to tried to present a clear roadmap to readers.

50) L541: Before engaging in complicated mixing scenarios, you should first try to find one model and one parameterisation that fits both the CFCs, tritium and carbon 14. Only if that search does not succeed should additional mixing be introduced. Please note that the binary mixture approach by Plummer et al. is only one way of doing so, and a particularly weak one at that because it assumes per default a piston-flow distribution of transit time of each component (other models can be integrated, but it becomes quickly very cumbersome). Another way to include the mixing of different reservoirs is to combine models (say two exponential models, each representing one distinct source) following Piotr Maloszewski and coworkers or Mike Stewart and Uwe Morgenstern. Binary plots such as those of figure 11 suffer from the limitation that you have to recalculate the mixing line for each parameterisation of each model, and they cannot really replace a multi-tracer lumped-parameter modelling approach, where the objective function reduces simultaneously the prediction error of all tracers.

Response: Thanks very much for the precious suggestions you offered. We believe that the modelling

approaches are commendable to interpret complicated mixing scenarios quantitatively. For example, the binary parallel lumped–parameter model (Morgenstern et al., 2015; Stewart et al., 2017), the binary mixing model that followed by two response function models (Jurgens et al., 2012), and the mathematical solutions that indicating the changes in water reserve in the relatively large systems with wide range of residence times (Custodio et al., 2018; Custodio and Custodio–Ayala, 2014). In addition, tracer–tracer binary plots were also the widely used method to determine the groundwater recharge mechanisms and to interpret the groundwater mixing (Cook et al., 2017; Darling et al., 2012; Han et al., 2015; Kagabu et al., 2017). We think that combining $^3$H and CFCs is a good tool to distinguish the modern precipitation recharge and to indicate the groundwater mixing properties (for example mixing old with young water, mixing irrigation infiltration water and young water) in the Manas River Basin, which has not been reported before. Recharge features (e.g. modern and paleo–meteoric recharge features, and the fractions of modern recharge water) are also the essential contents in the manuscript, which will not be all on account of the mean residence times estimated by $^3$H and CFCs in the revised manuscript. Further investigation work will be carried out in the Manas River Basin in the following several years (for example, time serious groundwater CFCs from some given wells will be measured). Deeper analysis concerning groundwater mixing obtained by the combination of two models using lumped–parameter models (binary mixing model) and to explore the changes in water reserve will be good choices in the next studies in the Manas River Basin.

51) L562: Solutions are obtained, explanations are devised.

Response: No change made.

52) L572: What are mixing rates ? You mean mixing ratios ?

Response: Agree and changes made. "rates" was changed to "ratios"

53) L575: "The thrust faults were found to play a paramount role on groundwater flow path". There are not conclusions, but hypotheses very weakly suggested by the analysis of the environmental tracers, which is itself very shaky. I hardly call that evidence. Please refrain from drawing conclusions if the data necessary to test hypotheses is not available (as is the case here).

Response: Agree and changes made. This sentence has been deleted. To make the conclusions to be more clear and well-founded, we have revised the conclusions and delete some incorrect statements. Yes, this conclusion is important but not supported by strong supporting evidences in the paper. Indeed, there are some results that show large differences on both sides of the thrust fault. For examples, there is a level difference of 130 m hydraulic drop (Fig. 1c) in the south margin in Shihezi (SHZ), $^3$H activities of groundwater decrease rapidly along the Manas River motion in the north of the fault but show relatively the highest values in the south of the fault (Fig. 8). These results still can not support the conclusion explicitly "The thrust fault were found to play a paramount role on groundwater flow paths …".

The revised conclusions are as follows (RL554–570):

"In this study, the environmental tracers and hydrochemistry have enabled us to identify the modern and paleo–meteoric recharge sources, to constrain the different end–members mixing rates, and to study the mixed groundwater mean residence times in faulted–hydraulic drop alluvium aquifer systems. The paleo–meteoric recharge in a cooler climate rather than the lateral flow from the higher elevation

precipitation in the Manas River downstream area was distinguished. The quite 'modern' groundwater with young (post–1940) water fractions of 87–100 % was obtained, indicating small mixing degree in the south of the fault. The short mean residence times (19 yrs) along with the higher $NO_3^-$ concentration (7.86 mg $L^{-1}$) than natural groundwater (5 mg $L^{-1}$) in the south of the fault (headwater area) imply invasion of modern contaminants, which should arouse people's attention. Large amplitudes of mixing rate varying from 12 to 91 % were widespread in the north of the fault due to the varying depth of long–screened boreholes or within the aquifer itself. Furthermore, the large water table fluctuations during groundwater pumping, vertical recharge through the thick unsaturated zone, and young water mixtures in different decades highlight the mixing diversity. The obtained strong correlations between groundwater mean residence times and hydrochemistry concentrations allow the first–order proxy at different times to be made. In addition, our study has also highlighted that mean residence times estimated by CFCs rather than $^3H$ were more appropriate in the arid Manas River Basin with thick unsaturated zone."

54) L585: "due to the highly complex groundwater system...". This is no explanation at all ! Indeed, devising a conceptual model that could explain why CFC derived "ages" correlate well with mineralisation while tritium derived "ages" do not could be a useful task (but you should first redo the calculation of the "ages" as suggested above). On the one hand, the correlation between CFC12 and hydrochemistry might be an artifact, given that the area sampled is so large and hydrogeologically diverse. On the other hand, there might be some kind of systematic shift between tritium and CFC ages if differences are due to the unsaturated zone. Maybe a diffusion model using the unsaturated zone thickness might be useful. Still much work to do…

Response: The sentence has been deleted. Thanks very much for the useful suggestions. Previous studies (Ji, 2016; Wang, 2007; Zhou, 1992) have shown that the groundwater flow paths were very complicated from the mountain to the plain areas. Precipitation recharge in the ground in the mountain areas, one part groundwater discharge into spring in the south of the intermountain depression, one part groundwater discharge into stream in the mountain areas and then recharge groundwater in the intermountain depression, and one part groundwater flow in the ground through the intermountain depression. The complicated groundwater flow systems make devising a conceptual model very difficult to implement. We think that more and much detailed work should be conducted from the mountain to intermountain depression areas to find out more evidences that interpretation the conceptual flow model. In addition, the thick unsaturated zone mainly distribute in the intermountain depression and piedmont plain areas (Fig. 1c). The interpretation of CFCs in Section 4.2.2 make us to assume that the unsaturated zone air CFC closely follows that of the atmosphere and thus the recharge time lag through the unsaturated zone is not consideration. In some cases the unsaturated zone can be ignored to obtain workable solutions (Custodio et al., 2018) though the unsaturated zone is not so thick with the Manas River Basin. We think that a diffusion model using the unsaturated zone thickness is a good guide for the further studies in the arid Manas River Basin as well as in other arid basins around the world.

55) Figure 7, 8 and 9: The figures are incredibly cluttered and very difficult to read, especially figure 9 (not to mention the legend).

Response: Agree and changes made. Figures 7, 8 and 9 were redrawn as follows:

[Figure]

**Figure 7.** Distributions of $^3$H and $^{14}$C activities with distance to mountain. The shaded regions indicate the upstream, midstream and downstream of Manas River. (It is Fig. 8 in the revised manuscript)

[Figure]

**Figure 8.** Tritium and CFCs (CFC–11, CFC–12 and CFC–113) output vs. mean residence times for different lumped–parameter models estimated using Eqs. (2) to (5). The input $^3$H activity and CFCs concentration are using the estimated $^3$H activities in precipitation in Urumqi station (Fig. 4) and Northern Hemisphere atmospheric mixing ratio (Fig. 3), respectively. (It is Fig. 10 in the revised manuscript)

[Figure]

**Figure 9.** (**a**) MRTs with EPM (1.5) of CFC–12 vs. CFC–11 & CFC–113, (**b**) CFC–12 MRTs with EPM (1.5) vs. EPM (2.2), DM & EMM, and (**c**) 3H MRTs with EPM (1.5) vs. EPM (2.2), DM & EMM. (It is Fig. 11 in the revised manuscript)

56) Figure 10: Why are there so few points on each graph, since you sampled at 29 locations according to table 1 ?

Response: Figure 10 shows that the x-axis represents the CFC–12 mean residence times with EPM (1.5), and Table 1 shows that there are 9 groundwater CFC–12 samples. For the sample G15, the hydrochemistry show abnormal data with much higher concentrations of $SO_4^{2-}$, $HCO_3^-$, and electrical conductivity than that of perimeter zone. Moreover, in the plot of CFC–12 mean residence times vs. hydrochemistry, G15 sample distributes abnormal. Therefore, only 8 groundwater samples are potted in Fig. 10 (It is Fig. 12 in the revised manuscript).

57) Figure 11: As I wrote above, binary mixing diagrams rapidly tend to show their limits. After two or three mixing lines for different models are drawn, reading becomes nigh impossible. Importantly, error bars are missing for the CFCs and for tritium. I suspect that with error bars, selecting a model visually will become impossible (the lack of sensitivity is another limitation of binary mixing diagrams, ).

Response: Agree and changes made. We deleted the mean residence times with different models but retain the measured CFC concentrations and 3H activities of groundwater in the diagrams.

[Figure]

**Figure 11.** Plots showing relationships of (**a**) CFC–113 vs. CFC–12 and (**b**) CFC–11 vs. CFC–12 in pptv for Northern Hemisphere air. The '+' denotes selected calendar years. The solid lines correspond to the piston flow (PF) and the short–dashed lines show the binary mixing (BM). The shaded regions in (**a**) indicate no post–1989 waters mixing. (It is Fig. 7 in the revised manuscript)

---

## Author Response (AR1)

"Groundwater mean transit times, mixing and recharge in faulted–hydraulic drop alluvium aquifers using chlorofluorocarbons (CFCs) and tritium isotope ($^3$H)" by Ma, B., Jin, M., Liang, X., Li, J., Hydrol. Earth Syst. Sci., doi:10.5194/hess-2018-143.

We appreciate the many valuable suggestions and helpful comments of **Anonymous Referee #1**. We have seriously considered all of the suggestions and comments and have attempted to address each of the comments point-by-point. Detail explanations are as follows.

Author's response – Line numbers referring to the old and revised version manuscripts are preceded by L and RL, respectively.

**Anonymous Referee #1**

**General Comments**

The paper reports CFC, tritium, carbon-14 and stable isotope measurements for groundwater in the Manas River Basin in China and uses them to estimate mean transit times for the complex mixtures of groundwaters in the area resulting from the complicated geology.

The complications of the subject combined with English that is not quite right make this a difficult read. However, the paper addresses relevant scientific questions suitable for publication in HESS, with novel concepts and ideas. Substantial conclusions are reached.

The methods are valid and described satisfactorily, and title and references are well done. There is a problem with the abstract (see below) and consequently the overall structure needs improvement. Some of the figures are complex and could be explained better.

Response: We would like to thank you very much for taking the time to review our manuscript and for your generally positive feedback. We have asked Chris Law (Wallace Academic Editing) to modify the language to help improve readability. We have reorganized the structure and tried our best to present a clear roadmap to readers. We also agree with you that some of the figures are complex which have also been pointed out by Ref #2. Some figures have been redrawn.

The outline of the manuscript have been reorganized as follows:

Title: Application of environmental tracers for investigation of groundwater mean residence time and aquifer recharge in faulted–hydraulic drop alluvium aquifers

1. Introduction
2. Geological and hydrogeological setting
3. Materials and methods
    3.1 Water sampling
    3.2 Analytical techniques
    3.3 Groundwater dating
        3.3.1 CFCs indicating modern water recharge
        3.3.2 The apparent $^{14}$C ages
        3.3.3 Groundwater mean residence time estimation
4. Results and discussion

Figures 6, 8 and 9 have been redrawn as follows:

[Figure]

Figure 6. Tritium concentration (TU) of groundwater water samples of upstream groundwater (UG), midstream groundwater (MG), and downstream groundwater (DG). Time series of tritium concentration in precipitation at Ottawa, Urumqi, Hong Kong, and Irkutsk were obtained by GNIP in IAEA (https://www.iaea.org/). The blue solid lines and shaded field were drawn using the half–life (12.32 yrs) of tritium decayed to 2014. (It is Fig. 4 in the revised manuscript)

[Figure]

**Figure 8.** Tritium and CFCs (CFC–11, CFC–12 and CFC–113) output vs. mean residence times for different lumped–parameter models estimated using Eqs. (2) to (5). The input $^3$H activity and CFCs concentration are using the estimated $^3$H activities in precipitation in Urumqi station (Fig. 4) and the Northern Hemisphere atmospheric mixing ratio (Fig. 3), respectively. (It is Fig. 10 in the revised manuscript)

[Figure]

**Figure 9.** (**a**) MRTs with EPM (1.5) of CFC–12 vs. CFC–11 & CFC–113, (**b**) CFC–12 MRTs with EPM

(1.5) vs. EPM (2.2), DM & EMM, and (**c**) 3H MRTs with EPM (1.5) vs. EPM (2.2), DM & EMM. (It is Fig. 11 in the revised manuscript)

**Specific Comments**

1)    A major problem is that there appears to be a disconnect between the abstract/conclusions and the rest of the paper. The following sentence from the abstract/conclusions:

"The thrust faults were found to play a paramount role on groundwater flow paths and MTTs due to their block water features, where the relatively long MTTs were found near the Manas City with shorter distance and smaller hydraulic gradients."

is not supported by any discussion in the paper. Yes, it may be supported by implication from the results, but such support needs to be made explicit (possibly in its own subsection since this is an important conclusion).

Response: This sentence has been deleted. To make the abstract and conclusions to be more clear and well-founded, we have revised the abstract/conclusions and delete some incorrect statements. Yes, this conclusion is important in the paper. Indeed, there are some results that show large differences on both sides of the thrust fault. For example, there is a level difference of 130 m hydraulic drop (Fig. 1c) in the south margin in Shihezi (SHZ), $^3$H activities of groundwater decrease rapidly along the Manas River motion in the north of the fault but show relatively the highest values in the south of the fault (Fig. 8). These results still cannot support the conclusion explicitly "The thrust fault were found to play a paramount role on groundwater flow paths …".

The revised abstract is as follows (RL12–26):

"Documenting groundwater residence time and the recharge source is crucial for water resource management in the alluvium aquifers of arid basins. Environmental tracers (CFCs, $^3$H, $^{14}$C, $\delta^2$H, $\delta^{18}$O) and groundwater hydrochemical components are used for assessing groundwater mean residence times (MRTs) and aquifer recharge in faulted–hydraulic drop alluvium aquifers in the Manas River Basin (China). The very high $^3$H activity (41.1–60 TU) in the groundwater in the Manas River upstream (south of the fault) indicates rainfall recharge during the nuclear bomb tests (since the 1960s). Carbon–14 groundwater age increases with distance (3000–5000 yrs in the midstream to > 7000 yrs in the downstream) and depth, as well as with decreasing $^3$H activity (1.1 TU) and $\delta^{18}$O values, confirming that the deeper groundwater is derived from paleometeoric recharge in the semi–confined groundwater system. MRTs estimated using an exponential–piston flow model vary from 19 to 101 yrs for CFCs and from 19 to 158 yrs for $^3$H; MRTs for $^3$H are much longer than those for CFCs probably due to the time lag (liquid vs. gas phase) through the thick unsaturated zone. The remarkable correlations between CFCs rather than $^3$H MRTs and pH, $SiO_2$, and $SO_4^{2-}$ concentrations allow estimating first–order proxies of MRTs for groundwater at different times. Relatively modern recharge is found in the south of the fault with young (post–1940) water fractions of 87–100 %, whereas in the north of the fault in the midstream area the young, water fractions vary from 12 to 91 % based on the CFC binary mixing method. This study shows that the combination of CFCs and $^3$H residence time tracers can help analyse groundwater MRTs and identify the recharge sources for the different mixing end–members."

The revised conclusions are as follows (RL542–554):

"In this study, we used environmental tracers and hydrochemistry to identify the modern and paleo–meteoric recharge sources, to constrain the different end–members mixing rates, and study the mixed groundwater MRTs in faulted–hydraulic drop alluvium aquifer systems. The paleo–meteoric recharge in a cooler climate was distinguished from the lateral flow from the higher elevation precipitation in the Manas River downstream area. The relatively modern groundwater with young (post–1940) water fractions of 87–100 % was obtained, indicating a small extent of mixing south of the fault. The short MRTs (19 yrs) along with the higher–than–natural $NO_3^-$ concentration (7.86 mg $L^{-1}$) south of the fault (headwater area) indicated the invasion of modern contaminants. This finding warrants particular attention. High mixing rate amplitudes varying from 12 to 91 % were widespread in north of the fault due to the varying depths of long–screened boreholes as well as within the aquifer itself. Furthermore, the mixing diversity was highlighted by the substantial water table fluctuations during groundwater pumping, vertical recharge through the thick unsaturated zone, and young water mixtures in different decades. The strong correlations between groundwater MRTs and hydrochemical concentrations enable a first–order proxy at different times to be used. In addition, this study has revealed that MRTs estimated by CFCs were more appropriate than those using $^3H$ in the arid MRB with a thick unsaturated zone."

2)    The meaning of the phrase "block water features" is not clear, possibly it means areas where there are strong (semi-vertical) contrasts in hydraulic conductivity (due to the thrust faults).

Response: Yes, the phrase "block water features" is not a very appropriate statement in this paper. What we want to tell the reader is that there are strong contrasts in hydraulic conductivity due to the thrust fault. The variant hydraulic conductivity also can be reflected by the geological and hydrogeological settings. Previous studies (Wu, 2007; Zhao, 2010) and other geological survey works in the Manas River Basin have indicated that the thrust faults shown in Fig. 1b are compressional faults and thus of water–blocking feature, which can explain the "a level difference of 130 m hydraulic drop is observed due to the thrust fault in the alluvium aquifer (Fig. 1c)".

A recent study by Bresciani et al (2018) has distinguished the mountain–front recharge (MFR) and mountain–block recharge (MBR) by using hydraulic head, chloride and electrical conductivity data in the arid basin. MFR predominantly consists of stream infiltration in the mountain–front zone, and MBR consists of subsurface flow from the mountain towards the basin. Manas River Basin aquifers may receive the recharge from the south mountain through the MFR mechanism, and more specific analysis will be carried out in the future work.

3)    Use of "apparent" ages in the preliminary discussion (Section 4.2.1) is defensible as described.

Response: We find that the phrase "apparent CFC ages" has been widely used in many other literatures (e.g. Darling et al., 2012; Hagedorn et al., 2011; Han et al., 2012; Happell et al., 2006; Koh et al., 2012; Plummer et al., 2006; Qin et al., 2011, 2012). However, a review paper by Suckow (2014) pointed out that the "apparent age" is "only well defined if the formula is given and if the tracer is stated". There are appropriate formulas for different tracers, such as $^{14}C$, $^{36}Cl$, $^{81}Kr$, $^3H/^3He$, and so on, but not for the CFCs, for $SF_6$ and for $^{85}Kr$. Therefore, Suckow (2014) thinks that, strictly speaking, the term "apparent age", should not be used for CFCs. This erroneous term "apparent age" for CFCs is also pointed out by Ref #2 ("L277: The paragraph on "apparent age" makes no sense for … and sampling").

We agree that the term "apparent age" for CFCs will not be used anywhere in our paper. As we know that the CFCs are synthetic organic compounds and largely released to the air since 1930s, and thus

they have been regarded as very good tracers for dating young water recharge time (post–1940 recharge). Therefore, we would like to use CFCs to explain the modern water recharge features.

The revised contents can be seen in Section 3.3.1 (RL172–191) and in Section 4.2.2 (RL340–406):

Section 3.3.1 (RL172–191):

[revised manuscript text omitted]

4)    Strictly, groundwater has "residence time" or "mean residence time"/"MRT" (being the time water takes to travel through a groundwater system to where it is sampled by a bore), rather than "transit time" or "mean transit time"/"MTT" which is generally reserved for streamflow (being the time for water to transit through the catchment and into the stream). Consequently, the word "residence" should be substituted for the word "transit" wherever "transit" appears. And also "MRT" for "MTT".

Response: Agree and changes made. The term "transit" was changed to "residence" and term "MTT" was changed to "MRT", and we insisted on the "residence" and "MRT" throughout the manuscript.

We re-read the literatures and found that term "transit time" was numerously used to indicate the time for water to transit through the catchment and into the stream (Cartwright et al., 2018; Cartwright and Morgenstern, 2015, 2016; Hrachowitz et al., 2009, 2010; Morgenstern et al., 2010; Stewart and Morgenstern, 2016). Stewart et al. (2010) pointed out that "Residence time is the time spent in the catchment since arriving as rainfall. Transit time is the time taken to pass through the catchment and into the stream." Leray et al. (2016) have adopted a general but robust definition for the residence time "the amount of time a moving element has spent in a hydrologic system", and considered the terms residence time, transit time, travel time, age, and exposure time as equivalent in their discussions. Custodio et al. (2018) used both residence times and transit times for groundwater samples collected from springs and deep wells. In our study, all of the groundwater samples were collected from the wells/artesian wells. Thus, we tend to use the term "residence" instead of "transit" in our manuscript.

5)    A selection of comments on the English are given below, to help the clarity of the writing. There

are many other very small infelicities in the English.

Response: We thank you very much for modifying the expressions of the manuscript. We have asked Chris Law (Wallace Academic Editing) to modify the language to help improve readability.

**Technical Corrections**

1) P1 L24-25 Change to "Quite 'modern' recharge is found in the south of the fault with young (post–1940) water fractions of 87–100 %, …" from "The quite 'modern' recharge in the south of the fault with young (post–1940) water fractions of 87–100 % is obtained, …"

Response: Agree and changes made. The sentence was changed to (RL23–25): "Relatively modern recharge is found in the south of the fault with young (post–1940) water fractions of 87–100 %, whereas in the north of the fault in the midstream area the young, water fractions vary from 12 to 91 % based on the CFC binary mixing method."

2) P2 L51 "Instead of" not "over for"

Response: Changes made (RL52). "over" was replaced by "than".

3) P2 L53 "closed" not "close"

Response: Agree and changes made (RL54). "close" was replaced by "closed".

4) P3 L89 "common" not "true"

Response: Agree and changes made (RL87). "true" was replaced by "common".

5) P3 L90-91 "Pumping from long-screened wells (of which there are over 10,000, Ma et al., 2018) …" not "Pumping from the long–screened over 10 000 boreholes (Ma et al., 2018) …"

Response: Agree and changes made. The sentence was changed to (RL88–90): "In particular, pumping from long–screened wells (of which there are over 10 000 boreholes, Ma et al., 2018) makes groundwater mixing mostly likely." Number "10 000" (not "10,000") is divided in groups of three using a thin space, complying with the "manuscript preparation guidelines for authors".

6) P3 L93 "result from" not "impacted by"

Response: Agree and changes made (RL90). "impacted by" was replaced by "result from".

7) P3 L94 "insufficiently recognised" not "insufficient recognition"

Response: Agree and changes made (RL91). "insufficient recognition" was replaced by "insufficiently recognised".

8) P4 L106 "total" not "totally"

Response: Agree and changes made (RL104). "totally" was replaced by "total".

9) P4 L107 "intermittently active" not "intermittent activity"

Response: Agree and changes made (RL105). "intermittent activity" was replaced by "intermittently active".

10) P4 L117 "depth" not "buried depth"

Response: Agree and changes made (RL115). Word "buried" was deleted.

11) P6 L177 "Manas River Basin" not "MRB"

Response: Agree and changes made (RL177). "MRB" was replaced by "Manas River Basin".

12) P9 L259 & 264 "slope" not "slop"

Response: Agree and changes made (RL277 and 282). Erroneous "slop" was changed to "slope".

13) P10 L300 "we use" not "one assign"

Response: Agree and changes made (RL350). "one assign" was replaced by "we use".

14) P10 L304 "increasing" not "elevated"

Response: Changes made. The sentence was deleted.

15) P11 L323 "indicates a larger fraction of 1960s precipitation recharge for G4 …" not "indicate that more fractions of the 1960s precipitation recharge was occurred for G4 …"

Response: Agree and changes made. The sentence was changed to (RL417–419): "First, increase in $^{3}$H activity in groundwater in the upstream area from 41.1 (G1 and G2) to 60 TU (G4) with distance indicated a larger fraction of 1960s precipitation for G4 than for G1 and G2; indeed, as seen in Fig. 2, near G4 samples exhibited the highest hydraulic gradient values.".

16) P12 L370 "generally" not "totally" and "overlap" not "overlapping"

Response: Agree and changes made (RL321). "totally" was deleted, and "overlapping" was replaced by "overlap".

17) P13 L397 delete "far"

Response: Agree and changes made (RL241). "far" was deleted.

18) P14 L413 "series" not "serious"

Response: Agree and changes made (RL468). "serious" was replaced by "series".

19) P14 L423 "Overall" not "Totally"

Response: Agree and changes made (RL477). "Totally" was replaced by "Overall".

20) P15 L448 "permit" not "permitting"

Response: Agree and changes made (RL502). "permitting" was replaced by "permit".

21) P15 L465 "other sources" not "either source" (?)

Response: Changes made. ", and giving more accurate prediction of the contaminants like nitrate than either source of information" was deleted.

22) P15 L466 "decreases" not "decrease"

Response: Agree and changes made (RL518). Erroneous "decrease" was changed to "decreases".

23) P16 L488 "which did not contribute groundwater recharge" not "which had non–contributes to groundwater recharge"

Response: Agree and changes made. The sentence was changed to (RL539–540): "…, which did not contribute to groundwater recharge in the arid regions of the Northwest China".

24) P17 L520 delete "have occurred"

Response: Agree and changes made. "have occurred" was deleted.

25) P19 L578 "… area) imply invasion of modern contaminants, …" not "… area), implying the modern contaminants invading, …"

Response: Agree and changes made. The sentence was changed to (RL547–548): "… area) indicated the invasion of modern contaminants."

"Groundwater mean transit times, mixing and recharge in faulted–hydraulic drop alluvium aquifers using chlorofluorocarbons (CFCs) and tritium isotope ($^{3}$H)" by Ma, B., Jin, M., Liang, X., Li, J., Hydrol. Earth Syst. Sci., doi:10.5194/hess-2018-143.

We appreciate the many valuable suggestions and helpful comments of **Anonymous Referee #2**. We have seriously considered all of the suggestions and comments and have attempted to address each of the comments point-by-point. Detail explanations are as follows.

Author's response – Line numbers referring to the old and revised version manuscripts are preceded by L and RL, respectively.

**Anonymous Referee #2**

1)    As indicated by the title, this manuscript presents the results of a groundwater dating and mixing study conducted using two different atmospheric tracers (CFCs and tritium). The two aims of the study were to (i) relate "ages" to local and general hydrogeological conditions and (ii) explore the possibility to use mineralisation as proxy for environmental tracers. I agree with referee #1 concerning the style, which is a huge disservice to the manuscript by its approximate use of technical terms and the general turn of phrase. I disagree however with the novelty (I do not see any) and the "substantial conclusions" (very unsubstantial and too dependent on mean transit time calculations that at present look extremely weak). As far as comment 4 of referee #1 is concerned, I think it is simply a matter of opinion and taste to use "transit time" instead of "residence time" (I prefer transit time because my work is related to solute transport problems, and "transit time" conveys this very idea of transport). One can argue over that, but it is really a hair splitting exercise.

Response: We would like to thank you very much for taking the time to review our manuscript. We think that the many valuable suggestions and helpful comments will help to improve our manuscript quality greatly.

We agree with you about the aims of the study and the opinion on the terms "transit time" and "residence time". We re-read the literatures and found that term "transit time" was numerously used to indicate the time for water to transit through the catchment and into the stream (Cartwright et al., 2018; Cartwright and Morgenstern, 2015, 2016; Hrachowitz et al., 2009, 2010; Morgenstern et al., 2010; Stewart and Morgenstern, 2016). Stewart et al. (2010) pointed out that "Residence time is the time spent in the catchment since arriving as rainfall. Transit time is the time taken to pass through the catchment and into the stream." Leray et al. (2016) have adopted a general but robust definition for the residence time "the amount of time a moving element has spent in a hydrologic system", and considered the terms residence time, transit time, travel time, age, and exposure time as equivalent in their discussions. Custodio et al. (2018) used both residence times and transit times for groundwater samples that collected from springs and deep wells. In our study, all of the groundwater samples were collected from the wells/artesian wells. Thus, we tend to use the term "residence time" instead of "transit time" in our manuscript.

In addition, we have reorganized the structure and tried our best to present a clear roadmap to readers. Some incorrect phrase or expressions, for example the "apparent CFC ages" (L277) and "The apparent ages (Table 2 and Fig. 5) estimated from the PFM …" (L509), which mislead the study and thus

have been revised and rewritten. In the revised manuscript, we tried to discuss the mean residence times and hydrochemistry evolution clearly and reasonably.

2)    Overall, the authors seem to have read sufficiently thoroughly the existing literature on the subject as well as the most recent developments (such as Kirchner's analysis of the effect of heterogeneity on mean transit time estimation using amplitude damping) and understood the different problems and pitfalls relevant for their study. However, the phrasing is sometimes very awkward and tends to obfuscate what the authors mean (see specific comments below).

Response: We have checked the manuscript carefully and have tried our best to modify some erroneous words and phrases that you and Ref #1 pointed out. Some ambiguous statements and sentences have also been rewritten. We have asked Chris Law (Wallace Academic Editing) to modify the language to help improve readability.

3)    But above all, I am missing a strong reason for this study to be published at all. As case study, it does not go beyond the classical scheme of sampling a few boreholes, analyse the groundwater samples for one or more tracers, calculate some kind of "age" and correlate it to depth or water chemistry. Doing so however, the authors try to apply different methods (lumped-parameter modelling, binary mixing) without presenting a clear roadmap.

Response: Manas River with the longest channel length and the largest river flow is representative among the hundreds of inland rivers in the Junggar Basin in the arid northwest China (Fig. 1a). There are more than 1 million people in the Manas River Basin. Groundwater is the very important and even the only water resources for the local residents. However, the paucity of research on groundwater mean residence time, mixing and recharge features have hampered the rational development and utilization of water resources in the Junggar Basin. We think that the case study of our manuscript is very essential and can be a good proxy for other analogous areas in the Junggar Basin.

Yes, we tried to estimate the mean residence time using lumped–parameter models. In addition, we also have tried to recognize the modern and paleo–meteoric recharge features. However, it is a pity that there are many deficiencies for the initial manuscript. We have reorganized the structure and tried our best to present a clear roadmap to readers. The outline of the manuscript have been reorganized as follows:

Title: Application of environmental tracers for investigation of groundwater mean residence time and aquifer recharge in faulted–hydraulic drop alluvium aquifers

1. Introduction
2. Geological and hydrogeological setting
3. Materials and methods
    3.1 Water sampling
    3.2 Analytical techniques
    3.3 Groundwater dating
        3.3.1 CFCs indicating modern water recharge
        3.3.2 The apparent $^{14}$C ages
        3.3.3 Groundwater mean residence time estimation
4. Results and discussion
    4.1 Stable isotope and major ion hydrochemistry

4)    Model choice in particular is strangely presented: first, "apparent age" is presented as "based on the hypothesis of pistonflow". Then that very piston-flow model is used although mixing is supposed to be "most likely" either within the aquifer or at the sampling point. This is completely contradictory and there is no reason not to apply another model to the CFC data (and for that matter, to the 14C data as well. See Custodio et al., 2018).

Response: We have deleted "apparent CFC ages" and rewritten the content of the model choice for mean residence time estimation (Section 3.3.3 (RL219–268)):

[revised manuscript text omitted]

$$g(\tau) = 0 \qquad \text{for} \quad \tau < \tau_{\mathrm{m}}(1-1/\eta) \tag{3a}$$

$$g(\tau) = \frac{\eta}{\tau_{\mathrm{m}}} e^{(-\eta\tau/\tau_{\mathrm{m}} + \eta - 1)} \qquad \text{for} \quad \tau \geq \tau_{\mathrm{m}}(1-1/\eta) \tag{3b}$$

The dispersion model (DM) mainly measures the relative importance of dispersion to advection, and is applicable for confined or partially confined aquifers (Małoszewski, 2000). Its RTD is given by

$$g(\tau) = \frac{1}{\tau\sqrt{4\pi D_{\mathrm{P}}\tau/\tau_{\mathrm{m}}}} e^{-\left(\frac{(1-\tau/\tau_{\mathrm{m}})^2}{4\pi D_{\mathrm{P}}\tau/\tau_{\mathrm{m}}}\right)} \tag{4}$$

The weighting function of the exponential mixing model (EMM) is

$$g(\tau) = \frac{1}{\tau_{\mathrm{m}}} e^{(-\tau/\tau_{\mathrm{m}})}, \tag{5}$$

where $\tau_{\mathrm{m}}$ is the mean residence time, $\eta$ is the ratio defined as $\eta = (l_{\mathrm{P}} + l_{\mathrm{E}})/l_{\mathrm{E}} = l_{\mathrm{P}}/l_{\mathrm{E}} + 1$, where $l_{\mathrm{E}}$ (or $l_{\mathrm{P}}$) is the length of area at the water table (or not) receiving recharge, $D_{\mathrm{P}}$ is the dispersion parameter, which is the reciprocal of the Peclet number (*Pe*) and defined as $D_{\mathrm{P}} = D/(vx)$, where $D$ is the dispersion coefficient (m$^2$ day$^{-1}$), $v$ is velocity (m day$^{-1}$), and $x$ is distance (m).

Each RTD has one or two parameters, MRT ($\tau_{\mathrm{m}}$) is determined by convoluting the input (the time series $^3$H and CFCs input in rainfall) to each model to match the output (the measured $^3$H and CFC concentrations in groundwater), and other parameters ($\eta$ and $D_{\mathrm{p}}$) are determined depending on the hydrogeological conditions. To interpret the ages of the MRB data set, EPM ($\eta$=1.5 and 2.2), DM ($D_{\mathrm{p}}$=0.03 and 0.1), and EMM models were used, after which MRTs with different RTDs were cross–referenced."

We read the literature carefully and think that both the method and idea are very innovative by Custodio et al. (2018), who related chloride (Cl) and $^{14}$C activity changes in recharge for aquifers in the arid area to indicate the effect of variations in recharge rate during the previous wetter–than–present period.

We think that it would be also good applications in other arid areas around the world because the climatic change is global. In our manuscript, the apparent [14]C ages estimation was adopted. The changes in groundwater reserves and [14]C content will be conducted in the future in the Manas River Basin.

5)  I know it is customary to interpret CFC data assuming piston-flow, but it is nonetheless a priori wrong. Model choice must be substantiated from knowledge of the hydrogeological situation and the sampling scheme (Maloszewski and Zuber, 1982; Leray et al., 2016). Later on in the manuscript however different models are used in the binary mixing plots, and model choice is discussed briefly. Why use the "apparent age" concept at all, then ? This is confusing and reads like the two authors have written separately different parts of the manuscript and then pasted the two parts together.

Response: Sorry, we have realized that the apparent CFC ages are not appropriate. As we know that the CFCs are synthetic organic compounds and largely released to the air since 1930s, and thus they have been treated as good tracers for dating young water recharge time (post–1940 recharge).

We totally agree with you that "Model choice must be substantiated from knowledge of the hydrogeological situation and the sampling scheme". Therefore, in the revised manuscript, we have rewritten the content and have discussed the model choice in detail in Section 3.3.3 (See Response 4).

The mean residence times estimated by the lumped–parameter models have been deleted in the binary mixing plots (Fig. 7). However, the binary plots of CFC vs. CFC and CFCs vs. [3]H are widely used in the literatures and can provide useful information on the co–existence of young water with old water (Cook et al., 2017; Han et al., 2007; Han et al., 2012; Koh et al., 2012; Qin et al., 2011), as well as can provide a powerful tool to recognize contamination, degradation and irrigation infiltration and so on (Cook et al., 2017; Han et al., 2015; Mahlknecht et al., 2017; Qin et al., 2012). However, these information mentioned above have not been clarified clearly before in the Manas River Basin, and even not in the Junggar Basin (Fig. 1a). Therefore, we think that the binary mixing plots are essential in the manuscript in order to tell the reader more information about groundwater mixing or recharge features.

6)  I also have my doubts concerning the calculations of the mean transit times as they are presented.

Response: Take the CFC–12 and exponential–piston flow model (1.5) for example, the calculation process of the mean residence times is as follows: First, we chose the Eq. (2b) as the convolution integral, and chose the exponential–piston flow model (Eq. (3a, 3b)) as the system response function. Second, we used the time series CFC–12 trend of Northern Hemisphere atmospheric mixing ratio (1940–2014, http://water.usgs.gov/lab/software/air/cure/) as input concentrations. We treated the calendar year 2015 (groundwater sampling time) as age=0 yrs by convoluting the input (times series of CFC–12 input) to the EPM (1.5), and increased the mean residence time $\tau_m$ from 1 to 500 yrs with the time step of 1 yr within 100 yrs and with the time step of 5 yrs between 101 to 500 yrs. Then we can get a sequence of results of output CFC–12 concentrations and mean residence times (vary from 1 to 500 yrs). Third, we plotted the output CFC–12 concentrations vs. mean residence times and then compared the measured groundwater CFC–12 concentrations to get the groundwater mean residence times. The computational procedures were conducted by using MATLAB (version R2014a). The output CFC–12 concentrations decreased from 526.4 pptv with $\tau_m$ of 1 yr to 3.0 pptv with $\tau_m$ of 155 yrs. As

the detection limit for each CFC is about 0.01 pmoL$^{-1}$ of water (be equal to 3.54 pptv with the laboratory temperature of 25℃), the output CFC–12 concentrations lower than 3.54 pptv can be neglected.

7) The method with which the tritium input has been reconstructed is not documented properly (which stations were used, and how long were the availabe time series?) and the estimated modern value (31 TU) seems extremely high compared to Western Europe for instance (about 6 TU). Is that because of the Chinese nuclear tests of the 60s and 70s that are being referred to in the introduction, or the result of some kind of regional effect?

Response: Agree and changes made. The input tritium ($^3$H) activity in Urumqi station was reconstructed as follows: First, as we know that there is a proportionality between the logarithmic precipitation $^3$H activity and the latitude in the Northern Hemisphere, we can reconstruct the Urumqi precipitation $^3$H activity according to the following equation:

$$\lg C_U = \lg C_H + \frac{\lg C_I - \lg C_H}{X_I - X_H} * (X_U - X_H) \tag{S1}$$

where $C_U$ is Urumqi precipitation $^3$H activity, $C_H$ is Hong Kong precipitation $^3$H activity, $C_I$ is Irkutsk (in Russia) precipitation $^3$H activity, $X_U$ is Urumqi latitude, $X_H$ is Hong Kong latitude, and $X_I$ is Irkutsk latitude. The precipitation $^3$H activity are available in the International Atomic Energy Agency (IAEA). The results based on Eq (S1) are show in Table S1.

Table S1. The reconstructed Urumqi precipitation $^3$H activity between 1969 and 1983.

| Date | $C_H$ (TU) | $C_I$ (TU) | $C_U$ (TU) | Ottawa precipitation (TU) |
|------|-----------|-----------|-----------|---------------------------|
| 1969 | 46.93 | 454.79 | 248.63 | 253.66 |
| 1970 | 29.38 | 464.15 | 222.82 | 190.77 |
| 1971 | 24.38 | 516.61 | 229.38 | 206.1 |
| 1972 | 27.86 | 247.23 | 138.36 | 92.34 |
| 1973 | 13.22 | 173.34 | 87.44 | 90.41 |
| 1974 | 17.47 | 216.58 | 110.90 | 98.07 |
| 1975 | 12.42 | 190.93 | 92.33 | 75.86 |
| 1976 | 11.71 | 148.8 | 75.69 | 58.91 |
| 1977 | 12.24 | 150.72 | 77.31 | 73.93 |
| 1978 | 10.89 | 103.11 | 56.72 | 73.63 |
| 1979 | 8.96 | 92.06 | 49.55 | 49.62 |
| 1980 | 7.63 | 141.8 | 65.20 | 49.54 |
| 1981 | 11.53 | 138.3 | 71.44 | 55.09 |
| 1982 | 9.73 | 66.92 | 40.08 | 47.29 |
| 1983 | 21.14 | 127.26 | 78.96 | 40 |

Second, a good linear regression between the $C_U$ and Ottawa precipitation $^3$H activity can be obtained as follows:

$$C_{Ui} = 1.0102 C_{Ottawa} + 11.647 \ (R^2 = 0.933, n = 15) \tag{S2}$$

where $C_{Ottawa}$ is Ottawa precipitation $^3$H activity, $C_{Ui}$ is the Urumqi annual precipitation $^3$H activity.

Third, considering the precipitation amount effect, the reconstructed Urumqi precipitation $^3$H activity

can be obtained as follows:

$$C_{RU} = C_{Ui} * \frac{P_i}{P_m} \qquad \text{(S3)}$$

where $C_{RU}$ is the reconstructed Urumqi precipitation [3]H activity shown in Fig. 6 (estimated [3]H), $P_i$ is the annual precipitation amount (1953–2015), $P_m$ is the annual mean precipitation amount in Urumqi between 1953 and 2015.

Yes, the estimated modern precipitation [3]H activity are indeed extremely high (mean value of 31.55 TU between 2000 and 2015). We speculate that the very high precipitation [3]H activity may be ascribed to both the Chinese atmospheric nuclear tests (around 350 km away to Manas River Basin) from 1964 to 1974 (Zhou, 1992) and the continental effect (Tadros et al., 2014), where the Manas River Basin is more than 3500 km away from the western pacific. The high precipitation [3]H activity was also shown in Fig. S1 (Cauquoin et al., 2015).

[Figure]

Figure S1. Comparison of modelled tritium in precipitation with surface snow samples from two ITASEAntarctic traverses (Proposito et al., 2002; Becagli et al., 2004) and some pre-bomb measurements. (The figure is from Cauquoin et al., 2015, Fig. 10d)

8)    Furthermore, the authors do address the non-uniqueness problems that are bound to arise when calibrating an exponential piston-flow or a dispersion model (2 free parameters each) using a single tritium measurement in aquifers that still retain some of the "bomb tritium" (see Stewart et al., 2010 for details), but in a terribly confusing way and without first explaining the rationale and the approach taken.

Response: Sorry. We should have given more details about the two free parameters each for the exponential–piston flow and the dispersion model. These parameters are substantial based on the geological/hydrogeological conditions. We have added some more detailed explanations concerning the rational and the approach taken in Section 3.3.3 (Response 4). The revised version can be seen in the "Response 4)".

9)    I suppose figure 8 was meant to show the range of parameters that match the measured tracer concentrations. That's commendable, but badly explained.

Response: Yes, thanks. We have redrawn the figure 8 to show the most useful information to the readers (see the "Specific Comments Response 55").

10) In the final step relating mineralisation to transit time, the authors finally select the EPM calibrated with the CFC12 measurements, but this is once again presented in a unclear fashion.

Response: There are some reasons that we select the EPM (1.5) calibrated with the CFC–12 to relate hydrochemistry evolution. First, groundwater CFCs results show that the urban air with CFC compounds contaminations, which generally cause elevated CFC than the global background atmospheric CFC concentrations (Northern Hemisphere), are unlikely (Fig. 7) in the Manas River Basin. Second, CFC–12 has much less propensity for degradation and/or contamination than CFC–11 and CFC–113. Third, the exponential–piston flow model is one of the mostly commonly used system response functions. The choice of exponential–piston flow model is suitable for the aquifer that has two segments, in which a segment of exponential flow followed by a segment of piston flow. As shown in Fig. 1c and relative hydrogeological discussions, exponential–piston flow model is a good choice to be chose as the system response function. In addition, the mean residence times mathematic relation between EPM (1.5) and other system response functions are obtained (Fig. 11). We can relate mineralization to mean residence time based on these mathematic relations.

11) The discussion is too long, relies too much on untested and untestable hypotheses, and presents so many singular and unfocused results that it is difficult for the reader to grasp a clear picture of their meaning and significance. The paragraph on "apparent age" should be scraped altogether and the different estimates of "age" (i.e. mean transit time of the respective model) and mixing ratios organised in a clear and synthetic manner.

Response: To avoid misleading, we have deleted some irrelevant content, such as the incorrect statement of "apparent age" for the CFCs, and the incorrect hypothesis of piston flow as mixing is supposed to be "most likely" either within the aquifer or at the sampling point. We also tried to present a clear roadmap to readers, including modern and paleo-meteoric recharge features, mean residence time estimated by LPMs, and mixing features as well as irrigation infiltration indications.

12) All in all, the manuscript must be seriously reorganised and streamlined. The calculation of mean transit times of the different tracers must be redone, removing entirely the "apparent age" nonsense and explaining clearly the different steps taken by the authors to (I) select a model (II) explore model parameter range and (III) compare the different results obtained from tritium, the CFCs and carbon 14. Interpretation of the obtained "ages" in terms of hydrogeology and its correlation to hydrochemistry must then be presented in a clear and synthetic fashion. Only when this is done might the manuscript rise above an unoriginal and confusing rehash of previous studies, and could be considered for publication.

Response: We reorganized the structure and tried our best to present a clear roadmap to readers. We rewrote the Section 3.3.3 in which system response functions were selected by considering the applicable hydrogeological situations. Parameter and mean residence times were recalculated and then explained in a much clearer way. Figures 8 and 9 (They are Figs. 10 and 11 in the revised manuscript) were redrawn to show much useful information to readers. The controversial "apparent CFC age" was deleted and then was shown in a more proper way to indicate the modern precipitation recharge.

**Specific Comments**

1) Please ask the help of a proof reader to help improve readability

Response: Agree. We have asked Chris Law (Wallace Academic Editing) to modify the language to help improve readability.

2) L11: Why is it crucial ? Please explain or leave that out.

Response: Agree. As we know that groundwater are the very important and even the only water resources for the local residents in the Manas River Basin of arid Northwest China (It is also the same in many other arid areas around the world). Two recharge sources which, modern precipitation recharge and paleo–meteoric recharge, are known as the main sources for the groundwater in arid areas. Knowledge the recharge sources is thus can provide correct guidance for extraction of groundwater. In addition, groundwater renewability is also an important factor for guiding extracting groundwater. Groundwater residence time reveals information about water storage, mixing and transport in subsurface water systems, which has contributions to the rational development of groundwater (Cartwright et al., 2017; Custodio et al., 2018; Dreuzy and Ginn, 2016; Leray et al., 2016). Therefore, we tend to retain the sentence (L11) in consideration of the explanation mentioned above.

3) L15: "indicating the rainfall recharge…" You mean that the young water component is higher than in samples with lower tritium activity.

Response: The inexact expression generates misunderstanding. We mean the very high $^3$H activity (41.1–60 TU) of groundwater indicate rainfall recharge during the nuclear bomb.

4) L29: The title of this section is not very telling, and this is not really what the study is about, is it?

Response: Agree and changes made (RL27). The title was changed to "Introduction".

5) L33: "may be renewable". What do you mean ? Something about short turnover time ?

Response: Changes made. We think that "renewable" may means groundwater being recharged by other water resources after extraction (or discharge).

The sentence was changed to (RL32–34): "This is particularly crucial in alluvium aquifers where fresh groundwater renewability is generally strong (Huang et al., 2017), thus functioning as potable water resources in the arid areas; also, alluvium aquifers are more vulnerable to anthropogenic contaminants and land–use changes (Morgenstern and Daughney, 2012)."

6) L37-39: Rewrite the entire sentence.

Response: Agree and changes made. The entire sentence was changed to (RL35–40): "Because the residence time distribution in subsurface water systems cannot be empirically measured, a commonly used approach is parametric fitting of trial distributions to chemical concentrations (Leray et al., 2016; Suckow, 2014). The widely used lumped–parameter models (LPMs; Małoszewski and Zuber, 1982; Jurgens et al., 2012), which commonly assume that the hydrologic system is at a steady–state, have been applied to subsurface water systems containing young water with modern tracers of variable input concentrations (e.g. seasonably variable stable isotope $^2$H and $^{18}$O, tritium, and $^{85}$Kr; Cartwright et al., 2018; McGuire et al., 2005; Morgenstern et al., 2015; Stewart et al., 2010)."

7) L38: "and may be inferred". You mean "must be inferred".

Response: Agree and changes made (RL35–40). Yes, the residence time distribution cannot be empirically measured. See "Specific Comments Response 6".

8) L39: "at" steady-state, not "in" steady-state.

Response: Agree and changes made (RL38). "in" was replaced by "at".

9) L40: "Three types of transit time". You mean three time windows ?

Response: Changes made. The sentence was changed to (RL40–41): "The groundwater residence time tracers can be classified into three types depending on their time span."

10) L46: It's not variability, rather time span.

Response: Agree and changes made (RL41). "variabilities" was replaced by "time span".

11) L48: "in a similar function with" should read "in a similar way to".

Response: Agree and changes made (RL48). "in a similar function with" was replaced by "in a similar way to".

12) L51: replace "over" with "than".

Response: Agree and changes made (RL52). "over" was replaced by "than".

13) L54: You mean that increasing transit time through the aquifer leads to increasing mineralisation.

Response: No change made. That "increasing transit time through the aquifer leads to increasing mineralisation" is reasonable under certain circumstance. However, groundwater mineralisation is a very complex process, which contains mineral dissolution, precipitation, hydrolysis, evaporation concentration, water–rock interaction, and so on (Ma et al., 2018). In addition, water corrosion, solubility of rocks, and groundwater runoff will also have strong impacts on mineralization (Shen et al., 1993). Thus, increasing transit time not necessarily leads to increasing mineralization. The sentence and cited literatures (L54) in the manuscript tend to show that there are some correlations between the transit time and mineralsation in some situations. Nevertheless, the correlation is still unclear in the Manas Rriver Basin, and that is right the research objective in our manuscript.

14) L55: Please explain why tritium is "the only true age tracer", namely because it is part of the water molecule.

Response: Agree and changes made. Yes, it is. The sentence was changed to (RL56): "The only true age for water is $^3$H, a component of the water molecule with a half–life of 12.32 yrs (Tadros et al., 2014)."

15) L56 (entire paragraph): Why mention the southern hemisphere at all, since the study takes place in the northern hemisphere ? This is useless information.

Response: Agree and changes made. We have deleted the statements concerning the Southern Hemisphere $^3$H activities and relevant references (L61–66).

16) L66: "may be used to estimate MTTs" should read "must be used to estimate MTTs". And explain

why (non-unicity problems…).

Response: Agree and changes made. Yes, we totally agree with your opinion. As the mean annual $^3$H activities peaks were several hundred times natural levels in the Northern Hemisphere due to the atmospheric thermonuclear tests in the Northern Hemisphere between the 1950s and 1960s, the nowadays rainfall $^3$H activities are still affected by the tail–end of the bomb–pulse in the Northern Hemisphere. Groundwater residence time cannot be assigned based on measurement of a single sample $^3$H activity.

The sentence was changed to (RL62–64): "Thus, measurement of a single sample of $^3$H activity does not accurately assess the groundwater MRTs in the Northern Hemisphere (Cook et al., 2017), and time–series $^3$H measurements with LPMs are required (Han et al., 2007; Han et al., 2015)."

17) L69: You are confusing residence time and degradation half-life. The residence time of the CFCs in the atmosphere is no different from that or tritium or any other tracer. The difference lies in their half-lives (degradation for CFCs, decay for tritium), which are very long for the CFCs.

Response: Agree and changes made. We agree with your opinion on the "residence time" and "degradation half–lives" for the atmospheric environmental tracers (e.g. CFCs and $^3$H). CFCs degrade slowly in the atmosphere and have relatively long degradation half–lives, while the decay half–life for $^3$H is much shorter with 12.32 yrs. The input function for CFCs is not area–specific as is the case with $^3$H. Therefore, the atmospheric concentrations for CFCs are uniform over large areas (Cartwright et al., 2017; Cook et al., 2017), while that for $^3$H often varies across latitudes and between seasons (Tadros et al., 2014).

The sentence was changed to (RL65–66): "In contrasting to $^3$H, CFCs degrade slowly in the atmosphere and have relatively long degradation half–lives, which permits their uniform atmospheric distributions over large areas."

18) L78-82: Please rephrase the entire sentence.

Response: Agree and changes made. The sentence was rewritten as follows (RL75–80): "However, groundwater MRTs may not be always be accurate when calculated using CFCs: for example, MRTs are much lower than the actual values if CFC inputs are entrapped excess air in the unsaturated zone during recharge (Cook et al., 2006; Darling et al., 2012) or contaminated in urban and industrial environments (Carlson et al., 2011; Han et al., 2007; Mahlknecht et al., 2017; Qin et al., 2007), and are much higher if CFC inputs are degraded in anaerobic groundwater (most notably CFC–11 and CFC–113; Cook and Solomon, 1995; Horneman et al., 2008; Plummer et al., 2006b)."

19) L89: "Mixing […] is particularly true…". You don't know that, it's a probable hypothesis!

Response: Agree and changes made. Though the previous study (Ma et al., 2018) have confirmed that the groundwater lateral–flow mixing is common in the alluvium aquifers of the Manas River Basin, more work and studies should be conducted to confirm the mixtures within the aquifers and long–screened wells. On the other hand, groundwater mixing can occur within the aquifer itself (Cook et al., 2017) as well as occur during pumping from long screened wells. However, for the case study of the Manas River Basin, mixing ascribed to pumping from long–screened wells has not been demonstrated before.

The sentence was changed to (RL87–88): "The hypothesis mixing within the aquifers and pumping from the long–screened wells is common in the faulted–hydraulic drop alluvium aquifers of the Manas River Basin (MRB) in the arid regions of Northwest China (Fig. 1)."

20) L93-95: "The MTTs that impacted….". This sentence makes no sense. Rewrite.

Response: Agree and changes made. The sentence was changed to (RL90–92): "The MRTs that result from a deep unsaturated zone (water table depth is 180 m) and contrasting geological settings (a level difference of 130 m hydraulic drop caused by the thrust fault) are still insufficiently recognised in the alluvium aquifer (Fig. 1c)."

21) L106: "with totally length" should read "with total length".

Response: Agree and changes made (RL104). "totally" was replaced by "total".

22) L107: "was intermittent activity" should read "was intermittently active".

Response: Agree and changes made (RL105). "intermittent activity" was replaced by "intermittently active".

23) L110: So the different aquifers are all fractured rock aquifers.

Response: Yes, the substantial fractures and fissures of rock aquifers in the mountain area allow groundwater migration to be possible. The literatures by Cui et al. (2007) and Zhou (1992) were cited in the relevant discussion.

24) L114: "is macroscopically similar". What do you mean with "macroscopically" ?

Response: Changes made. The sentence was changed to (RL112): "The groundwater flow direction is consistent with the Manas River flow direction."

25) L120: "is as large" should read "is as deep".

Response: Agree and changes made (RL118). "large" was replaced by "deep".

26) L124: How many samples were taken altogether ? And are there any information concerning screening depth and size (fully penetrating wells or not) ? This is important information to guide model choice.

Response: Agree and changes made. The important information concerning sample numbers as well as screening depth and size were added in the revised manuscript.

The sentences were changed to (RL122–126):

"In total, 29 groundwater (pumped from fully penetrating well, of which 3 are from spring and 3 are from the artesian well) were collected along the Manas River during June to August, 2015 (from G1 to G29 in Table 1 and Fig. 2). Groundwater were separated into three clusters including the upstream groundwater (UG, south of the Wuyi Road), midstream groundwater (MG, area between the Wuyi Road and the West Main Canal–Yisiqi), and downstream groundwater (north of the West Main Canal–Yisiqi) based on the hydrochemistry and stable isotope data."

27) L126: What was the rationale for separating the samples into three groups ? For instance, why is G13 MG while G26 is DG ? DG seems like the downgradient boundary. Did you use the piper diagrams to separate the samples ?

Response: Agree and changes made. Yes, you are exactly right that the three groundwater groups are based on the result of hydrochemistry and stable isotope. Yisiqi (Fig. 2) is the dividing line where there is an obvious change for the hydrochemistry and stable isotope along the Manas River motion (Fig. S2 from Ma et al., 2018). We also use piper diagrams to separate the samples (Fig. 4).

[Figure]

Figure S2. Relationship between $\delta^{18}O$ and Cl concentrations for waters as a means to differentiate hydrogeological processes in the Manas River Basin. The purple circles represent the upstream and midstream groundwater, and green, red, and black legends represent downstream groundwater (from Ma et al., 2018).

28) L152: "were followed" is used multiple times but should read "after" or "following".

Response: Agree and changes made (RL151 and 156). We have checked the manuscript carefully and seriously have corrected the erroneous words. "were followed" was changed to "were after" and we insisted on the "after" throughout the manuscript.

29) L173: "refers" is not the proper verb. Use "depends" for instance.

Response: Agree and changes made. The sentence was changed to (RL173–174): "The difference between the local and global background atmospheric mixing ratios of CFCs in the Northern Hemisphere – *CFC excess* – varies substantially based on the industrial development."

30) L179: What are low latitude countries ?

Response: Changes made. The vague expression was deleted. As the atmospheric CFC concentrations are affected by the industrial development, the time series of Northern Hemisphere atmospheric CFC concentrations were widely treated as the background values for the underdeveloped areas (it is also applied to the Manas River Basin in our manuscript).

31) L189: The entire procedure is correctly explained, and also the fact that "apparent age" implies piston-flow transit time distribution, but why use apparent age in the first place ? Piston-flow is one model among many, as the authors explain later in the manuscript. Furthermore, the entire concept of "age" is problematic and should be replaced by mean transit time or mean residence time (for an in-depth discussion, see Suckow, The age of groundwater-Definitions, models and why we do not

need this term, Applied Geochemistry 50, 2014, 222-230).

Response: Agree and changes made. The erroneous "apparent CFC age" has been deleted. The concept of "age" is replaced by mean residence times and estimated by lumped–parameter models with the exponential–piston flow model, dispersion model, and exponential model, which are discussed in Section 4.3.1. In addition, the groundwater recharge sources (e.g. to distinguish the modern and paleo–meteoric recharge features) identified by the CFCs are also discussed in Section 3.3.1 and Section 4.2.2.

32) L194: What do you mean by "closed system" ? Physically bounded ?

Response: Changes made. As we know that the measured tracer output concentration in the groundwater is compared to its historical input using the convolution integral, in which the system response function is selected based on the different hydrogeological subsurface flow systems (Jurgens, et al., 2012; Małoszewski and Zuber, 1982; the details of calculation is shown in "Response 6"). The tracer output concentration can be measured from the spring, stream, well, and so on. There is no absolutely physical boundary for the subsurface flow system. Therefore, the ambiguous statements "closed system" is not rigorous and thus is deleted. See Section 3.3.3 (Response 4).

33) L208: "were given below as transit time distribution function" should read "were selected and are given below".

Response: Agree and changes made. This sentence was deleted and more detailed interpretations were added (See Section 3.3.3 in the Response 4).

34) L219: You should also explain here how you planned to choose between these competing models.

Response: Agree and changes made. The system response function is selected based on the hydrogeological settings and the sampling position of tracer output concentration measured. We explain the basis on the choice of the models. See Section 3.3.3 (Response 4).

35) L235: Why present an equation you will not be using for lack of appropriate data ?

Response: Agree and changes made. We have deleted this equation and the accompanying description.

36) L240: This is true for the piston-flow model only ! See Custodio et al. for details.

37) L250: So the entire paragraph boils down to using literature values for the initial 14C activity. Make it shorter and to the point.

Response: Agree and changes made. Custodio et al. (2018) indicated that the calculated groundwater $^{14}$C age is really an apparent age due to the mixture of waters with different transit times, which was also indicated by other researchers (e.g. Cook et al., 2017; Suckow, 2014). Therefore, we deleted some unnecessary description and unimportant information in the text. Some sentences were reduced and revised. The revised content can be seen in Section 3.3.2 (RL193–217):

"Carbon–14 ($^{14}$C, half–life: 5730 yrs) activity in groundwater is often used to estimate groundwater age over time periods of approximately 200 and 30 000 yrs, and to determine the recharge from mixing water in various climate conditions (Cook, et al., 2017; Custodio et al., 2018; Huang et al., 2017).

Calculation of [14]C groundwater age may be complicated if dissolved inorganic carbon is derived from a mixture of sources or [14]C originating from the atmosphere or soil zone is significantly diluted by the dissolution of [14]C–free carbonate minerals in the aquifer matrix and biochemical reactions along the groundwater flow paths (Clark and Fritz, 1997). Although only minor carbonate dissolution is likely, determination of groundwater residence times requires [14]C correction (Atkinson et al., 2014). When the dissolution of carbonate during recharge or along the groundwater flow path may dilute the initial soil $CO_2$, $\delta^{13}C$ can be used to trace the process (Clark and Fritz, 1997). An equation for the reaction between $CO_2$–containing water with a carbonate mineral is commonly written as follows (modified after Pearson and Hanshaw, 1970):

$$CO_2 + H_2O + CaCO_3(\delta^{13}C_{carb} = 0) \rightarrow Ca^{2+} + 2HCO_3^-(\delta^{13}C_{DIC}), \tag{R1}$$

where $\delta^{13}C_{carb}$ is the dissolved carbonate $\delta^{13}C$ value (approximately 0; Clark and Fritz, 1997), and $\delta^{13}C_{DIC}$ is the measured $\delta^{13}C$ value in groundwater.

Depending on knowing the measured [14]C activity after adjustment for the geochemical and physical dilution processes in the aquifer (without radioactive decay), the groundwater apparent [14]C ages ($t$) can be calculated from the following decay equation:

$$t = -\frac{1}{\lambda_{^{14}C}} \times \ln \frac{a^{^{14}}C}{a_0^{^{14}}C}, \tag{1}$$

where $\lambda_{^{14}C}$ is the [14]C decay constant ($\lambda_{^{14}C} = \ln 2/5730$), and $a^{^{14}}C$ is the measured [14]C activity of the

DIC in groundwater. As mentioned above, the estimated ages are really apparent ages due to the mixture of waters with wide range of ages (Custodio et al., 2018; Suckow, 2014).

Previous studies in the arid northwest China (Edmunds et al., 2006; Huang et al., 2017) have concluded that a volumetric value of 20 % "dead" carbon derived from the aquifer matrix was recognized, which is consistent with the value (10–25 %) obtained by Vogel (1970). Therefore, the initial [14]C activity ($a_0^{^{14}}C$) of 80 pMC is used to correct groundwater [14]C ages (results are shown in Table 1), despite this simple correction makes no attempt to correct the age of individual samples that may have experienced different water–rock interaction histories."

38) L264: Check the discussion paper by Benettin et al. in review in HESS for the latest developments on the "evaporation slope".

Response: Agree and changes made. We agree with the opinion on the evaporation slope that a steeper slope would be obtained due to the large source variability under different meteorological conditions in different seasons indicated by Benettin et al. (2018). As we know that water fractionation is affected by various factors, like the water surface temperature, air humidity, wind speed, and so on, but we can always obtain a linear trend from a source water with low salinity (Ma et al., 2015). However, in practice, the evaporation trend line is often obtained from various sources water, which is often not true evaporation line. In our manuscript, the evaporation line is estimated according to the surface water (ditch water, reservoir water, and Manas River water), which are all collected in summer of 2015. The ditch, reservoir and Manas River are always connected to one another (Fig. 1b), and all are recharged from the mountain areas in the same season. Though there are minor differences of the water sources for the surface water, the linear trend obtained based on these surface water may have implications for the surface water evaporation.

Therefore, we use the evaporation slope and to add some statements (as follows) for the rationality in the revised manuscript (RL282–286):

"A recent study (Benettin et al., 2018) indicated that the evaporation line obtained from various sources of water is often not the true evaporation line. All samples of surface water in the present study were collected in the summer of 2015 and were recharged from the mountain areas in the same season. Although they were collected from different areas (ditch water, reservoir water, and Manas River water), the linear trend obtained may have implications for surface water evaporation."

39) L274: The entire paragraph is too short and should explain clearly the approach adopted to calculate "ages" from the tracer data (model and model parameter choices !). I strongly advise against using binary mixing diagrams, and encourage the authors to use a multi-tracer modelling approach trying to find a single optimum or optimal parameter regions for the different tracers.

Response: Changes made. The entire paragraph was deleted. We reorganized the structure of the Section "4. Results and discussion", rewrote some contents and deleted some incorrect expressions, and added statements concerning the model choice in Section "3. Materials and methods". However, we tend to retain the binary mixing diagrams (the mean residence times with different models have been deleted in the diagrams) to explain the young and old groundwater mixing features and to identity the impact of human activity on the groundwater.

40) L277: The paragraph on "apparent age" makes no sense for the reasons given above. I disagree with the proposition that "they [apparent ages] provide a good first approximation for groundwater age". There is no reason to prefer the piston-flow model which is implied by the "apparent age" concept over other models. This argument has been for years a lazy way to skip responsibility in choosing one model based on knowledge of the hydrogeological situation and sampling.

Response: Agree and changes made. The "apparent CFC ages" has been deleted. In addition, we tend to discuss the modern precipitation recharge features and to distinguish the modern and paleo–meteoric recharge using CFCs in Section 4.2.2 (RL340–406):

[revised manuscript text omitted]

41) L297: "which confirms". A performative statement confirms nothing. You are supposing this is the case !

Response: Agree and changes made. We have revised this sentence (RL402–406): "Studies on the MRB (Ma et al., 2018; Wang, 2007; Zhou, 1992) have shown that groundwater mainly recharged by the river fast leakage in the upstream area and piedmont plain, where the soil texture consists of pebbles and sandy gravel (Fig. 1c); this suggests that the unsaturated zone air CFC closely follows that of the atmosphere, so the recharge time lag through the unsaturated zone is not considered."

42) L317: Shortly explain the method used to estimate the tritium input (linear regression ? And how long were the time series used ?). The reference to Han et al. is not very useful as the authors of that paper themselves refer to an IAEA publication without further explanations.

Response: Agree and changes made. The explanation is seen in "Response 7". We added short statements in the revised manuscript as follows (RL234–237):

"The historical precipitation $^3$H activity in Urumqi station (Fig. 4) is reconstructed with the data available from the International Atomic Energy Agency (IAEA) using a logarithmic interpolation method. Precipitation $^3$H activity between 1969 and 1983 at Hong Kong and Irkutsk with different latitudes are used (data is available at <https://www.iaea.org/>)."

We think that the reconstructed tritium ($^3$H) activity in Urumqi station can be added as the supplement if necessary.

[Figure]

Figure 6. Tritium concentration (TU) of groundwater water samples of upstream groundwater (UG), midstream groundwater (MG), and downstream groundwater (DG). Time series of tritium concentration in precipitation at Ottawa, Urumqi, Hong Kong, and Irkutsk were obtained by GNIP in IAEA (https://www.iaea.org/). The blue solid lines and shaded field were drawn using the half–life (12.32 yrs) of tritium decayed to 2014. (It is Fig. 4 in the revised manuscript)

43) L318: A background of 31 TU is very high compared to Western Europe (about 6 TU). How come ?

Response: As shown in Fig. 4 and reconstructed tritium [3]H activity in Urumqi, the estimated modern precipitation [3]H activity are indeed extremely high (mean value of 31.55 TU between 2000 and 2015). We speculate that the very high precipitation [3]H activity may be ascribed to both the Chinese atmospheric nuclear tests (around 350 km away to Manas River Basin) from 1964 to 1974 (Zhou, 1992) and the continental effect (Tadros et al., 2014), where the Manas River Basin is more than 3500 km away from the western pacific.

44) L413: What do you mean by "serious" ?

45) L414: "tend toward more discrete with their increase". I do not understand this part of the sentence.

Response: Changes made. The spelling error "serious" was changed to "series". The sentence was changed to (RL468–469): "Figure 11 shows that different LPMs yield different MRTs for the same time series of [3]H and CFC concentrations. MRTs obtained from different LPMs tend to become more discretized by model with increasing MRTs."

46) L448: The paragraph on hydrochemistry is not bad, but underdeveloped and bad organized. State again what you're looking for first. A good correlation between hydrochemistry and "ages" calculated using some of the TTD models might be a way to constrain or guide model choice, but the authors do not really state that explicitly, although that would be interesting and relatively new.

Response: Yes, we agree with your opinion on the relationship between hydrochemistry and "ages"

calculated using the TTD models. Thanks very much, the suggestion of using hydrochemistry to guide model choice is definitely interesting and relatively new. We will bear in mind in the future studies in the Manas River Basin. In the present study, we tend to seek the correlations between groundwater MRTs and hydrochemical concentrations firstly.

47) L491: The entire chapter 4.5.1 makes no sense. You must first decide which model is the most appropriate, and then calculate metrics such as mean transit time, young water fraction, etc… You cannot both calculate water fractions using a binary mixing strategy (assuming piston-flow) AND later use an EPM. The same remark applied to chapter 4.5.2.

Response: We agree with your opinions that "first decide which model is the most appropriate, and then calculate metrics such as mean transit time, young water fraction, etc…". We think that the application of mixing models (even combining two lumped–parameter models) is a good method to quantitatively analysis groundwater mixing ratios, and this method is also used more and more widespread. However, CFCs are also good tracers to distinguish groundwater mixing from the different recharge sources, like to recognize modern and paleo–meteoric recharge features, to distinguish the fraction of young groundwater. We have totally revised the entire chapter 4.5.1 and reorganized the structure of the manuscript (See "Response 3"). The erroneous phrases and expressions including apparent CFC age and groundwater ages estimated from the piston flow model have been deleted.

48) L498: "no post-1988.5". Please round this off…

Response: Agree and changes made (RL370–371). "no post–1988.5" was changed to "no post–1989". "1989.5" was changed to "1990".

49) L509: Why do you treat "apparent age" as some kind of different measure of transit time than MTTs "estimated from the EPM" ? This is doing the analysis the wrong way around. First find a way to select a model, then discuss the obtained "ages" instead of hypothesizing on tons and tons of different "ages" that are meaningless because they were obtained disregarding the actual situation. This leads nowhere.

Response: Agree and changes made. The erroneous sentence and term "apparent CFC ages" been deleted. We reorganized the structure and revised the relative content to tried to present a clear roadmap to readers.

50) L541: Before engaging in complicated mixing scenarios, you should first try to find one model and one parameterisation that fits both the CFCs, tritium and carbon 14. Only if that search does not succeed should additional mixing be introduced. Please note that the binary mixture approach by Plummer et al. is only one way of doing so, and a particularly weak one at that because it assumes per default a piston-flow distribution of transit time of each component (other models can be integrated, but it becomes quickly very cumbersome). Another way to include the mixing of different reservoirs is to combine models (say two exponential models, each representing one distinct source) following Piotr Maloszewski and coworkers or Mike Stewart and Uwe Morgenstern. Binary plots such as those of figure 11 suffer from the limitation that you have to recalculate the mixing line for each parameterisation of each model, and they cannot really replace a multi-tracer lumped-parameter modelling approach, where the objective function reduces simultaneously the prediction error of all tracers.

Response: Thanks very much for the precious suggestions you offered. We believe that the modelling

approaches are commendable to interpret complicated mixing scenarios quantitatively. For example, the binary parallel lumped–parameter model (Morgenstern et al., 2015; Stewart et al., 2017), the binary mixing model that followed by two response function models (Jurgens et al., 2012), and the mathematical solutions that indicating the changes in water reserve in the relatively large systems with wide range of residence times (Custodio et al., 2018; Custodio and Custodio–Ayala, 2014). In addition, tracer–tracer binary plots are also widely used methods to determine the groundwater recharge mechanisms and to interpret the groundwater mixing (Cook et al., 2017; Darling et al., 2012; Han et al., 2015; Kagabu et al., 2017). We think that combining $^3$H and CFCs is a good tool to distinguish the modern precipitation recharge and to indicate the groundwater mixing properties (for example mixing old with young water, mixing irrigation infiltration water and young water) in the Manas River Basin, which has not been reported before. Recharge features (e.g. modern and paleo–meteoric recharge features, and the fractions of modern recharge water) are also the essential contents in the manuscript, which will not be all on account of the mean residence times estimated by $^3$H and CFCs in the revised manuscript. Further investigation work will be carried out in the Manas River Basin in the following several years (for example, time serious groundwater CFCs from some given wells will be measured). Deeper analysis concerning groundwater mixing obtained by the combination of two models using lumped–parameter models (binary mixing model) and to explore the changes in water reserve will be good choices in the next studies in the Manas River Basin.

51) L562: Solutions are obtained, explanations are devised.

Response: Thanks, no change made.

52) L572: What are mixing rates ? You mean mixing ratios ?

Response: Agree and changes made (RL544). "rates" was changed to "ratios"

53) L575: "The thrust faults were found to play a paramount role on groundwater flow path". There are not conclusions, but hypotheses very weakly suggested by the analysis of the environmental tracers, which is itself very shaky. I hardly call that evidence. Please refrain from drawing conclusions if the data necessary to test hypotheses is not available (as is the case here).

Response: Agree and changes made. This sentence has been deleted. To make the conclusions to be more clear and well-founded, we have revised the conclusions and delete some incorrect statements. Yes, this conclusion is important but not supported by strong supporting evidences in the paper. Indeed, there are some results that show large differences on both sides of the thrust fault. For example, there is a level difference of 130 m hydraulic drop (Fig. 1c) in the south margin in Shihezi (SHZ), $^3$H activities of groundwater decrease rapidly along the Manas River motion in the north of the fault but show relatively the highest values in the south of the fault (Fig. 8). These results still cannot support the conclusion explicitly "The thrust fault were found to play a paramount role on groundwater flow paths …".

The revised conclusions are as follows (RL543–555):

"In this study, we used environmental tracers and hydrochemistry to identify the modern and paleo–meteoric recharge sources, to constrain the different end–members mixing rates, and study the mixed groundwater MRTs in faulted–hydraulic drop alluvium aquifer systems. The paleo–meteoric recharge in a cooler climate was distinguished from the lateral flow from the higher elevation precipitation in the

Manas River downstream area. The relatively modern groundwater with young (post–1940) water fractions of 87–100 % was obtained, indicating a small extent of mixing south of the fault. The short MRTs (19 yrs) along with the higher–than–natural $NO_3^-$ concentration (7.86 mg $L^{-1}$) south of the fault (headwater area) indicated the invasion of modern contaminants. This finding warrants particular attention. High mixing rate amplitudes varying from 12 to 91 % were widespread in north of the fault due to the varying depths of long–screened boreholes as well as within the aquifer itself. Furthermore, the mixing diversity was highlighted by the substantial water table fluctuations during groundwater pumping, vertical recharge through the thick unsaturated zone, and young water mixtures in different decades. The strong correlations between groundwater MRTs and hydrochemical concentrations enable a first–order proxy at different times to be used. In addition, this study has revealed that MRTs estimated by CFCs were more appropriate than those using $^3H$ in the arid MRB with a thick unsaturated zone."

54) L585: "due to the highly complex groundwater system...". This is no explanation at all ! Indeed, devising a conceptual model that could explain why CFC derived "ages" correlate well with mineralisation while tritium derived "ages" do not could be a useful task (but you should first redo the calculation of the "ages" as suggested above). On the one hand, the correlation between CFC12 and hydrochemistry might be an artifact, given that the area sampled is so large and hydrogeologically diverse. On the other hand, there might be some kind of systematic shift between tritium and CFC ages if differences are due to the unsaturated zone. Maybe a diffusion model using the unsaturated zone thickness might be useful. Still much work to do…

Response: The sentence has been deleted. Thanks very much for the useful suggestions. Previous studies (Ji, 2016; Wang, 2007; Zhou, 1992) have shown that the groundwater flow paths were very complicated from the mountain to the plain areas. Precipitation recharge in the ground in the mountain areas, one part groundwater discharge into spring in the south of the intermountain depression, one part groundwater discharge into stream in the mountain areas and then recharge groundwater in the intermountain depression, and one part groundwater flow in the ground through the intermountain depression. The complicated groundwater flow systems make devising a conceptual model very difficult to implement. We think that more and much detailed work should be conducted from the mountain to intermountain depression areas to find out more evidences that interpretation the conceptual flow model. In addition, the thick unsaturated zone mainly distribute in the intermountain depression and piedmont plain areas (Fig. 1c). The interpretation of CFCs in Section 4.2.2 make us to assume that the unsaturated zone air CFC closely follows that of the atmosphere and thus the recharge time lag through the unsaturated zone is not consideration. In some cases the unsaturated zone can be ignored to obtain workable solutions (Custodio et al., 2018) though the unsaturated zone is not so thick with the Manas River Basin. We think that a diffusion model using the unsaturated zone thickness is a good guide for the further studies in the arid Manas River Basin as well as in other arid basins around the world.

55) Figure 7, 8 and 9: The figures are incredibly cluttered and very difficult to read, especially figure 9 (not to mention the legend).

Response: Agree and changes made. Figures 7, 8 and 9 were redrawn as follows:

[Figure]

**Figure 7.** Distributions of $^3$H and $^{14}$C activities with distance to mountain. The shaded regions indicate the upstream, midstream and downstream of Manas River. (It is Fig. 8 in the revised manuscript)

[Figure]

**Figure 8.** Tritium and CFCs (CFC–11, CFC–12 and CFC–113) output vs. mean residence times for different lumped–parameter models estimated using Eqs. (2) to (5). The input $^3$H activity and CFCs concentration are using the estimated $^3$H activities in precipitation in Urumqi station (Fig. 4) and the Northern Hemisphere atmospheric mixing ratio (Fig. 3), respectively. (It is Fig. 10 in the revised manuscript)

[Figure]

**Figure 9.** (**a**) MRTs with EPM (1.5) of CFC–12 vs. CFC–11 & CFC–113, (**b**) CFC–12 MRTs with EPM (1.5) vs. EPM (2.2), DM & EMM, and (**c**) 3H MRTs with EPM (1.5) vs. EPM (2.2), DM & EMM. (It is Fig. 11 in the revised manuscript)

56) Figure 10: Why are there so few points on each graph, since you sampled at 29 locations according to table 1 ?

Response: Figure 10 shows that the x-axis represents the CFC–12 mean residence times with EPM (1.5), and Table 1 shows that there are 9 groundwater CFC–12 samples. For the sample G15, the hydrochemistry show abnormal data with much higher concentrations of $SO_4^{2-}$, $HCO_3^-$, and electrical conductivity than that of perimeter zone. Moreover, in the plot of CFC–12 mean residence times vs. hydrochemistry, G15 sample distributes abnormal. Therefore, only 8 groundwater samples are potted in Fig. 10 (It is Fig. 12 in the revised manuscript).

57) Figure 11: As I wrote above, binary mixing diagrams rapidly tend to show their limits. After two or three mixing lines for different models are drawn, reading becomes nigh impossible. Importantly, error bars are missing for the CFCs and for tritium. I suspect that with error bars, selecting a model visually will become impossible (the lack of sensitivity is another limitation of binary mixing diagrams, ).

Response: Agree and changes made. We deleted the mean residence times with different models but retain the measured CFC concentrations and $^3$H activities of groundwater in the diagrams.

[revised manuscript text omitted]

$$g(\tau) = \frac{\eta}{\tau_{\mathrm{m}}} e^{(-\eta\tau/\tau_{\mathrm{m}} + \eta - 1)} \qquad \text{for} \quad \tau \geq \tau_{\mathrm{m}}(1 - 1/\eta) \tag{2b3b}$$

The dispersion model (DM) mainly measures the relative importance of dispersion to advection, and is applicable for confined or partially confined aquifers (Małoszewski, 2000). Its RTD is given by

305
$$g(\tau) = \frac{1}{\tau\sqrt{4\pi D_{\mathrm{P}}\tau/\tau_{\mathrm{m}}}} e^{-\left(\frac{(1 - \tau/\tau_{\mathrm{m}})^2}{4\pi D_{\mathrm{P}}\tau/\tau_{\mathrm{m}}}\right)} \tag{34}$$

The weighting function of the exponential mixing model (EMM) is

$$g(\tau) = \frac{1}{\tau_{\mathrm{m}}} e^{(-\tau/\tau_{\mathrm{m}})}, \tag{45}$$

where $\tau_{\mathrm{m}}$ is the mean residence time, $\eta$ is the ratio defined as $\eta = (l_{\mathrm{P}} + l_{\mathrm{E}})/l_{\mathrm{E}} = l_{\mathrm{P}}/l_{\mathrm{E}} + 1$, where $l_{\mathrm{E}}$ (or $l_{\mathrm{
[revised manuscript text omitted]

---

## Author Response (AR2)

**Responses to Editor and Reviewers**

To:

Prof. Christine Stumpp

Editor

Hydrology and Earth System Sciences

Dear Prof. Stumpp,

We gratefully acknowledge the two reviewers for their helpful and insightful comments to improve this manuscript. We are submitting the revised manuscript titled "Application of environmental tracers for investigation of groundwater mean residence time and aquifer recharge in fault–influenced hydraulic drop alluvium aquifers" (HESS-2018-143) to ***Hydrology and Earth System Sciences***. The authors have completed revisions on the previous manuscript and have addressed a point-by-point reply to the comments below (Response in blue).

Author's response – Line numbers referring to the old and revised version manuscripts are preceded by L and RL, respectively.

**Editor comments**

The same two reviewers had a closer look at the revised versions. Both are experts in the research field and have different recommendations on the acceptance of the manuscript. I still see the potential that the others can revise the manuscript and that it can be a valuable contribution. It is absolutely necessary to particularly:

1) take into consideration the minor comments of reviewer #1 (see comment and attached file).

Response: We would like to express our sincere gratitude to you for reviewing this paper. The point-by-point reply to the minor comments of reviewer #1 can be seen in "Response to **Anonymous Referee #1**".

2) do a thorough language check by a native speaker or use a public service

Response: Agree. We have checked and revised the manuscript carefully and seriously. We also have asked a public service for help to do a thorough language check. Certificate as shown below:

**Certificate of Seeditors**

This document certifies that the paper listed below has been edited to ensure that the language is clear and free of errors. The edit was performed by professional editors at Seeditors, a division of Delfue Communications. The intent of the author's message was not altered in any way during the editing process. The quality of the edit has been guaranteed, with the assumption that our suggested changes have been accepted and have not been further altered without the knowledge of our editors.

**Title of the Paper**

*Application of environmental tracers for investigation of groundwater mean residence time and aquifer recharge in fault–influenced hydraulic drop alluvium aquifers*

**Ref. Number**

SED01038

*Signature of responsible editor:*

[Figure]

*Date of Issue* : **15 Dec. 2018**

[Figure]

[Figure]

Seeditors is based in the Netherlands, and has created an online workplace for the world -connecting clients with top freelance professionals and experienced small companies from Europe to North America.

**Xuefeng Zhu**

Dr. Xuefeng Zhu
Chief President , Seeditors

Lepelaarstraat 7, 2623NW, Delft, the Netherlands;
**KVK**:69163677;
**T** +31(0) 616828836;
**E** support@seeditors.com;
**W** www.seeditors.com.

3) go back to Maloszewski et al. 1983 and carefully check the assumptions and the physical meaning for the different functions used in the manuscript and adapt interpretation of results accordingly (see comment reviewer #2)

Response: Agree. Response can be seen in "Response to **Anonymous Referee #2**, Specific comments".

4) consider clarification about age and apparent age.

Response: Agree. Groundwater age cannot be measured directly. Hydrogeologists use "groundwater age" to denote the time since recharge occurred or to indicate the time difference that "a water parcel needs to travel from the groundwater surface to the position where the sample is taken (i. e. *idealized groundwater age*; Suckow, 2014)". A piston flow is premised that without mixture or non-dispersive flow along the flow paths since recharge (Cartwright et al., 2017; de Dreuz and Ginn, 2016; Suckow, 2014). However, water mixing occurs at anytime and anywhere during groundwater flow processes. The water sample taken from a well or a spring is all the time a mixture of groundwater with different ages. In our manuscript, a lumped parameter model is used to describe the distribution of residence times and at the same time to estimate the mean residence time. In Eq. (2) in the manuscript, residence time is completely equivalent to the "groundwater age" and represents the time difference ($\tau$) between input time ($t\text{-}\tau$) and time of sampling $t$. Equation (2) attributes a weight ($g(\tau)$) to each of the residence times that describes to which percentage a residence time contributes to the whole mean.

$$C_{out}(t) = \int_0^\infty C_{in}(t-\tau) g(\tau) e^{-\lambda_{^3H} \tau} d\tau \qquad \text{for } {}^3H \text{ tracer} \qquad (2a)$$

$$C_{out}(t) = \int_0^\infty C_{in}(t-\tau) g(\tau) d\tau \qquad \text{for CFCs tracer} \qquad (2b)$$

Since groundwater age cannot be measured directly, one can derive a groundwater age using a mathematical formula for some tracers. As the tracer concentrations (or isotope ratios, etc.) are measured based on real-world samples, the derived groundwater age is different from the residence time mentioned above. Furthermore, the age distribution in the sample is unknown, it is not possible to use Eq. (2) to estimate the mean residence time of the sample. Therefore, the widely used "apparent age" in the literatures is obtained when a mathematical formula is given and at the same time the tracer is stated. Suck (2014) emphasized that the applicable tracers to estimate the apparent ages are ${}^{14}C$, ${}^3H/{}^3He$, ${}^{36}Cl$, and ${}^{81}Kr$, rather than the CFCs, $SF_6$, and ${}^{85}Kr$.

To clarify the difference between age and apparent age, statements were added in the revised manuscript for the apparent ${}^{14}C$ ages (RL213–216) and the mean residence times (RL242–250):

RL213–216: Since groundwater age cannot be measured directly, and the age distribution in the sample is unknown, one can derive an apparent age using a mathematical formula for the groundwater ${}^{14}C$ sample (Suckow, 2014). "Apparent" here describe the fact that the age is not corresponding to the time difference between recharge and sampling during which piston flow is assumed for a water parcel (Cartwright et al., 2017; Suckow, 2014).

RL242–250: A wide range of the groundwater residence times (ages) has been reported in an arid unconfined aquifer because recharge occurs under various climate conditions (Custodio et al., 2018). Furthermore, the groundwater residence time with wide variabilities that governed by the distribution of

flow paths of varying length cannot be measured directly (de Dreuz and Ginn, 2016; Suckow, 2014). A lumped parameter model may be an alternative approach to describe the distribution of residence times, which at the same time describes a mean residence time for the mixtures of different residence times. With the aid of gaseous tracers (e.g. $^{3}$H, CFCs, $SF_6$ and $^{85}$Kr) one can describe the distribution of tracer concentrations (Stewart et al., 2017; Zuber et al., 2005) to obtain the groundwater MRTs.

If the authors cannot consider these fundamental changes, I have to reject the manuscript.

Response: All of the comments are accepted. We have addressed all of these fundamental changes and have indicated how we will incorporate the valuable suggestions in the revised manuscript.

**Anonymous Referee #1**

**General Comments**

1)   This version of the paper is better organised and clearer than the previous version. In particular, the revised Figs. 8 and 9 (now 10 and 11) are much better. Using CFC mixing model ages for interpretation (instead of the problematical CFC apparent ages) is also a big improvement. As before, I find that the paper addresses relevant scientific questions suitable for publication in HESS, with novel concepts and ideas. Substantial conclusions are reached. However, there are still problems with the English. Some suggestions for improvement in clarity are given below.

How are the mixing model calculations carried out? No information is given in the Methods Section. Are they made using the TracerLPM program of Jurgens et al. (2012)? Or have the authors developed their own program? We need to know this to assess the mixing ages.

Response: The mixing model calculation processes are as follows:

Take the CFC–12 and exponential–piston flow model (1.5) for example, the calculation process of the mean residence times is as follows: First, we chose the Eq. (2b) as the convolution integral, and chose the weighting function of the exponential–piston flow model (Eq. (3a, 3b)) as the system response function. Second, we used the time series CFC–12 trend of the Northern Hemisphere atmospheric mixing ratio (1940–2014, http://water.usgs.gov/lab/software/air/cure/) as input concentrations. We treated the calendar year 2015 (groundwater sampling time) as age=0 yrs by convoluting the input (times series of CFC–12 input) to the EPM (1.5). The mean residence times $\tau_\mathrm{m}$ increased from 1 to 100 yrs with time step of 1 yr and increased from 101 to 500 yrs with time step of 5 yrs. Then we got a sequence results of output CFC–12 concentrations and mean residence times (vary from 1 to 500 yrs). Third, we plotted the output CFC–12 concentrations vs. mean residence times and then compared the measured groundwater CFC–12 concentrations to get the groundwater mean residence times. The computational procedures were carried out by using MATLAB (version R2014a). The output CFC–12 concentrations decreased from 526.4 pptv (with $\tau_\mathrm{m}$ of 1 yr) to 3.0 pptv (with $\tau_\mathrm{m}$ of 155 yrs). As the detection limit for each CFC is about 0.01 pmoL$^{-1}$ of water (be equal to 3.54 pptv with the laboratory temperature of 25℃), the output CFC–12 concentrations lower than 3.54 pptv can be neglected.

2)   Use of acronyms is effective (and widely practised) but should be moderate, otherwise it makes

difficulties for readers. I think MRB (Manas River Basin), MRT (mean residence time), EPM (exponential piston flow model), DM (dispersion model) and EMM (exponential mixing model) as used here are ok, but LPM and RTD should be spelled out wherever they are used (i.e. replaced by 'lumped parameter model' and 'residence time distribution').

Response: Agree and changes made. Manas River Basin (MRB), mean residence times (MRTs), exponential piston flow model (EPM), dispersion model (DM) and exponential mixing model (EMM) are used when they first appear. The abbreviations are used in the following context. "lumped parameter model" and "residence time distribution" are used throughout the manuscript.

**Detailed comments**

1) L44-48 This is a very muddled sentence and needs to be rewritten. Not all the radioisotopes mentioned have long half-lives. And the gases (CFCs and SF6) are not radioisotopes, nor do they have specific half-lives.

Response: Agree and changes made (RL52–57). The sentence was revised as: Second, the atmospheric concentrations of synthetic organic compounds (chlorofluorocarbons, CFC–11, CFC–12, and CFC–113; and sulfur hexafluoride, $SF_6$), radioactive solute tracers such as $^{14}C$, $^{36}Cl$, and noble gases ($^{4}He$, $^{85}Kr$, $^{39}Ar$, and $^{81}Kr$), are used to determine groundwater MRTs with much wider time spans (decades to hundred millenniums; Aggarwal, 2013).

2) L75 Delete surplus "be"

Response: Agree and changes made (RL85). "be" was deleted.

3) L76 Change to ".. : for example, MRTs estimated from CFCs would be much smaller than actual values if excess air in the unsaturated zone affected CFC concentrations during recharge (Cook et al., 2006; .."

Response: Agree and changes made (RL85–89). The sentence was revised as: However, groundwater MRTs may be not always accurate based on CFCs. For example, MRTs estimated from CFCs would be underestimated if excess air in the unsaturated zone affects the CFC concentrations during recharge (Cook et al., 2006; Darling et al., 2012), or when CFC inputs are contaminated in urban and industrial environments (Carlson et al., 2011; Han et al., 2007; Mahlknecht et al., 2017; Qin et al., 2007).

4) L87 Change to "Mixing within the aquifers .. long-screened wells is expected to be common .."

Response: Agree and changes made (RL98–100). The sentence was revised as: Mixing within the aquifers and during the pumping process from the long–screened wells is expected to be common in the fault–influenced hydraulic drop alluvium aquifers of the Manas River Basin (MRB) in the arid Northwest China (Fig. 1a, b).

5) L90 Change to ".. (with water table depths of up to 180 m) .."

Response: Agree and changes made (RL101–103). The sentence was changed to: MRTs that result from a deep unsaturated zone (with water table depths of up to 180 m) and contrasting geological settings (hydraulic head drops of as much as 130 m caused by the thrust fault) are still insufficiently

recognised in the alluvium aquifer (Fig. 1c).

6) L122 Change to ".. 29 groundwater samples (pumped from fully penetrating wells, 3 springs or 3 artesian .."

Response: Agree and changes made (RL135–137). The sentence was revised as: A total of 29 groundwater samples (pumped from fully penetrating wells, 3 springs or 3 n wells) were collected along the Manas River between June and August 2015 (from G1 to G29 in Table 1 and Fig. 2).

7) L127 "minutes" not "min"

Response: Agree and changes made (RL141). "min" was replaced by "minutes".

8) L182 Change to "The computational .." by omitting "Concrete"

Response: Agree and changes made (RL200). The sentence was changed to" The computational process was conducted following Plummer et al. (2006a)".

9) L216 "although" not "despite"

Response: Agree and changes made (RL238). "despite" was changed to "although".

10) L223 Better to spell out LPMs (i.e. lumped parameter models) here, and wherever else it occurs.

Response: Agree and changes made (RL245). "LPMs" was changed to "Lumped parameter models" and we insist on the "lumped parameter models" throughout the manuscript.

11) L245 Also RTDs (residence time distributions)

Response: Agree and changes made (RL274). "RTDs" was changed to "residence time distributions" and we insist on the "residence time distributions" throughout the manuscript.

12) L268 "compared" not "cross-referenced"

Response: Agree and changes made (RL297). "cross-referenced" was changed to "compared".

13) L301 ".. to one or both of two recharge .." not ".. to two recharge .."

Response: Agree and changes made (RL331–333). The sentence was changed to: Groundwater whose isotopic values are more depleted than the modern precipitation usually would be ascribed to one or both of two recharge sources including snowmelt/precipitation at higher elevation and precipitation fallen during cooler climate.

14) L313 What do the authors mean by "qualitative recharge"? Rephrase.

Response: Agree and changes made (RL343–346). The sentence was changed to: An overlap between surface water and UG indicates the same recharge sources, because some alignment of river water and groundwater isotopic values is a qualitative indication of recharge under climate conditions similar to contemporary conditions (Huang et al., 2017).

15) L389 Use "but not" instead of "rather than"

Response: Agree and changes made (RL428). "rather than" was changed to "but not".

16) L403 "groundwater is mainly recharged by fast river leakage" not "groundwater mainly recharged

by the river fast leakage"

Response: Agree and changes made (RL440–442). The sentence was revised as: Studies on the MRB (Ma et al., 2018; Wang, 2007; Zhou, 1992) have shown that groundwater is mainly recharged by fast river leakage in the upstream area and piedmont plain, where the soil texture consists of pebbles and sandy gravel (Fig. 1c).

17) L414 Add words. "indicated input of some fractions" not "indicated some fractions"

Response: Agree and changes made (RL450–451). The sentence was changed to: All of the $^3$H values in UG (G1, G2, and G4) and G23 (belonging to MG) are higher than 34.3 TU, which indicate input of some fractions of the 1960s precipitation recharge.

18) L462-464 Unclear sentence. "But river leakage and rainfall input could have come only from the piedmont plain (Ma et al., 2018), thus a smaller proportion of piston flow in the EPM could give an EPM ratio of 2.2 ($I_E$ in Eq. (3) would only be for the piedmont plain in Fig. 1c)." not "River leakage and rainfall input were possible from the piedmont plain (Ma et al., 2018), thus a less proportion of piston flow by the EPM with an EPM ratio of 2.2 ($I_E$ in Eq. (3) is only in the piedmont plain in Fig. 1c) was also used."

Response: Agree and changes made (RL502–505). The sentence was revised as: But the river leakage and rainfall input could have come only from the piedmont plain (Ma et al., 2018), thus a smaller proportion of piston flow in the EPM could give an EPM ratio of 2.2 ($I_E$ in Eq. (3) would only be for the piedmont plain in Fig. 1c).

19) L499-500 Unclear sentence. "Nevertheless, the homogeneous aquifers, being at steady–state, justify the use of LPMs to calculate MRTs in this study." Not "Nevertheless, in this study the homogeneous aquifers, being at steady–state, justifying the use of LPMs to calculate MRTs."

Response: Agree and changes made (RL557–559). The sentence was revised as: Nevertheless, the homogeneous aquifers, being at steady–state, justify the use of lumped parameter models to calculate MRTs in this study.

20) L503 "closed" not "close"

Response: Changes made (RL563). "close" was changed to "similar".

21) L504 "better" not "higher"

Response: Agree and changes made (RL564). "higher" was replaced by "better".

22) L519 "MRT" not "MTT"

Response: Agree and changes made (RL579). "MTT" was changed to "MRT".

23) L539 "On the other hand" not "However"

Response: Agree and changes made (RL599). "However" was replaced by "On the other hand".

24) L552 "and different young water inputs in different decades" not "and young water mixtures in different decades"

Response: Agree and changes made (RL617–619). The sentence was revised as: Furthermore, the mixing diversity is highlighted by the substantial water table fluctuations during groundwater pumping,

vertical recharge through the thick unsaturated zone, and different young water inputs in different decades.

25) Table 1 caption. Change order to match the columns in the table. ".. stable isotopes, CFCs, tritium .."

Response: Agree and changes made (RL814). Table 1 caption was revised as: **Table 1.** Chemical–physical parameters, stable isotopes, CFC concentrations, tritium ($^3$H), and $^{14}$C in groundwater samples in the Manas River Basin.

26) Table 2 caption. Change order to match columns and add words at end. ".. partial pressure (pptv), fraction of post-1940 water, modern precipitation recharge year, and mean residence times based on different lumped parameter models (EPM, DM and EMM)."

Response: Agree and changes made (RL816). Table 2 caption was revised as: **Table 2.** Calculated results for CFC atmospheric partial pressure (pptv), fraction of post–1940 water, modern precipitation recharge year, and mean residence times (EPM, DM and EMM).

27) Figure 11 caption has too much jargon. Change to "(a) Mean residence times (MRTs) for CFC-12 vs. MRTs for CFC-11 and CFC-113 data using the EPM (1.5) model. (b) MRTs for CFC-12 with EPM (1.5) vs. those with other models. (c) MRTs for $^#$H vs. those with other models.

Response: Agree and changes made (RL878). Figure 11 caption was revised as: **Figure 11.** (**a**) Mean residence times (MRTs) for CFC–12 vs. MRTs for CFC–11 and CFC–113 data using the EPM (1.5) model. (**b**) MRTs for CFC–12 with EPM (1.5) vs. those with other models. (**c**) MRTs for $^3$H vs. those with other models.

28) Figure 12(a) and (b) The x-axes should have "MRT" not "MTT".

Response: Agree and changes made (RL885). Figure 12 was revised as:

[Figure]

29) Figure 12 caption. Change to "(a) pH and silica ($SiO_2$) and (b) sulfate ($SO_4$), bicarbonate ($HCO_3$), and total dissolved solids (TDS) vs. mean residence times (MRTs). The MRTs are from CFC-12 data using the EPM (1.5) model. The dashed red line .."

Response: Agree and changes made (RL888). Figure 12 caption was revised as: **Figure 12. (a)** pH and silica ($SiO_2$) and **(b)** sulfate ($SO_4^{2-}$), bicarbonate ($HCO_3^-$), and total dissolved solids (TDS) vs. mean residence times (MRTs). The MRTs are from CFC–12 data using the EPM (1.5) model. The dashed red line in (**a**) is from Morgenstern et al. (2015).

**Anonymous Referee #2**

**General Comments**

The authors have obviously taken great pain to respond to the reviewers' criticisms constructively. They have failed completely however to address the fundamental issues raised by the reviewers. Changes are numerous and quite substantial, but superficial. Hence, I still stand to my initial assessment that the authors have written a relatively well done case study that however falls completely short of any kind of originality or innovativeness. The manuscript imitates previous works, but even in what the authors have shown of the revised version, it still suffers from its lack of focus. Estimating groundwater residence times from multiple tracers has been done and redone so many times and is not a sufficient reason for a publication in HESS, except if a methodological breakthough of sorts can be presented (this is not at all the case here).

Furthermore, the authors are STILL inconsistent in the use they make of the weighting functions, presenting three potential models (EPM, EMM and DM) in the method section, but retaining the piston-flow model for the analysis of the carbon 14 data. If anything, this shows that the authors have not fundamentally modified their initial (erroneous) analysis, although prompted to do so by the reviewers.

Response: We sincerely thank you for taking the time to review our manuscript and for your insightful comments. We think that it is not contradictory to estimate groundwater mean residence time using a lumped parameter model with different weighting functions and to determine groundwater $^{14}$C apparent age using a mathematical formula. As shown in the "Response to Editor comments 4", apparent age is derived from the $^{14}$C tracer. Because the apparent $^{14}$C age is derived from a real-world sample, in which the age distribution in the sample is unknown and the sample is an aggregation of many piston flow lines of water. We treated the apparent $^{14}$C ages with caution. Therefore, we did not compare the apparent $^{14}$C ages with mean residence times that derived from a lumped parameter model with different weighting functions (EPM, EMM, and DM). Furthermore, in the revised manuscript, groundwater $^{14}$C activity is used to help to investigate groundwater recharge features (i.e. the modern and paleo-meteoric recharge features).

**Specific comments**

1)    Remark 3: One can always argue that any kind of environmental or hydrological study is "essential" because of human induced pressures, climate change, necessity to provide policy support data, etc… What I meant to say was however much more simply that I do not see anything new or innovative in the manuscript. I still do not. In particular, the authors reproduce the overall shortcomings of many recent publications on multi-tracer groundwater studies, where the multiple tracer measurements are not used to derive one of more consistent parameterised models (see Maloszewski et al., Journal of Hydrology 66, 1983 for a good example), but instead juggle more or less successfully between different estimates of the mean transit time obtained from each tracer, sometimes comparing the results of models assuming incompatible weighting functions.

Response: Both the study by Małoszewski et al. (1983) and many recent studies (i.e. Cartwright and Morgenstern, 2015, 2016; Cartwright et al., 2018; Han et al., 2015; Morgenstern and Daughney, 2012; Morgenstern et al., 2015) have determined groundwater mean residence time (or mean transit time) using a lumped parameter model. The mean residence times were determined based on different response functions (EPM, EMM, and DM) that describe the residence time distributions. In fact, the earlier study by Małoszewski and Zuber (1982, 1996), as well as the studies in recent years (i.e. Jurgens et al., 2012), have systematically summarized possible applicability of particular models (like piston flow model, exponential mixing model, exponential–piston flow model, dispersion model, etc.) in the different subsurface flow systems. The fact is that lumped parameter model premises that the idealized flow system is at a steady state, which differs from the actual hydrogeological conditions. Integrated application of the different response functions mentioned above is reasonably to determine groundwater mean residence time. Małoszewski et al. (1983) pointed out that "*applications are limited to the frequently used exponential (EM) and dispersive model (DM)*" in their paper. Recent studies by Cartwright and Morgenstern (2015, 2016) compared the mean residence times determined from different models (EPM, EMM, and DM). Morgenstern et al. (2010) obtained an excellent correlation between silica ($SiO_2$) and mean residence time ($R^2$=0.997) in their paper, where an exponential–piston flow model with an exponential fraction of 80% was used (rather than the exponential fraction of 70% and 90%). Examples mentioned above help us to get a more comprehensive understanding of the mean residence time in a hydrological system by comparing the different models. In our manuscript, exponential–piston flow model, exponential mixing model, and dispersion model were used because these three models are all possible applicability for the hydrological systems in the Manas River Basin. We believe that the response functions used in our manuscript (EPM, EMM, and DM) are considered carefully and that the corresponding mean residence times are proper to be explained.

2)    Remark 4: "The groundwater residence times (ages) often display a wide range due to the recharge under various climate conditions". This is correct, but only in a very particular case of arid to hyperarid environments having known a wetter period. In temperate climates, the distribution of transit time at the outlet of a groundwater system is due to the differences in length and flow time of the flow tubes contributing to discharge.

Response: Agree and changes made. The sentence was changed to (RL242–245): A wide range of the groundwater residence times (ages) has been reported in an arid unconfined aquifer because recharge occurs under various climate conditions (Custodio et al., 2018). Furthermore, the groundwater residence time with wide variabilities that governed by the distribution of flow paths of varying length cannot be measured directly (de Dreuz and Ginn, 2016; Suckow, 2014).

As we know that residence time varies both temporally following hydrological fluctuations and spatially following geological settings. Also, topographic and geomorphic conditions as well as evaporation intensity will influence the residence time. Custodio et al. (2018) reported a typical research on the residence time with wide ranges in a large unconfined aquifer, in which a wet period intercalated in an arid climate sequence having known. Furthermore, factors that change the groundwater flow paths may lead to variation of the residence time.

Remark 4: "The exponential–piston flow model (EPM) describes aquifer that containing a segment of

exponential flow followed by a segment of piston flow. Piston flow assumes that water mixing from different flow lines is minimal and receiving little or no recharge in the confined aquifer, and the exponential flow assume a full mixing of water in the unconfined aquifer and receiving areally distributed recharge". This description is not correct, and confirms my impression that the authors do not quite know what they are doing. The EPM does not prescribe the order in which the two components are physically arranged (this is explicitly explained in Maloszewski and Zuber, 1983), because mathematically, it makes no difference for the output concentration. And as far as the exponential model is concerned, mixing occurs at the sampling point and is only exactly true for semi-confined aquifers (see Haitjema, Journal of Hydrolog 172, 1995).

Response: The statement "The exponential–piston flow model (EPM) describes aquifer that containing a segment of exponential flow followed by a segment of piston flow" is used to describe the particular model (i.e. exponential–piston flow model), which shows us an imagination of the schematic diagram of the idealized aquifer configuration (Jurgens et al., 2012; Małoszewski and Zuber, 1982, 1996). In general, a groundwater flow pattern that unconfined aquifer receives vertical water recharge following confined aquifer without vertical water recharge is the most common from mountain to plain in the arid areas. Furthermore, mixing may occur both in the long screened well and at the sampling point for the exponential model, and it is also true for the unconfined aquifers (Jurgens et al., 2012; Małoszewski and Zuber, 1982, 1996).

Remark 4: "the MRTs with different RTDs were cross–referenced". What does that mean? Comparing the estimated mean transit times really leads nowhere. Either they are close and the authors will claim there is "satisfactory agreement", or they are not and an entire section of the discussion will be devoted to that. Model choice still is an open research question, so either the authors present a solid argument as to which model they chose, or some kind of methodology allowing to reject some models (as flawed as it is, the method by Plummer et al. is one example of that).

Response: Model choice should be based on the subsurface water flow system pattern as well as the hydrogeology setting. It is seen from Fig. 1 that the subsurface water flow system is controlled by the hydrogeological conditions and the geological settings (i.e. the thrust fault with block water feature). Aquifer recharge is mainly from the lateral flow in the mountain area as well as the fast river leakage in the intermountain depression and in the piedmont plain. Moreover, the river leakage occurs mainly along the river flow motion with small areas. Vertical recharge from the precipitation in the intermountain depression and in the piedmont plain is rare, and is little and even is none in the oasis plain in the arid Manas River Basin. Hence, there is not an exact idealized aquifer configuration (Jurgens et al., 2012; Małoszewski and Zuber, 1982, 1996) to be applied to groundwater samples in our study area. What we tend to do is comparing the mean residence times determined from different idealized models. It is also a choice to do so in other study areas (e.g. Cartwright and Morgenstern, 2015, 2016).

Remark 4: "In our manuscript, the apparent 14C ages estimation was adopted." I have already criticised this in the first review. "Apparent age" means you are assuming a piston-flow weighting function. This is not consistent with the method section presenting the EPM, the DM and the EMM as potential models describing the transit time distribution of the aquifer.

Response: Partly agree. In the revised manuscript, we added statement for the term "apparent age". The clarification for "Apparent [14]C ages" is seen in the "Response to editor comment 4". Apparent [14]C age is determined from a real-world sample using a defined formula (Eq. 1 in the manuscript). As the

distribution function for the residence times is unknown in the sample, it represents an unknown average of the residence times.

3)    Remark 50: "We think that combining 3H and CFCs is a good tool to distinguish the modern precipitation recharge and to indicate the groundwater mixing properties". I do not quite know what the authors mean by groundwater mixing properties, but the rest of the paragraph is not an answer to my point at all concerning the very limited use of binary mixing diagrams in multi-tracers studies. The authors just equivocate without actually trying to propose a counter argument.

Response: In the revised manuscript, diagram of $^3$H vs. CFCs helps to investigate recharge features due to the large difference of the temporal pattern in the input functions between CFCs and $^3$H. Compared with plots of tracer ratios, tracer–tracer concentration plots have some advantages because they reflect more directly the measured quantities and potential mixtures, such as mixing with irrigation water or young water mixtures in different decades (RL466–480).

4)    Remark 57: Figure 11 has been kept by the authors, who at the same time have deleted references to "apparent age" from their manuscript, but not from the analysis! Do they realize that the plots of figure 11, being calculated using a piston-flow model, show this very "apparent age" they agreed makes no sense?

Response: In the revised manuscript, Figure 11 (It is Fig. 7 in the revised manuscript) helps to identify samples containing young (post–1940) and old (CFC–free) water or exhibiting contamination or degradation. The young water fractions are also determined using the binary mixing method. It is nothing to do with "apparent age" in Fig. 7 in the revised manuscript.

[revised manuscript text omitted]